

**Review article: A systematic review of terrestrial dissolved organic carbon in northern**
**permafrost**
Liam Heffernan[1], Dolly N. Kothawala[1], Lars J. Tranvik[1]
[1]Limnology/Department of Ecology and Genetics, Uppsala University, Norbyvägen 18D,
Uppsala 75236, Sweden
Correspondence email: liam.heffernan@ebc.uu.se


## Abstract

As the permafrost region warms and permafrost soils thaw, vast pools of soil organic carbon (C) become vulnerable to enhanced microbial decomposition and lateral transport into aquatic ecosystems as dissolved organic carbon (DOC). The mobilization of permafrost soil C can drastically alter the net northern permafrost C budget. DOC entering aquatic ecosystems becomes biological available for degradation as well as other types of aquatic processing. However, it currently remains unclear which landscape characteristics are most relevant to consider in terms of predicting DOC concentrations entering aquatic systems from permafrost regions. Here, we conducted a systematic review of 111 studies relating to, or including, concentrations of DOC in terrestrial permafrost ecosystems in the northern circumpolar region published between 2000 – 2022. We present a new permafrost DOC dataset consisting of 2,276 DOC concentrations, collected from the top 3 m in permafrost soils across the northern circumpolar region. Concentrations of DOC ranged from $0.1 – 500$ mg L$^{-1}$ (median = 41 mg L$^{-1}$) across all permafrost zones, ecoregions, soil types, and thermal horizons. DOC concentrations were greatest in the sporadic permafrost zone (101 mg L$^{-1}$) while lower concentrations were found in the discontinuous (60 mg L$^{-1}$) and continuous (59 mg L$^{-1}$) permafrost zones. The highest median DOC concentrations of 66 mg L$^{-1}$ and 63 mg L$^{-1}$ were found in coastal tundra and permafrost bog ecosystems, respectively. Coastal tundra (130 mg L$^{-1}$), permafrost bogs (78 mg L$^{-1}$), and permafrost wetlands (57 mg L$^{-1}$) had the highest DOC concentrations in the permafrost lens, representing a potentially long-term store of DOC. Other than in Yedoma ecosystems, DOC concentrations were found to increase following permafrost thaw and were highly constrained by total dissolved nitrogen concentrations. This systematic review highlights how DOC concentrations differ between organic- or mineral-rich deposits across the circumpolar permafrost region and identifies coastal tundra regions as areas of potentially important DOC mobilization. The quantity of permafrost-derived DOC exported laterally to aquatic ecosystems is an important step for predicting its vulnerability to decomposition.





## 1. Introduction

Persistent freezing temperatures since the late Pleistocene and Holocene has led to the
accumulation and preservation of 1,460 – 1,600 Pg of organic carbon (C) in northern
circumpolar permafrost soils (Hugelius et al., 2014; Schuur et al., 2018). However, in recent
decades, there has been an amplified level of warming at high latitudes, occurring at four-times
the speed of the global average (Rantanen et al., 2021). This is leading to widespread and rapid
permafrost thawing. Under the high C emissions representative concentration pathway (RCP8.5),
90% loss of near-surface permafrost is projected to occur by 2300, with the majority of loss
occurring by 2100 (McGuire et al., 2018). Increasing temperatures and widespread thaw exposes
permafrost C to heterotrophic decomposition, potentially leading to enhanced emissions of
greenhouse gases to the atmosphere in the form of carbon dioxide ($CO_2$; Schuur et al., 2021) and
methane ($CH_4$; Turetsky et al., 2020). Alternatively, previously frozen soil organic carbon may
be mobilized into the aquatic network as dissolved organic carbon (DOC), the quantity and
quality of which will likely depend on local and regional hydrology, and landscape
characteristics (Tank et al., 2012; Vonk et al., 2015). At high latitudes (>50°N), lakes and rivers
of various sizes cover 5.6% and 0.47% of the total area, respectively (Olefeldt et al., 2021), and
the landscape C balance at these high latitudes is highly dependent on aquatic C processing
(Vonk & Gustafsson, 2013). The increased leaching of recently thawed DOC from permafrost
soils will not only increase the currently estimated 25 – 36 Tg DOC year$^{-1}$ exported into the
freshwater system, and subsequently  into the Arctic Ocean (Holmes et al., 2012; Raymond et al.,
2007), but will also likely lead to enhanced greenhouse gas emissions from freshwater
ecosystems (Dean et al., 2020). However, uncertainty remains as to which terrestrial ecosystems
are likely to contribute the highest concentrations of laterally transported permafrost DOC and of
this, which is expected to contribute the DOC most vulnerable to mineralization.
The contribution of mineralized permafrost C to atmospheric $CO_2$ and $CH_4$ balances, known
as the permafrost C feedback (Schaefer et al., 2014), remains poorly constrained due to
uncertainty of the magnitude and location of permafrost C emissions (Miner et al., 2022). The
lateral transport of DOC represents a source of terrestrial C that can potentially play an important
role in both terrestrial and aquatic biogeochemical cycles and is thus an important fraction of the
permafrost C feedback. Warming of near surface permafrost causes widespread thawing (Camill,



2005; Jorgenson et al., 2006), which can lead to drastic changes in hydrology, vegetation, and
soil carbon dynamics (Liljedahl et al., 2016; Pries et al., 2012; Varner et al., 2022). When
permafrost is present, the lateral transport of DOC is restricted to flow paths within the unfrozen,
organic rich active layer (Woo, 1986). Deeping of the seasonally thawed active layer due to top-
down permafrost thaw can lead to longer flow paths for DOC, allowing for enhanced
decomposition or adsorption to mineral particles, resulting in reduced DOC export (Kicklighter
et al., 2013; Striegl et al., 2005). Alternatively, thermokarst formation can affect the entire soil
profile, leading to surface inundation, and shifting ecological conditions and vegetation
communities associated with greater DOC production (Turetsky et al., 2007). This can cause
greater hydrological connectivity, resulting in increased runoff in permafrost peatlands (Connon
et al., 2014) or increased connectivity to regional hydrology through thermo-erosion gullies or
thaw slumps (Kokelj & Jorgenson, 2013) in tundra ecosystems. Permafrost landscape dynamics,
including the mode of permafrost thaw and ecological conditions present following thaw, will
play a key role in the biogeochemical and ecohydrological processes that constrain DOC
mobilization, i.e., export and mineralization upon export. The freshwater DOC pool represents a
mix of C derived from a variety of ecosystem types and sources, and the ecological conditions of
each source will have a significant impact on the quantity and quality of this mobilized DOC.
Determining the relative contribution and impact on mineralization of these DOC sources
represents a potentially important step in reducing uncertainty in the permafrost climate
feedback.
Here, we conduct a systematic review and compiled 111 studies published between 2000 –
2022 on DOC concentrations in the top 3 m of terrestrial ecosystems found in the northern
circumpolar permafrost region. A quantitative assessment of studies pertaining to DOC
concentrations in permafrost soils can identify evidence-based recommendations for future topics
and areas of research to improve our understanding on terrestrial and aquatic biogeochemical
cycling in northern permafrost regions. Our database contains ancillary data describing the
geographical and ecological conditions associated with each DOC concentration, allowing us to
reveal patterns in DOC concentrations and lability measures for 562 sampling sites across
multiple ecosystem types and under varying disturbance regimes. This study represents the first
systematic review of DOC concentrations within terrestrial permafrost ecosystems found in the



circumpolar north. As such, it provides unique and valuable insights into identifying ecoregions,
or landscape characteristics, associated with the highest DOC concentrations, and thus regions
with the greatest potential for DOC mobilization. Mobilization rates represent DOC loss and
include specific discharge of DOC (g DOC $m^{-2}$), export rate of DOC per day (g C $m^{-2}$ $day^{-1}$) and
per year (g C $m^{-2}$ $year^{-1}$), and biodegradable DOC (BDOC; %). We hypothesized that (i) the
highest DOC concentrations would be found in organic rich wetland ecosystems, (ii) disturbance
would lead to increased export and biodegradability of DOC, and (iii) the most biodegradable
DOC would be found in Yedoma and tundra ecosystems.

## 2. Methods

This systematic review used a methodological framework proposed by Arksey &
O'Malley (2005) and follows five steps: 1) develop research questions and a search query; 2)
identify relevant studies; 3) study selection; 4) data extraction; and 5) data analysis, summary,
and reporting. The literature search was guided by four research questions: 1) what are the
concentrations of DOC found in terrestrial ecosystems across the northern circumpolar
permafrost region?; 2) what are the rates of export and/or degradation (mobilization) of DOC
within these ecosystems?; 3) What are the major controls on DOC concentrations and rates of
mobilization?; and 4) how are concentrations and mobilization rates impacted by thermokarst
formation?

### 2.1 Literature Search

Based on *a priori* tests, we used the following search query string to find papers using
information found in their title, abstract, and keywords: ("dissolved organic carbon") AND
(permafrost OR thermokarst OR "thaw slump") AND (soil OR peat) AND (export OR degrad*
OR decomposition OR mineralization). We used Web of Science, Science Direct, Scopus,
PubMed, and Google Scholar to generate a database of tier 1, peer-reviewed articles published
between 2000 – 2022. The search function on Science Direct does not support the use of
wildcards such as "*", so "degrad*" was changed to "degradation". We removed duplicate
references found across multiple databases using Mendeley© referencing software (v1.17.1,
Mendeley Ltd. 2016). Once this initial database was complied, we used the same search query
string as above to search for additional articles on the first 15 pages of Google Scholar. This



resulted in the addition of a further 150 articles to be included in our systematic screening
process.

### 2.2 Systematic Screening of Peer-Reviewed Publications

The selection of relevant studies was comprised of inclusion criteria and relevance
screening in three steps. In the first step we placed limits on initial study searches in the
electronic databases mentioned above. Studies were included in the review if they were primary
research, published in English, and published between 2000 – 2022 (Table 1). Only quantitative
studies conducted in terrestrial ecosystems within the northern circumpolar permafrost region, as
defined by Brown et al., (1997), and reporting DOC concentration and mobilization rates were
included. Studies not meeting these criteria were eliminated and the remaining studies proceeded
to the second screening step.

Table 1. Summary of criteria used to identify suitable studies in the preliminary screening stage

|  | **Inclusion criteria** | **Exclusion criteria** |
|---|---|---|
| **Timeline** | Study published between 2000 – 2022 | Study published prior to 2000 |
| **Study type** | Primary research article published in peer-reviewed journal using quantitative methods | Thesis/dissertations and secondary research studies (reviews, commentaries, editorials) |
| **Language** | Published in English | Studies published in other languages |
| **Region** | Conducted within the northern circumpolar permafrost region | Conducted outside of the northern circumpolar permafrost region |
| **Outcome** | Studies on DOC concentration, export or degradation in permafrost environments | Studies not on DOC concentration, export or degradation in permafrost environments |


In the second step, the primary relevance of articles was screened, based on article titles,
abstracts, and keywords, and the eligibility criteria provided in Table 2. Studies deemed
irrelevant were eliminated and the remaining studies proceeded to the third and final screening



step, or secondary screening stage, which was based on was based on more specific eligibility
criteria (Table 2) applied to the full text.

Table 2. Primary and secondary relevance screening tools. Primary screening tool used in the article title, abstract, and keyword screening stage. Secondary screening tool used in full-text screening stage

| Screening stage | Screening questions | Response details |
|---|---|---|
| **Primary** | Does the study involve quantitative data collected from a permafrost environment? | Yes – reports on quantitative data collected from a permafrost environment<br><br>No – does not report on the above |
| Primary and Secondary | Is the study region within the northern circumpolar permafrost region? | Yes – reports on quantitative data (including field observations and lab data) collected from the circumpolar permafrost environment.<br><br>No – study region is not in the northern circumpolar permafrost regions; other examples could be mountainous permafrost or Tibetan plateau |
| **Primary** and Secondary | Is the article in English and NOT a review, book chapter, commentary, correspondence, letter, editorial, case report, or reflection? | Yes – study is in English and is a primary research article that includes quantitative studies (field and lab based), including model-based research as it relies on observational data.*<br><br>No – study is not in English and/or is a review, book, editorial, working paper, commentary, conference proceeding, supplementary text, or qualitative study which does not address outcomes relevant to this review |
| **Primary and Secondary** | Does the study involve the concentration, export or degradation of terrestrially derived DOC? | Yes – reports on terrestrial DOC concentration, export, or degradation, including concentrations and characterization<br><br>No – does not report on terrestrial DOC concentration, export, or degradation |
| **Secondary** | Is the article in English, longer than 500 words, and published between 2000 - 2022? | Yes – study is published between 2000 – 2022<br><br>No – study is published prior to 2000 |

*For model-based studies, the original field/lab data used to parametrise or develop the model
was used. If this data was taken from previously published work, then those studies were used
and the model-based study removed.



*2.3 Database compilation*


A database with reported DOC concentrations and mobilization rates i.e., rates of either
DOC export or degradation, was compiled using data from all studies that were deemed relevant
following the study selection phase. The database was compiled to compare DOC concentrations
and mobilization rates between different sites. We define a site as an area where either soil,
water, or ice samples were taken from that has similar vegetation composition, water table
position, permafrost regime, and was either disturbed or pristine. Site descriptions were derived
from the text of each study. Where possible, individual daily measurements of DOC
concentrations and mobilization rates were taken. When replicates of the same daily
measurement were provided, we used the mean of those replicates, which was relevant for 10
studies within the database, representing 72 DOC concentrations. All data was extracted from
data tables, text, supplementary material, or extracted from data figures using WebPlotDigitizer
(https://automeris.io/WebPlotDigitizer).
All studies reported measuring DOC concentrations collected from either open-water, pore
water, ice, or soil using a median filter pore size of 0.45 μm with first and third quartiles pore
size of 0.45 and 0.7 μm. Measurements from all 12 months of the year were included in the
database with the majority occurring during the growing season (May – August), a small portion
during the non-growing season, and the remaining sampling times were either not reported or are
averages over multiple sampling occasions. We included data from studies that were both field
and lab based. However, any data where a treatment was applied was excluded, except for
temperature treatments during incubation experiments when assessing the biodegradability of
DOC. When lab-based studies included an incubation, only Day 0 DOC concentrations were
used when comparing DOC concentrations across studies. We chose to remove any DOC
concentrations from samples taken below 3 m depth, which represented 3% of all DOC
measurements. These measurements were removed for better comparability with the current best
estimation of soil organic carbon stocks within the northern circumpolar permafrost zone
(Hugelius et al., 2014). We also removed any DOC concentrations greater than 500 mg L$^{-1}$,
which represented 2% of all DOC concentrations. Samples that were above 500 mg L$^{-1}$ and were
sampled below 3 m represented 1% of all DOC concentrations.



Site averaged daily DOC concentrations (mg L$^{-1}$) and mobilization rates were estimated from
the average concentration and mobilization rates measured within a single day or sampling
occasion. Repeated measurements at a site, either over the growing season or multiyear
measurements, were treated as an individual estimate of DOC concentrations and mobilization
rates. Other continuous variables that were similarly estimated include soil moisture, water table
position, organic layer depth, active layer depth, bulk density of soil, soil carbon content (%),
soil nitrogen content (%), carbon:nitrogen, pH, electrical conductivity (μS cm$^{-1}$), specific UV
absorbance at 254 nm (SUVA; L mg C$^{-1}$ m$^{-1}$), total dissolved nitrogen (mg L$^{-1}$), nitrate (mg L$^{-1}$),
ammonium (mg L$^{-1}$), chloride (mg L$^{-1}$), calcium (mg L$^{-1}$), and magnesium (mg L$^{-1}$). Mean annual
temperatures and precipitation, sampling depth, filter size, the number of days over which
sampling took place, how many years following disturbance measurements were taken were also
recorded. Several continuous variables other than those mentioned above were also recorded in
the database, but not used for analysis if they represented < 20% of the database. We chose 20%
as the cut-off point for use in comparison of the relationship between DOC concentrations and
mobilization with other site continuous variables.
Categorical variables included in the database were site location within the permafrost zone
(continuous, discontinuous, sporadic; Brown et al., 1997) and ecoregion (arctic tundra, sub-arctic
tundra, sub-arctic boreal, and continental boreal; (Olson et al., 2001). We included site surface
permafrost conditions (present or absent), the thermal horizon layer sampled (active layer,
permafrost, permafrost free, water, and thaw stream), and if present what type of disturbance
occurred at the site (fire, active layer thickening, thermokarst terrestrial, or thermokarst aquatic).
We also included the soil class found at the site (Histel, Histosol, Orthel, and Turbel; USDA,
1999) and whether the DOC was from the organic or mineral soil.  To assess the influence of
sampling approach and method of analysis, we included method of DOC extraction
(centrifugation of soil sample, leaching of soil, dialysis, grab sample, ice core extraction,
potassium sulphate extraction, lysimeter, piezometer, pump, rhizons) and DOC measurement
method (combustion, persulphate, photometric, or solid-phase extraction).
Sites were classified according to ecosystem type, and these included coastal tundra, forest,
peatland, permafrost bog, permafrost wetland, retrogressive thaw slump, upland tundra, and
Yedoma. Ecosystem classification is based on the general site description in the article, the
provided ecosystem classification within the article, and site data including vegetation
composition, permafrost conditions, and ecoregion. Yedoma sites include pristine forest, upland
tundra, and coastal tundra, as well as retrogressive thaw slumps and other thermokarst features
found within the Yedoma permafrost domain (Strauss et al., 2021). The ecosystem classification
retrogressive thaw slump only includes these thermokarst features found outside the Yedoma
permafrost domain. Each ecosystem type was further classified based on the type of permafrost
thaw or thermokarst formation that occurred there. These thaw or thermokarst types included
thermokarst bog, thermokarst wetland, active layer thickening, retrogressive thaw slump,
exposure, thermo-erosion gully, and active layer detachment.
*2.4 Database analysis*
All statistical analyses were carried out in R (Version 3.4.4, R Core Team, 2015). We used
Kruskal-Wallis analysis to test for differences in median DOC concentrations among various
categorical variables such as permafrost zones, ecoregions, soil class, thermal horizon, and
ecosystems. Post-hoc comparisons of median DOC concentrations among these categories were
performed using pairwise Wilcox test. We used ANOVAs and Bonferroni post-hoc tests on
linear mixed effects models, that include ecosystem type as a random factor, to evaluate
significant differences in DOC concentrations between methods of DOC extraction and
measurement. For regression analysis, data was transformed using a Box Cox transformation and
the optimal λ using the *MASS* package (Ripley et al., 2019). We used analysis of covariance
(ANCOVA) to test for differences in DOC concentrations in different thermal horizons (i.e.,
active layer and intact permafrost lens) between ecosystem types, while controlling for the month
in which sampling occurred. Permafrost lens DOC concentrations are determined from soil and
pore water within the permafrost layer and extracted via frozen cores, whereas active layer
samples are taken from soil cores or porewater that are unfrozen at the time of sampling. We
used partial least squares regression (PLS) to assess the performance of continuous and
categorical variables in predicting DOC concentrations. Predictor variables were categorized
based on their Variable Importance in Projections (VIP) method in the *plsVarSel* package
(Mehmood et al., 2012), whereby variables with a score > 1 are deemed to be significant. PLS
was performed using the *pls* package (Mevik & Wehrens, 2007) and we chose to use PLS as it is
tolerant of co-correlation of predictor variable, deviations from normality, and missing values, all
of which were found within the database. In the PLS ecosystem classes were subdivided into
pristine or disturbed (i.e., impacted by permafrost thaw). Pristine sites were further subdivided
by the thermal horizon in which the DOC concentrations were measured (active layer and
permafrost lens). To evaluate the change in ecosystem DOC concentrations following
thermokarst formation, based on all studies from the systematic review, we calculated the
response ratio using the *SingleCaseES* package (Pustejovsky et al., 2021). We define thermokarst
as the process by which ice-rich permafrost deposits undergo complete thaw, resulting in surface
subsidence and the formation of a new, thermokarst feature that is ecological different regarding
water table position, redox conditions, and vegetation type, from the preceding pristine
ecosystem. Very few studies in our database report DOC concentrations for both pristine and
thermokarst affected ecosystem (< 20 %). To include as much data as possible we chose an
effect size metric that is unlikely to be influenced by studies with large sample number and
variance. The response ratio is;
$Pristine\ to\ Thermokarst\ Effect\ Response\ ratio = \ln(\frac{X_P}{X_T})$      Eqn. 1
where $X_P$ = mean DOC concertation of pristine ecosystems and $X_T$ = mean DOC concertation of
thermokarst effected ecosystems (Lajeunesse, 2011). This represents the log proportional
difference in mean DOC concentrations between thermokarst and pristine ecosystems, where a
positive response ratio indicates a decrease in DOC concentrations following thermokarst. The
distribution of the data was inspected visually and with the Shapiro–Wilk test. We tested
homogeneity of variances using the *car* package and Levene's test (Fox and Weisberg, 2011).
We report uncertainty using the interquartile range (lower, median, and upper quartiles), except
for response ratios which we report as ± 95% confidence intervals. We here define the statistical
significance level at 5%.
**3. Results**
3.1 *Database generation*
Our initial search using Web of Knowledge, Science Direct, Scopus, PubMed, and
Google Scholar returned a total of 577 unique papers published between 2000 – 2022 that assess
the concentrations and rates of mobilization of DOC in terrestrial ecosystems within the northern

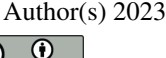



circumpolar permafrost region. Of these initial 577 studies, 111 remained after the systematic
screening process (Table 1 & 2). From these 111 studies we generated our database. The final
database of 111 studies contained a total of 3,340 DOC concentrations (mg L$^{-1}$), with 2,845 DOC
concentrations between 0 – 500 mg L$^{-1}$, found within the top 3 m of permafrost soils from field
and lab-based studies (using only Day 0 lab-based DOC concentrations). These concentrations
were taken from 562 different sampling locations, representing 8 different ecosystem types
(Figure 1) across the northern circumpolar permafrost region. All studies except, for one
(Olefeldt et al., 2012), reported DOC concentrations.

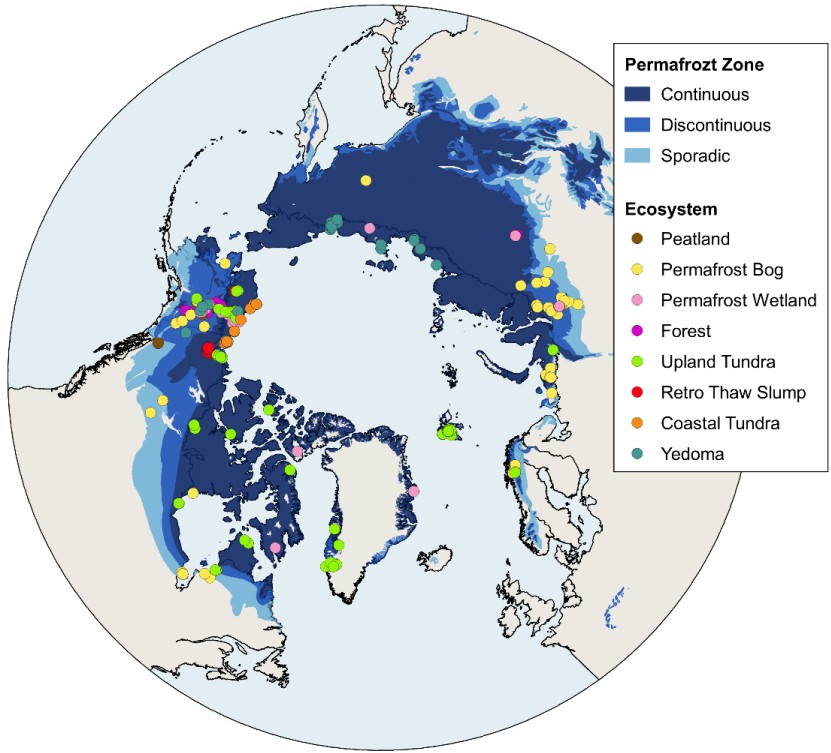


Figure 1. Map of sampling locations where DOC measurements (n=562) from the top 3 m for
each ecosystem type. In many cases, the same sampling location was used in multiple studies
leading to some overlap, therefore the number of sampling sites included in the data set (562)
are not all clearly identifiable from this map. Retro Thaw Slump = Retrogressive Thaw Slump.
Blue shading represents permafrost zonation (Brown et al., 1997).



The final database contained a considerably lower number of DOC mobilization
measurements. The database includes 16 measurements of specific discharge of DOC (g DOC m$^{-}$
$^{2}$) from 3 studies, 9 export rate of DOC per day (g C m$^{-2}$ day$^{-1}$) and per year (g C m$^{-2}$ year$^{-1}$)
measurements were each found in 2 studies. The number of specific discharge, export of DOC
per day, and export of DOC per year measurements combined were <1% of the number of DOC
concentration measurements. As such they were not considered for analysis of DOC
mobilization. A total of 146 BDOC (%) measurements, 4% of the total number of DOC
concentration measurements, were found in 14 studies. These measurements of BDOC were
from Yedoma (30:5, number of measurements:studies), Upland Tundra (55:5), Forest (18:3),
Permafrost Wetland (12:2), and Permafrost Bog (31:5) ecosystems. Given the low number of
other forms of DOC mobilization and relatively comparable spread of BDOC measurements
across ecosystem types, we chose to include BDOC measurements in our analysis despite a low
total number of measurements compared to DOC concentrations, and we consider this lower
sample size during our interpretation of results.
Filter size used in studies ranged from 0.15 – 0.7 μm. The majority of studies used a filter
size of 0.45 μm (1,375 out of 2,845 DOC measurements), 0.7 μm was the second most common
filter size (n = 489), followed by 0.22 μm (n = 332) and 0.6 μm (n = 143). Two studies used a
filter size of 0.15 μm totalling 18 DOC measurements and remaining studies (n = 12) did not
provide a filter size. DOC concentrations were found to differ between different filter sizes
(ANOVA: $F_{(4, 2339)} = 22.9$, $p < 0.001$). DOC concentrations from samples filtered using 0.7 μm
were lower (median = 11 mg L$^{-1}$) than 0.45 μm and 0.22 μm filtered samples (median = 53 and
42 mg L$^{-1}$, respectively). We consider the effects of filter size to be minor. DOC concentrations
were found to be significantly different between samples subject to the 11 different extraction
methods used (ANOVA: $F_{(10, 2515)} = 21.8$, $p < 0.001$), and between water based and soil (solid)
based extraction methods (ANOVA: $F_{(1, 2524)} = 182.1$, $p < 0.001$). Median DOC concentrations of
the 4 methods of extraction directly from soils (leaching from soil under field moisture
conditions, leached from dried soils, centrifuged soils, and extracted using $K_2SO_4$) were 57 mg
L$^{-1}$, with upper and lower quartiles of 20 and 120 mg L$^{-1}$, respectively. The 7 water-based
extraction methods had a median DOC concentration of 24 mg L$^{-1}$, with upper and lower
quartiles of 8 and 59 mg L$^{-1}$, respectively. DOC concentrations differed (ANOVA: $F_{(3, 2515)} =$





36.2, $p < 0.001$) between samples subject to different dissolved organic carbon measurement
methods, with median values of 37 and 48 mg L$^{-1}$ for the combustion, and photometric methods,
respectively. Median values measured using the persulphate were higher at 97 mg L$^{-1}$.
Combustion was the most common method, accounting for 2,170 DOC concentrations, followed
by persulphate (n = 230) and photometric (n = 31). In this study we did not focus on
systematically testing the effect of filter sizes, extraction methods, or DOC measurement
methods. Our goal was to assess the concentration and mobilization of DOC in terrestrial
permafrost ecosystems and the assessment of methods is outside the scope of our study. Rather,
we compare DOC concentrations collected from samples using a variety of these methods and
suggest that future studies use this information to decide on methods to be consistent with
compiled measurements, thus far.
*3.2 DOC concentrations and study regions*
Upon inspection of DOC concentrations in the database, we determined that the data was
non-normally distributed. The DOC concentrations were skewed toward the lower end of our 0 –
500 mg L$^{-1}$ range; thus, we report median, upper, and lower quartiles below. Across all studies,
within the top 3 m, the median DOC concentration was 41 mg L$^{-1}$, with upper and lower
quartiles of 12 and 86 mg L$^{-1}$, respectively. DOC concentrations were found to differ among the
three permafrost zones (chi-square = 32, df = 2, $p < 0.001$; Figure 2a). The highest median DOC
concentrations were found within the sporadic permafrost zone (n = 83; 62 mg L$^{-1}$), lower
quartile (LQ) and upper quartile (UQ) of 23 and 167 mg L$^{-1}$, respectively. The lowest median of
33 mg L$^{-1}$ (LQ and UQ of 11 and 88 mg L$^{-1}$, respectively) was found in the continuous
permafrost zone (n = 1,648), with the greatest density of samples having lower DOC
concentrations than observed in the violin plots of both he discontinuous and sporadic (Figure
2a). This change in DOC concertation's along the latitudinal gradient of the permafrost zonation
was also seen in the latitudinal gradient associated with ecoregion (chi-square = 78, df = 3, $p <$
*0.001*; Figure 2b). The highest DOC concentrations were found in the continental boreal (n =
389; 56 mg L$^{-1}$; LQ = 24 mg L$^{-1}$; UQ = 80 mg L$^{-1}$) and Sub-Arctic Boreal (n = 442; 58 mg L$^{-1}$;
LQ = 20 mg L$^{-1}$; UQ = 107 mg L$^{-1}$) ecoregions, and lowest in the Arctic Tundra (n = 1,209; 25
mg L$^{-1}$; LQ = 9 mg L$^{-1}$; UQ = 84 mg L$^{-1}$) and Sub-Arctic Tundra (n = 493; 43 mg L$^{-1}$; LQ = 15
mg L$^{-1}$; UQ = 76 mg L$^{-1}$) ecoregions. Inspection of the distribution of DOC cocnetrations across



the ecoregions highlights that the Arctic Tundra ecoregion had the highest density of samples at
the lowest DOC concentration (Figure 2b).

These latitudinal differences are also reflected in the observed differences (chi-square =

20, df = 3, *p < 0.001*) in DOC concentrations found within different soil classes. The highest
DOC concentrations are found within organic rich Histosol (n = 37; 61 mg L$^{-1}$; LQ = 32 mg L$^{-1}$;
UQ = 71 mg L$^{-1}$) and Histel soils (n = 935; 53 mg L$^{-1}$; LQ = 16 mg L$^{-1}$; UQ = 88 mg L$^{-1}$; Figure
2c), with the distribution of the data from these soils types having a higher density at greater
DOC concentrations (Figure 2c). Histel and Histosol soils are the main type of permafrost soil
found within the sporadic and discontinuous permafrost zone and both boreal ecoregions
(Hugelius et al., 2014). Mineral rich Orthels (n = 741; 38 mg L$^{-1}$; LQ = 11 mg L$^{-1}$; UQ = 102 mg
L$^{-1}$) and Turbels (n = 820; 31 mg L$^{-1}$; LQ = 12 mg L$^{-1}$; UQ = 74 mg L$^{-1}$), mineral permafrost
soils that have experienced cryoturbation, had the lowest DOC concentrations. The median DOC
concentrations found within the top 3 m of these soil classes represent <1% of the soil organic
carbon stock found in the top 3 m of each soil class (Hugelius et al., 2014). DOC concentrations
also differed within the thermal horizon of these different soil classes (chi-square = 91, df = 3, *p*
*< 0.001*; Figure 2d). The highest DOC concentrations were found in permafrost free sites (n =
202; 57 mg L$^{-1}$; LQ = 47 mg L$^{-1}$; UQ = 69 mg L$^{-1}$), which were largely Histosol soils (19%) or
Histel soils (74%) that have experienced thermokarst formation. In areas where permafrost was
present, DOC concentrations were highest in the active layer (n = 1,400; 45 mg L$^{-1}$; LQ = 14 mg
L$^{-1}$; UQ = 88 mg L$^{-1}$) and the permafrost lens (n = 729; 30 mg L$^{-1}$; LQ = 10 mg L$^{-1}$; UQ = 123
mg L$^{-1}$).





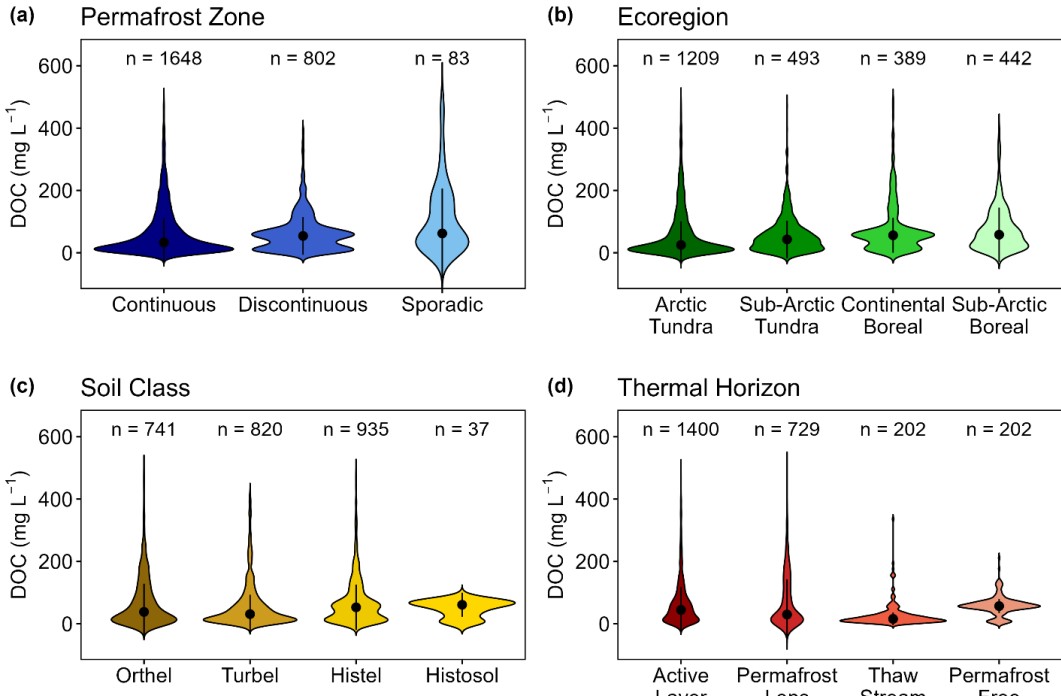


Figure 2. Violin plots of DOC concentrations (mg L$^{-1}$) found in the top 3 m across (a) permafrost zones, (b) ecoregions, (c) soil classes, and (d) thermal horizons. (a) Dark to light blue shading represents the permafrost zones Continuous, Discontinuous, and Sporadic, according to Brown et al., (1997). (b) Dark to light green shading represents the ecoregions Arctic Tundra, Sub-Arctic Tundra, Continental Boreal, and Sub-Arctic Boreal, according to Olson et al., (2001). (c) Dark to light yellow shading represents the soil classes Histosol, Histel, Orthel, and Turbel, according to the USDA Soil Taxonomy (USDA, 1999). Histosols are organic rich, non-permafrost soil. Histels, Orthels, and Turbels are permafrost-affected soils (Gelisol order). Histels are organic rich, Orthels are non cryoturbated affected mineral soils, and Turbels are cryoturbated permafrost soils. (d) Dark to light red shading represents the thermal horizons Active Layer, Permafrost Lens, Thaw Stream, and Permafrost Free. Active layer represents the seasonally unfrozen soil layer above the permafrost layer. Permafrost Lens represents the permanently frozen (below 0°C) layer. Thaw Stream represents flowing surface waters following permafrost thaw. Permafrost Free represents areas that are not underlain by permafrost. Black dots on each violin plot represents the median. Black vertical lines represent the interquartile range with the upper and lower limits representing the 75$^{th}$ and 25$^{th}$ percentiles, respectively. Either side of the black vertical line represents a kernel density estimation. This shape shows the distribution of the data, with wider areas representing a higher probability that samples within the database will have that DOC concentrations. The number of samples (n) found in each sub-category is found above each corresponding violin plot.






### 3.3 Trends in DOC concentrations across ecosystems

Similar to other categorical variables (i.e. permafrost zone, ecoregion, soil class, and
thermal horizon data), DOC concentrations within each of the eight ecosystem types were found
to be non-normally distributed, with median values skewed toward the lower end of the 0 – 500
mg L$^{-1}$ range of concentrations (Figure A1). Permafrost bogs and permafrost wetlands were the
most represented in the database with regards to DOC concentrations, with a total of 685
concentrations from 38 studies and 679 concentrations from 22 studies, respectively. The
majority of permafrost bog measurements came from studies with field sites within Canada
(Figure 1), as was the case for upland tundra and retrogressive thaw slump DOC concentration
data. The majority of permafrost wetland sample locations were found in Russia, whereas the
majority of the 399 coastal tundra sampling locations were in the USA. The least represented
ecosystem classes included the  peatland ecosystem class, which is not strictly a permafrost
ecosystem as the other are, and the Yedoma ecosystem class (118 DOC concentrations from 9
studies). DOC concentrations differed significantly across the eight ecosystem types (chi-square
= 700, df = 7, $p < 0.001$; Figure 3). The highest DOC concentrations were found in coastal
tundra (66 mg L$^{-1}$; LQ = 24 mg L$^{-1}$; UQ = 140 mg L$^{-1}$) and permafrost bogs (63 mg L$^{-1}$; LQ = 36
mg L$^{-1}$; UQ = 111 mg L$^{-1}$) ecosystems. The lowest DOC concentrations were found in
permafrost wetlands (7 mg L$^{-1}$; LQ = 6 mg L$^{-1}$; UQ = 26 mg L$^{-1}$) and Yedoma ecosystems(9 mg
L$^{-1}$; LQ = 2 mg L$^{-1}$; UQ = 20 mg L$^{-1}$), both of which had only slightly lower median DOC
concentrations than retrogressive thaw slumps (15 mg L$^{-1}$; LQ = 7 mg L$^{-1}$; UQ = 26 mg L$^{-1}$).






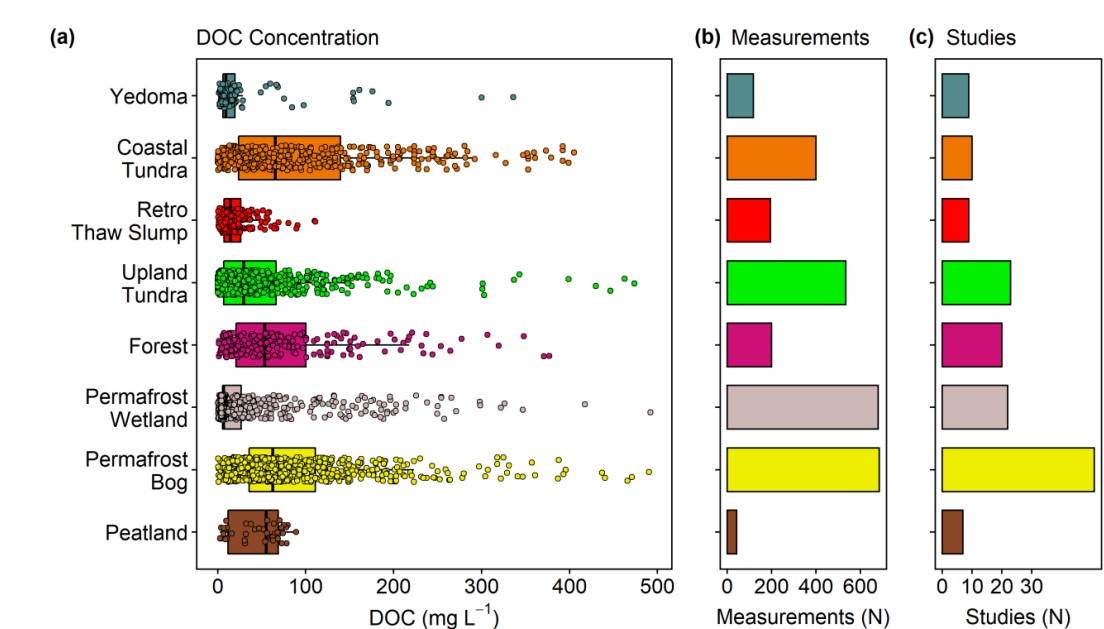


Figure 3. Boxplot and jitter plot of (a) DOC concentrations (mg L$^{-1}$), (b) the number of DOC measurements, and (c) number of studies including DOC measurements were taken from the top 3 m for each ecosystem type. Retro Thaw Slump = Retrogressive Thaw Slump. Boxes represents the interquartile range (25 – 75%), with median shown as black horizontal line. Whiskers extend to 1.5 times the interquartile range (distance between first and third quartile) in each direction. Jitter points represent the concentration of each individual DOC measurement, with random variation applied to each points location vertically in the plot, to avoid overplotting. Yedoma = dark teal. Coastal Tundra = orange. Retro Thaw Slump = red. Upland Tundra = green. Forest = purple. Permafrost Wetland = light pink. Permafrost bog = yellow. Peatland = brown.

When grouping all DOC concentrations by ecosystem types and differentiating between the active layer and permafrost lens thermal horizons, we found that DOC concentrations differed between the active layer and permafrost for all ecosystems (ANCOVA: $F_{(1, 1277)} = 49.8$, $p < 0.001$) except for permafrost bogs (chi-square = 0.37, df = 1, $p = 0.5$) and Yedoma (chi-square = 3.5, df = 1, $p = 0.06$) ecosystems (Figure 4). Within the permafrost lens thermal horizon, the highest DOC concentrations were found in coastal tundra (n = 103; 130 mg L$^{-1}$; LQ = 60 mg L$^{-1}$; UQ = 179 mg L$^{-1}$) and permafrost bogs (n = 248; 78 mg L$^{-1}$; LQ = 19 mg L$^{-1}$; UQ = 163 mg L$^{-1}$) sites, and lowest found in Yedoma sites (n = 91; 8 mg L$^{-1}$; LQ = 6 mg L$^{-1}$; UQ = 16





mg $L^{-1}$). The highest active layer DOC concentrations were in permafrost bogs (n = 276; 64 mg
$L^{-1}$; LQ = 41 mg $L^{-1}$; UQ = 102 mg $L^{-1}$) and forest (n = 185; 57 mg $L^{-1}$; LQ = 26 mg $L^{-1}$; UQ =
110 mg $L^{-1}$) sites, and lowest found in permafrost wetland sites (n = 274; 10 mg $L^{-1}$; LQ = 5 mg
$L^{-1}$; UQ = 47 mg $L^{-1}$).

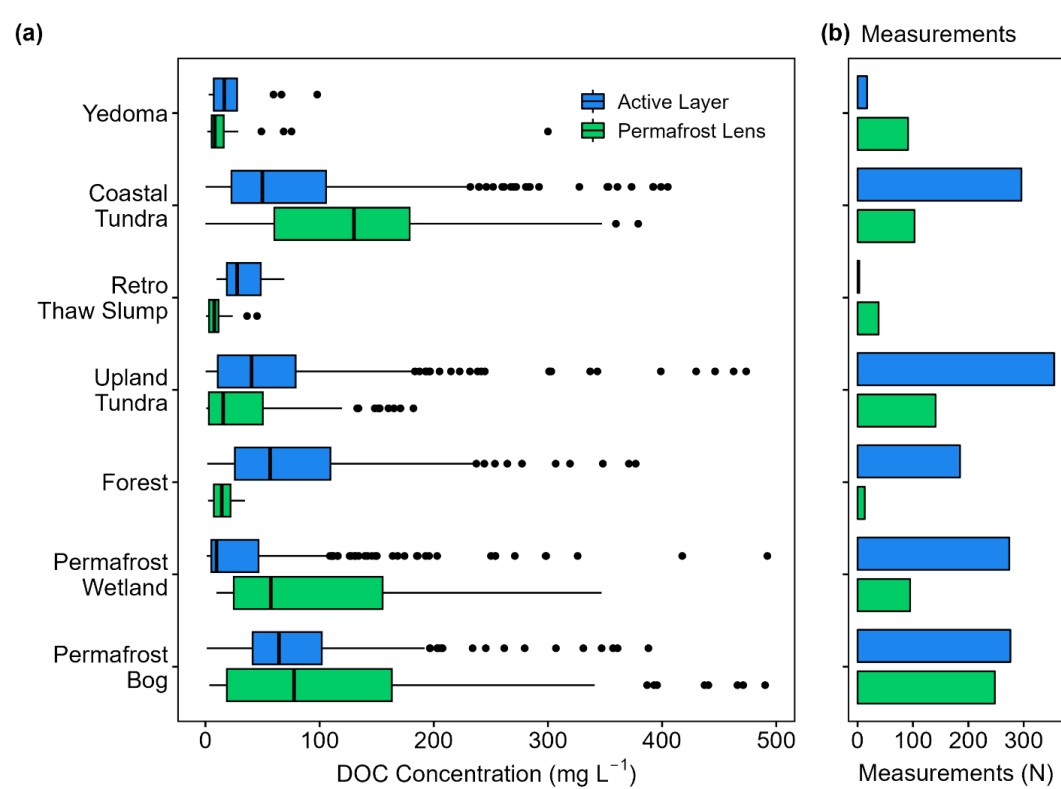


Figure 4 . Boxplot of (a) DOC concentrations (mg $L^{-1}$) and (b) (b) the number of DOC
measurements in the Active Layer and Permafrost Lens thermal horizons of each ecosystem
type. Only DOC concentrations from ecosystems with these thermal horizons present is used,
thus no peatland or permafrost-free sites are included. Retro Thaw Slump = Retrogressive
Thaw Slump. Boxes represents the interquartile range (25 – 75%), with median shown as black
horizontal line. Whiskers extend to 1.5 times the interquartile range (distance between first and
third quartile) in each direction. Blue boxplots represent DOC concentrations in the active layer.
Gren boxplots represent DOC concentrations in the permafrost lens.

*3.4 Drivers of DOC concentrations*




No continuous variables recorded in the dataset were available for all DOC concentration
database entries, with no sites containing data for all continuous variables. This limited our
ability to explore relationships between continuous environmental and ecological data and DOC
concentrations across the permafrost region. To address drivers of DOC concentrations across
the circumpolar permafrost region we used partial least squares regression (PLS) as it is tolerant
to missing values. Multiple PLS regressions were run using various combinations of continuous
and categorical data with similar model performance throughout. We chose the PLS to predict
DOC concentrations using environmental continuous variables and ecosystem type as this
contained the lowest background correlation. The most parsimonious PLS regression extracted 5
significant components, captured 79% variation of the predictor variables, and explained 37% of
the variance in DOC concentrations in the dataset. The majority of the variance in DOC (35%) is
explained along the first two axes of the model. The model was robust and not overfitted as
model predictability was moderate ($Q^2 = 0.35$) and background correlation was low (0.006).
The PLS plot (Figure 5a) shows the correlation between DOC concentrations and
selected environmental and ecological variables for the first two axes of the model. The two
variables with the greatest positive and negative effect on DOC concentrations were total
dissolved nitrogen content (mg L$^{-1}$) and C:N ratios, respectively (Figure 5b). The positive
relationship between DOC and total dissolved nitrogen soil carbon content (SoilC), and negative
relationship with the specific UV absorbance at 254 nm (SUVA), may be a result of ecosystem
properties. The aromatic content of organic matter is positively correlated with SUVA (Weishaar
et al., 2003), with high SUVA values being used as an indication of high aromatic content
(Hansen et al., 2016). Ratios of C:N have been shown to be a good proxy for decomposition
(Biester et al., 2014), where high C:N values indicate higher decomposition. The strong negative
relationship with C:N ratios indicates that DOC concentrations decrease with increased
decomposition. Other than higher soil carbon content (SoilC) in permafrost bogs, there was no
clear or obvious trends in SoilC, TDN, C:N ratios, and SUVA across ecosystem types (Figure
A3). The PLS demonstrates that ecosystem type strongly affects DOC concentrations, with DOC
positively related with the highest ecosystems where the highest DOC concentrations are
observed, permafrost bogs and coastal tundra, and negatively related to the lower DOC
ecosystems, permafrost wetland and retrogressive thaw slumps (Figure 5). This negative





relationship may be due to the higher latitudes these ecosystems are generally found at, which is
supported by the negative relationship with DOC and the climate indicators mean annual
temperature (MAAT) and mean annual precipitation (MAP). Additionally, it may be due to the
high number of thermokarst affected sites found within these ecosystem classes, particularly
retrogressive thaw slumps. There is a clear negative relationship between DOC concentrations
and disturbed permafrost wetlands, retrogressive thaw slumps, and permafrost bogs.

**(a)**

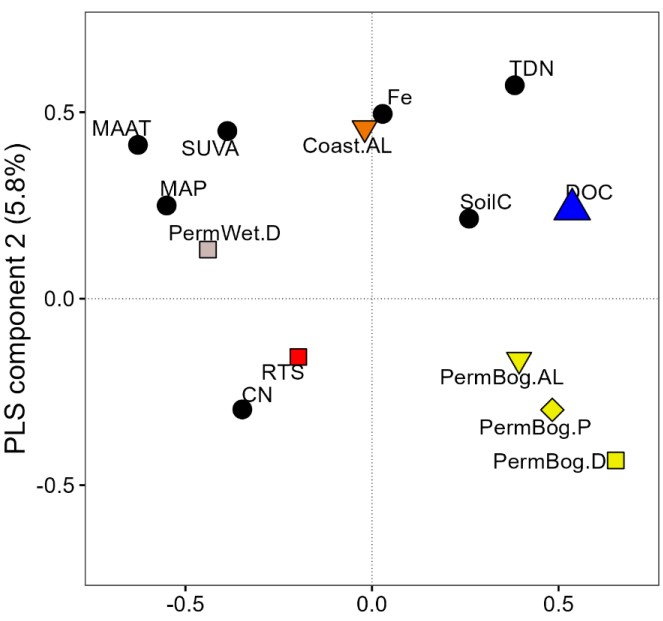

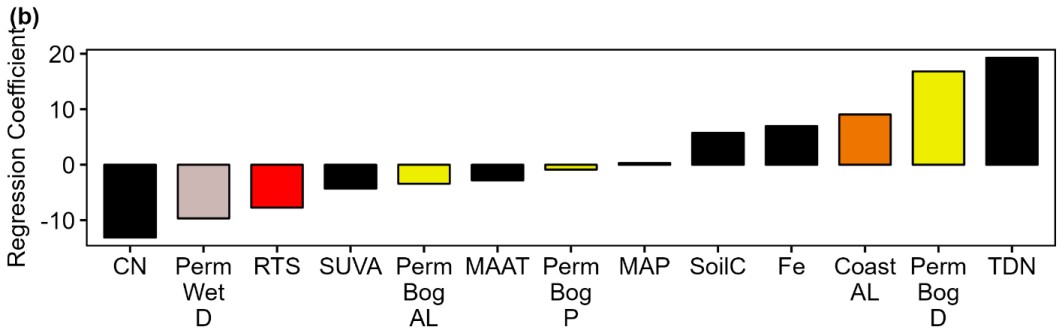






Figure 5. Partial least squares regression (PLS) (a) loadings plot explaining 37% of the variability observed in DOC concentrations. PLS component axis 1 explains 28.8% of this variability, whereas PLS component axis 2 explains 5.8%. The remaining axes explain the variability in DOC are not shown for clarity. (b) Bar plot of PLS regression coefficients showing the relative importance of each variable in predicting DOC concentrations. Regression coefficients on y-axis are normalized so their absolute sum is 100, with positive and negative values indicating the direction of the relationship. In the loadings plot squares depict ecosystem classes and the blue triangle represents DOC concentrations. Black circles in the (a) loadings plot and black bars in the (b) bar plot represent continuous environmental data that had at lest 20% coverage of DOC data,. All continuous data was log transformed, mean centered, and standardized. Continuous data variables are represented by the colour black. CN = carbon:nitrogen ratio. SUVA = the specific UV absorbance at 254 nm (L mg C$^{-1}$ m$^{-1}$). MAP = mean annal precipitation (mm). MAAT = mean annual temperature. SoilC = carbon content of soil (g C kg$^{-1}$). TDN = total dissolved nitrogen (mg L$^{-1}$). Fe = dissolved iron ((mg L$^{-1}$). PermWet.D = disturbed permafrost wetland ecosystem class and is light pink (as in Figure 3) to represent this ecosystem class. RTS = retrogressive thaw slump ecosystem class and is red (as in Figure 3) to represent this ecosystem class. Coast.AL = active layer of coastal tundra ecosystem class and is orange (as in Figure 3) to represent this ecosystem class. PermBog.AL = active layer of permafrost bog ecosystem class and is yellow (as in Figure 3) to represent this ecosystem class. PermBog.P = permafrost lens of permafrost bog ecosystem class and is yellow (as in Figure 3) to represent this ecosystem class. PermBog.D = disturbed permafrost bog ecosystem class and is yellow (as in Figure 3) to represent this ecosystem class.

### 3.5 Response and mobilization of DOC and BDOC to thermokarst formation

The highest DOC concentrations were found in pristine permafrost bog (75 mg L$^{-1}$; LQ = 37 mg L$^{-1}$; UQ = 149 mg L$^{-1}$, n = 442) and coastal tundra ecosystems (72 mg L$^{-1}$; LQ = 25 mg L$^{-1}$; UQ = 151 mg L$^{-1}$ n = 427; Figure 6a). No thermokarst affected coastal tundra ecosystems were recorded within the dataset. Whereas, in permafrost bogs DOC concentrations were found to differ across different thermokarst disturbances (ANOVA: $F_{(3, 720)} = 23.04$, $p < 0.001$), with the lowest found in thermokarst wetlands (10 mg L$^{-1}$; LQ = 9 mg L$^{-1}$; UQ = 30 mg L$^{-1}$, n = 16). DOC concentrations were also found to differ between thermokarst affected and pristine sites in upland tundra ecosystems (ANOVA: $F_{(3, 539)} = 5.91$, $p < 0.001$). The highest DOC concentrations in upland tundra ecosystems were found in sites that had experienced active layer thickening (53 mg L$^{-1}$; LQ = 41 mg L$^{-1}$; UQ = 80 mg L$^{-1}$, n = 142), whereas the lowest were found in sites that had experienced active layer detachment (4 mg L$^{-1}$; LQ = 3 mg L$^{-1}$; UQ = 5 mg L$^{-1}$, n = 6). Pristine sites had the highest DOC concentrations in both Yedoma (11 mg L$^{-1}$; LQ = 6 mg L$^{-1}$; UQ = 21 mg L$^{-1}$, n = 114) and forest (49 mg L$^{-1}$; LQ = 22 mg L$^{-1}$; UQ = 86 mg L$^{-1}$, n = 189) ecosystems. However, in permafrost wetland ecosystems pristine sites had the lowest DOC concentrations (7 mg L$^{-1}$; LQ = 6 mg L$^{-1}$; UQ = 57 mg L$^{-1}$, n = 766) with sites that were




affected by both thermokarst wetland formation (21 mg L$^{-1}$; LQ = 11 mg L$^{-1}$; UQ = 37 mg L$^{-1}$, n
= 17) and active layer thickening (41 mg L$^{-1}$; LQ = 34 mg L$^{-1}$; UQ = 47 mg L$^{-1}$, n = 12) having
higher DOC concentrations.

BDOC was found to differ between thermokarst disturbances within ecosystem types in

only Yedoma (ANOVA: $F_{(2, 27)} = 23.09$, $p < 0.001$) and permafrost wetland (ANOVA: $F_{(1, 10)} =$
15.87, $p < 0.001$) ecosystems. The highest BDOC was found in both of these ecosystem types
also, with 54% (n = 5) in pristine Yedoma sites and 49% (n = 8) in thermokarst wetland affected
permafrost wetland sites (Figure 6b), with the latter exhibiting the highest BDOC across all
permafrost affected sites followed by thaw slumps (18%, n = 11) in Yedoma ecosystems and
active layer thickening (40%, n = 1) in upland tundra sites. The lowest median BDOC of 4%
were seen in thermokarst bogs (n = 5) and active layer thickening (n = 3) affected sites, with
pristine sites experiencing BDOC of 9% (n = 15). However, not all ecosystem types in the
database had BDOC data for both pristine and disturbance sites. For example, only pristine sites
data was available for forests, whereas there was no pristine site data available for upland tundra
sites. No BDOC data was available for coastal tundra sites.

All ecosystem types that had BDOC data, reported BDOC observed following 40 – 90

incubation days, and this also corresponded to the highest BDOC values for each ecosystem type
(Figure A4). When comparing the greatest BDOC observed within this incubation length
window, we found that values varied across ecosystem type (ANOVA: $F_{(5, 131)} = 14.6$, $p <$
0.001). The highest loss rates were observed in Yedoma and permafrost wetland ecosystems,
whereas the lowest we observed in organic rich forest and permafrost bog ecosystems (Figure
A4). Forest (ANOVA: $F_{(1, 16)} = 2.31$, $p = 0.15$) and permafrost bog (ANOVA: $F_{(3, 24)} = 2.49$, $p =$
0.09) BDOC did not differ over incubation length, whereas Yedoma (ANOVA: $F_{(4, 25)} = 24.92$, $p$
$< 0.001$) and permafrost wetland (ANOVA: $F_{(1, 10)} = 15.87$, $p < 0.01$) did differ over time, with
their max occurring during this 40 – 90-day incubation length. This suggests that when incubated
for the same number of days, we would expect greater BDOC in Yedoma and permafrost
wetland ecosystems. Note, for this analysis BDOC values from all thermokarst and non-
thermokarst affected sites within an ecosystem type were included.



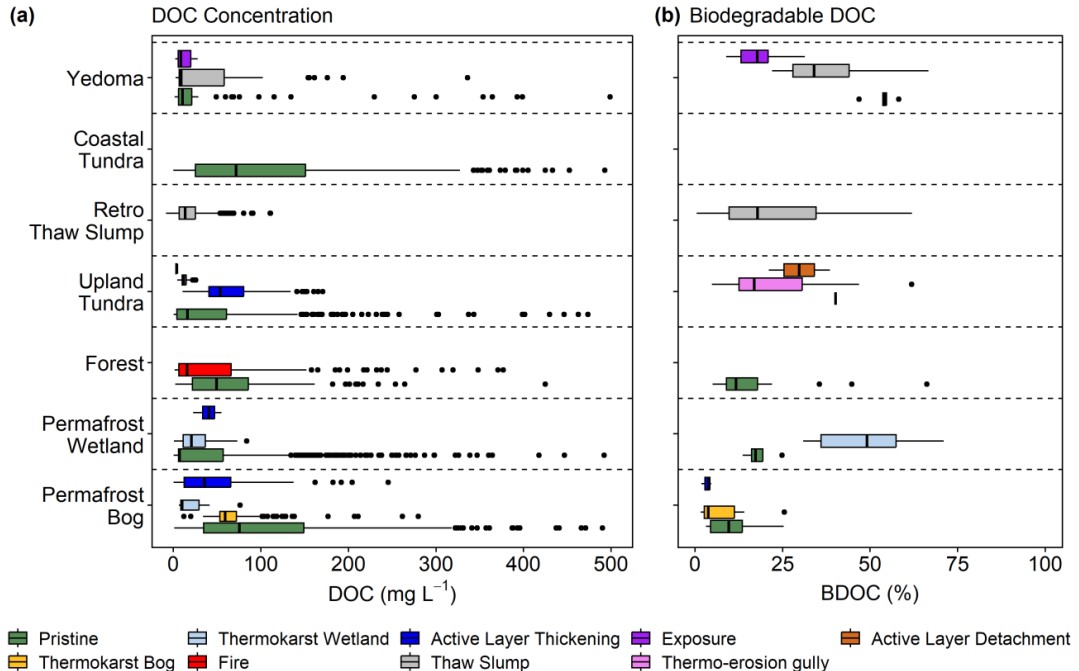


Figure 6. DOC concentrations (mg L$^{-1}$) and biodegradable DOC (BDOC; %) from the top 3 m
following disturbance including data from both field based and incubation studies. (a) DOC
concentrations from each ecosystem type following disturbance where data was available. (b)
Biodegradable DOC (BDOC) from each ecosystem type following disturbance where data was
available. BDOC loss was determined following 3 – 304 days of incubation. Data from different
incubation lengths was combined due to low sample size. Retro Thaw Slump = Retrogressive
Thaw Slump. Boxes represents the interquartile range (25 – 75%), with median shown as black
horizontal line. Whiskers extend to 1.5 times the interquartile range (distance between first and
third quartile) in each direction, with outlier data plotted individually as black dots. Note colours
associated with boxplots in this figure are only relevant for this figure.

Response ratios comparing the change in DOC concentrations between pristine and

thermokarst affected sites were calculated from our dataset from 108 studies using Eq. 1 (Figure
7). Only 17 studies provided data for both pristine and thermokarst affected ecosystems, with 87
papers providing DOC concentrations from pristine and 34 from thermokarst affected sites.
When considering all ecosystems together we found that response ratios were negative,
suggesting that DOC concentrations were higher in thermokarst affected sites compared to
pristine sites (Figure 7). These negative response ratios were most evident in permafrost bogs,
where they found throughout the entire column and individual thermal horizons. The greatest



increase in DOC concentrations following thermokarst was seen when comparing DOC
concentrations in the permafrost lens of permafrost bogs, and to a lesser extent permafrost
wetlands (Figure 7). Only in Yedoma ecosystems did we see positive response ratios throughout
the entire profile, suggesting a decrease in DOC concentrations following thermokarst formation
in Yedoma sites. This was also seen for DOC concentrations within the permafrost lens of
upland tundra sites, which include DOC concentrations from retrogressive thaw slumps and
thermo-erosion gullies in their thermokarst affected sites. The large confidence intervals for
some response ratios suggests high variability in the response of DOC concentrations to
thermokarst formation.



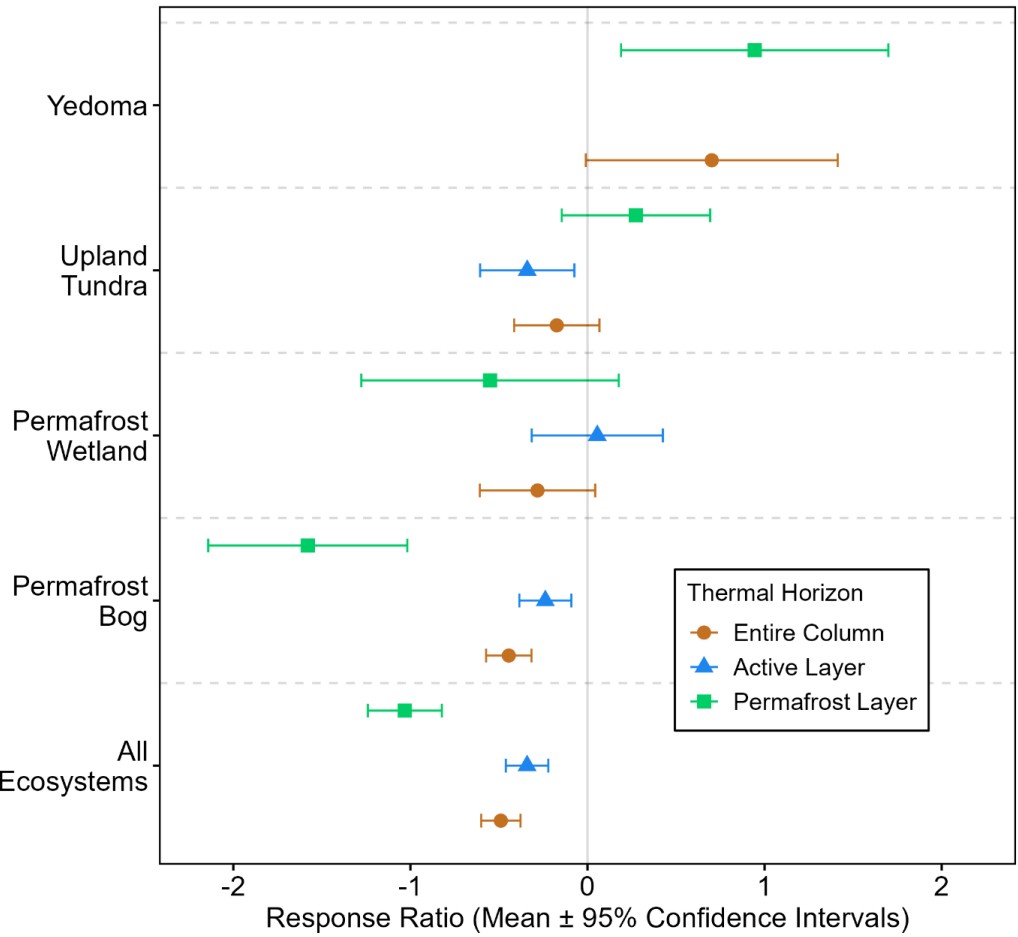


Figure 7. Response ratios of DOC concentrations from the top 3 m following thermokarst

formation (calculated using Eq. 1). Response ratio means allow for relative comparison of

changes in DOC following thermokarst formation between different ecosystem types. Negative

values indicate lower DOC concentrations found in pristine ecosystems, whereas positive value

indicates a decrease in DOC concentrations following thermokarst. Studies reporting DOC

concentrations from Exposures, Retrogressive Thaw Slumps, and Thermo-Erosion Gullies from

sites within the continuous permafrost zone were combined into the Upland Tundra ecosystem

category. This did not include DOC concentrations from studies within the Yedoma permafrost

domain (Strauss et al., 2021). Blue line represent DOC concentrations in the active layer, as per

Figure 4. Green lines represent DOC concentrations in the permafrost lens, as per Figure 4.





Brown lines represent DOC concentrations from the entire column (i.e., both active layer and
permafrost lens).

## 4. Discussion

In this systematic review, we evaluated patterns of DOC concentrations in the top 3 m of
terrestrial ecosystems across the northern circumpolar permafrost region based on results from
111 studies and 2,845 DOC measurements. We focused on comparing concentrations of DOC in
soils across various geographical regions, ecological conditions, and disturbance types. Our
synthesis shows that median DOC concentrations across ecosystems range from $9 - 61$ mg L$^{-1}$,
which represents similar albeit slightly higher DOC concentrations when compared to the
median DOC concentrations found in top soils of other land cover groups below 50°N (25 mg L$^{-1}$
; Langeveld et al., 2020), globally distributed lakes (6 mg L$^{-1}$; Sobek et al., 2007), and lakes
across the permafrost region (11 mg L$^{-1}$; Stolpmann et al., 2021). In general, we show that
organic soils have higher DOC concentrations than mineral soils, and that DOC concentrations
are positively related to total dissolved nitrogen concentrations and negatively to C:N ratios,
which corroborate previous findings of factors correlating with DOC concentrations (Aitkenhead
& McDowell, 2000; Lajtha et al., 2005). Overall, we found that properties associated with
ecosystem type are the main constraint on DOC concentrations. Furthermore, disturbance
through permafrost thaw has little impact on measured DOC concentrations, however this may
be due to the loss of biologically reactive DOC or the loss of an initially larger pulse of DOC
having been previously mobilised prior to the timing of sampling.

*4.1 Environmental factors influencing DOC*

Our database confirmed our first hypothesis that the highest DOC concentrations would be
found in organic rich soils. Previous synthesis efforts estimating global distributions of terrestrial
DOC concentrations have presented similar findings (Guo et al., 2020; Langeveld et al., 2020).
Both of these previous studies also show that some of the highest terrestrial DOC concentrations
are found within the northern circumpolar permafrost region, highlighting that these high DOC
concentrations found in organic rich permafrost soils are of global significance. Concentrations
of DOC in the top 3 m of soils closely mirrored stocks of SOC across the circumpolar permafrost
region (Hugelius et al., 2014). Organic rich Histosol and Histel soils contain the greatest SOC





per km², followed by Turbels and Orthels (Hugelius et al., 2014), as was seen in DOC
concentrations across these soil types (Figure 2a). While the highest DOC concentrations are
found within organic rich soils, the amount of C found as DOC represent a small amount of the
total SOC pool. Using the current best estimates of Histel SOC stocks (Hugelius et al., 2020), the
DOC pool represents <1% of the total C stock in permafrost-affected peatlands as has been
shown for both permafrost and global soils (Guo et al., 2020; Prokushkin et al., 2008).
*4.2 Thermal horizons*
In many ecosystems, DOC concentrations are greatest in the active layer nearer the
surface (Figure 4). This trend has also been observed in the vertical distribution of DOC across
global soils, with 50% of the DOC pool found in the top 0 – 30 cm (Guo et al., 2020). The
production of DOC is associated with soil microbial activity (Guggenberger & Zech, 1993) and
plant inputs (Moore & Dalva, 2001), and the microbial production of DOC via input of labile
substrates has been shown to decrease with depth in permafrost (Hultman et al., 2015; Monteux
et al., 2018; Wild et al., 2016). Furthermore, the organic matter content decreases and mineral
content increases with depth, this depth trend and decrease in DOC with depth is particularly
evident between the active layer and permafrost lens in forest ecosystems (Figure 4a). While
permafrost and non-permafrost bogs do also see a shift in microbial community with depth
(Heffernan & Cavaco et al., 2022; Lamit et al., 2021), the movement of modern, surface derived
DOC down into deeper layers has also been observed (Chanton et al., 2008; Estop-Aragonés et
al., 2018). These, combined with the large, frozen SOC stores found at depth (Hugelius et al.,
2020) and hydrological isolation (Quinton, Hayashi, & Chasmer, 2011), results in a DOC pool
that remains relatively similar across thermal horizons in permafrost bogs (Figure 4b).
Intriguingly, in both coastal tundra and permafrost wetland ecosystems, DOC concentrations
were found to be higher in the permafrost lens than in the active layer. This suggests that DOC
within the active layer of these ecosystems experienced some degree of mobilization, either via
export to the aquatic network or enhanced decomposition within soils. The higher DOC
concentrations found within the permafrost lens of these ecosystems may represent a vulnerable
DOC pool to enhanced mineralization following permafrost thaw (Figure 6).
*4.3 Variation in DOC amongst permafrost zones and ecoregions*





Permafrost soils are estimated to store 1,035 ± 150 Pg C globally within the top 0-3 m

(Hugelius et al., 2014), with the highest storage of SOC found in the organic rich Histosols and
Histels. While persistent low temperatures are the main common factor which has led to the
accumulation of such high SOC amongst all permafrost soils, environmental factors associated
with the different ecosystem types are the main driving factors in differences amongst DOC
concentrations. The source of the permafrost DOC pool is from recent plant leachate inputs, or
from the decomposition and solubilization of SOC. Thus, the molecular composition of the DOC
pool is derived from a mixture of current and historical vegetation inputs. There are clear current
and historical shifts in dominant vegetation seen in the permafrost region from the south (boreal)
to north (arctic tundra), as well as across ecosystem types (upland forest, upland tundra, arctic
and boreal wetland). However, the majority of vegetation and its leachates found in the
permafrost region are generally found to produce relatively stable DOC (in terms of BDOC)
consisting of lignin-derived compounds, highly aromatic polyphenolic compounds, and low
molecular weight organic acids (Chen et al., 2018; Drake et al., 2015; Ewing et al., 2015; Selvam
et al., 2017). While differences in the stability of different DOC source end-members have been
shown (MacDonald et al., 2021), differences in redox conditions are likely a major driver in
differences in the accumulation and mineralization of DOC across permafrost ecosystem types
(Mohammed et al., 2022).

Similar to their globally significant stores of SOC (Hugelius et al., 2020), the accumulation

of high DOC concentrations found in peatlands, permafrost bogs, and permafrost wetlands, is a
result of the prevalence of cold and anoxic conditions throughout the Holocene (Blodau, 2002).
This leads to a reduction in microbial decomposition, and the accumulation of both the SOC and
DOC pool. Our results suggest that the pristine permafrost bog and permafrost wetland DOC
pool is relatively stable following permafrost thaw (Figure 6, 7a). Peatland vegetation, in
particular *Sphagnum* mosses, produces litter that has anti-microbial properties and is decay
resistant (Hamard et al., 2019; Limpens, Bohlin, & Nilsson, 2017), limiting the amount of SOC
that is degraded and assimilated into the DOC pool (Tfaily et al., 2013). This is further enhanced
by the build-up of decomposition end products and the thermodynamic constraint on decay
observed in anoxic soils (Beer et al., 2008). Permafrost has been continuously present in
peatlands across the northern circumpolar permafrost region for the past 6,000 years, with the





greatest rates of permafrost formation occurring within the past 3,000 years (Treat & Jones,
2018). Thus, a large proportion of the organic matter found peatlands and wetlands in this region
were present prior to permafrost aggradation (i.e., permafrost formation), which indicates that
permafrost formed epigenetically in these areas. Permafrost aggradation impacts soil
biogeochemical properties, leading to potentially less decomposed organic matter with higher
C/N ratios than non-permafrost equivalent soils, particularly in permafrost wetlands (Treat et al.,
2016). This can lead to the build-up of high DOC concentrations that are vulnerable to potential
mobilization following thermokarst. Decomposition in epigenetic permafrost bogs following
thermokarst has been shown to be relatively slow (Heffernan et al., 2020; Manies et al., 2021),
which further supports our finding (Figure 6) that the large DOC pool found in these systems in
relatively stable following permafrost thaw. The permafrost wetland DOC pool that accumulates
following thermokarst may represent a potentially labile DOC pool (Figure 7a), but this is likely
due to fresh, plant derived inputs rather than the exposure and mineralization of previously
frozen organic matter (Figure 7a).
Coastal tundra and forest ecosystems had similarly high DOC concentrations to those found
in permafrost bogs (Figure 3a). Coastal tundra and forest ecosystems represented the highest
concentrations of DOC in mineral permafrost soils. Concentrations of coastal permafrost DOC
were significantly lower in the active layer compared to within the permafrost lens (Figure 4a).
This is contrary to findings that deeper coastal permafrost consists of low organic matter
Pleistocene marine sediments (Bristol et al., 2021) and the proximity of the active layer to
vegetation inputs, although this productivity and inputs are vulnerable to projected climatic
warming and regional "browning" and "greening" (Lara et al., 2018). Recent work has shown
that DOC in the active layer within the coastal permafrost is more biodegradable that OC in the
permafrost lens (Speetjens et al., 2022) and a substantial proportion of organic carbon derived
from thawing coastal permafrost is vulnerable to mineralization upon thawing, particularly when
exposed to sea water (George Tanski et al., 2021). Export of terrestrial coastal permafrost DOC
directly into the Arctic Ocean can significantly influence marine biogeochemical cycles and food
webs within the Arctic ocean (Bruhn et al., 2021). Arctic coasts are eroding at rates of up to 25 m
yr$^{-1}$ (Fritz, Vonk, & Lantuit, 2017) and exporting large quantities of terrestrial organic matter
export directly to the ocean that is rapidly mineralized (Tanski et al., 2019). Enhanced DOC



export from these coastal tundra ecosystems may disrupt aquatic food webs through altering

nutrient and light supply, as has been shown for Swedish coastal systems (Peacock et al., 2022).

These coastal tundra sites represent a large DOC pool that is highly vulnerable to enhanced

mobilization and deserve further attention.

The remaining ecosystems characterised by mineral soils with an upper organic layer, i.e.,

Forests, Upland Tundra, and Yedoma, followed a clear latitudinal climate gradient of increasing

DOC concentrations from north to south. While not included in the most parsimonious PLS

model (Figure 5), Yedoma and Upland Tundra ecosystems were found to negatively correlate

with DOC concentrations (Figure A5). The greatest proportions of OC and nutrients used for

DOC production are found in shallow organic layers (Semenchuk et al., 2015; Wild et al., 2013)

in these ecosystems. Beneath the upper organic horizons in these mineral soils processes such as

sorption of DOC to minerals and the formation of Fe-DOC or Al-DOC complexes may remove

DOC from the dissolved pool (Kawahigashi et al., 2006) and mechanically protect it from

mobilization (Gentsch et al., 2015). In forest ecosystems, large amounts of SOC have

accumulated in surface organic layers (Hugelius et al., 2014) through increased vegetative inputs

due to warmer and longer growing seasons. This organic layer depth, and the impact of soil

temperature, moisture, and pH on SOC found there, strongly influences the production,

concentration, and composition of DOC (Neff & Hooper, 2002; Wickland et al., 2007).

Furthermore, the sorption of DOC to charcoal (Guggenberger et al., 2008), and high lignin and

phenolic input from vegetation (O'Donnell et al., 2016) produce a difficult to degrade DOC pool,

leading to the accumulation of the large DOC pool in this ecosystem type.

*4.4 Vulnerability of DOC to enhanced mobilization following thermokarst*

We define DOC mobilization as DOC lost from an ecosystem either via export or

degradation. Our second hypothesis that permafrost thaw would lead to enhanced mobilization of

DOC cannot be fully supported by the findings from this database. Using our chosen systematic

approach and focusing on data from terrestrial ecosystems, our database was limited to 3 studies

which represented <1% of the DOC concentration data. Several previous studies have detailed

the export of DOC in Arctic inland waters, see Table 2 in Ma et al., (2019), that have been

excluded using this approach. We acknowledge the limitation in our approach regarding the





inclusion of DOC export data. Thus, this database cannot be used to determine how permafrost
thaw will influence DOC export from terrestrial ecosystems within the northern circumpolar
permafrost region. Currently, Arctic rivers are estimated to export $25 – 36$ Tg DOC year$^{-1}$ (Amon
et al., 2012; Holmes et al., 2012), with this being dominated by modern carbon sources (Estop-
Aragonés et al., 2020), most likely derived from the top 1 m of terrestrial ecosystems. Using
current best estimates of the areal extent and soil organic carbon stores in the top 1 m of
Histosols, Histels, Orthels and Turbels (Hugelius et al., 2014), and if we assume that the DOC
pool represents ~1% of the SOC pool, we estimate that <1% of the current DOC pool found in
the top 1 m of Histosols, Histels, Orthels and Turbels is exported annually to Arctic rivers.
Quantifying the proportion of these DOC pools annually lost, and particularly the proportions
lost in headwater streams while being exported to Arctic rivers, is vital to assess the importance
of the mobilization of the terrestrial permafrost DOC pool.
Our calculated response ratios (Figure 7) for all ecosystems, indicating the difference in DOC
concentrations between pristine and permafrost thaw affected sites, partly supports of our second
hypothesis that disturbance would lead to increased export and biodegradability of DOC. The
increase in DOC following thaw observed in permafrost bogs is likely due to increased inputs
due to increased runoff and shifts in vegetation following permafrost thaw (Burd, Estop-
Aragonés, Tank, & Olefeldt, 2020), a relatively stable soil organic carbon pool at depth due to
several millennia of microbial processing (Manies et al., 2021), the prevalence of anoxic
conditions, and the potential hydrological isolation of thermokarst bogs (Quinton, Hayashi, &
Pietroniro, 2003). While not included in our analysis, DOC found near the surface of the
permafrost lens in forest ecosystems has been shown to be more biodegradable than DOC found
in the active layer (Wickland et al., 2018), and may represent a decrease in DOC following
thermokarst not captured here. Our findings of limited mobilization of permafrost bog DOC
upon thawing are supported by the findings that the $^{14}$C signature of DOC in Arctic rivers is
dominated by modern sources (Estop-Aragonés et al., 2020). However, we do see a reduction in
DOC concentrations in thermokarst affected sites at the higher latitude Yedoma upland tundra,
and permafrost wetland ecosystems. This reduction in DOC concentrations in these ecosystems
may be due to the greater biodegradability and lability of the DOC found there (Figure 6b),
supporting our third hypothesis that the most biodegradable DOC would be found in higher



latitude ecosystems. Permafrost DOC in higher latitude ecosystems, particularly Yedoma
ecosystems characterised by syngenetic permafrost aggradation which have not undergone
centuries to millennia of soil formation and microbial processes, have been shown contain a
greater proportion of low oxygen, aliphatic compounds and labile substrates (Ewing et al.,
2015b; MacDonald et al., 2021). This leads to a greater biolability and rapid mineralization of
DOC (Vonk et al., 2015), potentially causing the reduction in DOC concentrations observed
following thaw. If this hypothesis is to be found true across all high latitude ecosystems with
further data, it further highlights the vulnerability of the large DOC pool found in coastal tundra
ecosystems.

In this study, we focus on the dissolved fraction of the OC pool, however the particulate

fraction should also be considered when discussing the mobilization of terrestrial OC in
permafrost landscapes. In boreal freshwater networks, particulate organic carbon (POC)
represents a small but highly labile fraction of terrestrially derived OC exported to the fluvial
network (Attermeyer et al., 2018). The degradation of permafrost derived POC is much slower
than that of POC in the boreal freshwater network and POC derived from younger sources along
the riverbank (Shakil, Tank, Kokelj, Vonk, & Zolkos, 2020). The DOC pool in Arctic
freshwaters in dominated by modern terrestrial sources (Estop-Aragonés et al., 2020), whereas
the POC pool has been shown to be dominated by older sources in both permafrost peatland
dominated areas (Wild et al., 2019), following the formation of retrogressive thaw slumps
(Keskitalo et al., 2021), and in thermokarst affected periglacial streams (Bröder et al., 2022).
This older POC has been shown to accumulate following export due to low lability and
degradation and mineral association, which suggests that upon thermokarst formation, previously
frozen OC exported in the particulate phase is not readily consumed by microbes and that
permafrost derived DOC is the more labile fraction of exported terrestrial OC.

*4.5 Future considerations for study design*

Determining the fate of mobilized terrestrial DOC in both permafrost thaw affected, and

pristine sites should be prioritized in future studies to constrain current estimates of the
permafrost C climate feedback. There are large spatial gaps in the database, particularly in areas
with large stock of permafrost C such as the Hudson Bay Lowlands and Mackenzie River Basin,





both in Canada and two of the three largest deposits of permafrost peatland C in the circumpolar
permafrost region (Olefeldt et al., 2021). Similarly, coastal tundra sites, which along with
permafrost bog represent the ecosystems with the highest DOC concentrations, were sampled
only along the northern shoreline of Alaska and the Yukon (USA and Canada, respectively;
Table S1). From our analysis of this database, we determine that DOC mobilization is poorly
understood for terrestrial permafrost ecosystems. To address this, the two main needs of future
studies are 1) more direct estimates of DOC fluxes and export from terrestrial ecosystems into
aquatic ecosystems, and 2) more DOC degradation (BDOC) and mineralization studies. Our
results suggest that the high concentrations of DOC in permafrost bogs remains relatively stable
upon thermokarst formation, although individual studies do indicate that thawing peat may
provide a reactive source of DOC (Panneer Selvam et al., 2017). Whereas the database did not
include any studies that reported on the mineralization of DOC from coastal tundra sites. Further
sampling and assessing the mineralization of DOC is required to characterize the potential pool
of vulnerable DOC in areas with high DOC concentrations. Overall, our database and systematic
approach only included 5 studies (Olefeldt & Roulet, 2012, 2014; Olefeldt et al., 2012;
Prokushkin et al., 2006; Prokushkin et al., 2005) that explicitly reported rates of DOC discharge,
export, or fluxes from terrestrial ecosystems into the fluvial network. Given the importance of
terrestrial DOC as a source for $CO_2$ production within the aquatic network (Weyhenmeyer et al.,
2012), and the findings that previously frozen DOC is being exported to the freshwater network
(Estop-Aragones et al., 2020), improved estimates of the quantity of terrestrial DOC being
exported is essential to determine the potential aquatic greenhouse gas fluxes derived from the
mineralization of terrigenous organic matter. To improve current estimates of the permafrost C
feedback further studies are needed to determine how much DOC is laterally exported from
terrestrial ecosystems, and the mineralization potential of this DOC along the terrestrial-
freshwater-aquatic continuum.
Lastly, we suggest that future studies should consider a standardization of methods and
approached used to determine DOC concentrations for better comparison across studies. In
constructing this database we identified 3 different filter sizes, 11 different extraction procedures,
and 4 different measurement methods. The most common filter size used was 0.45 μm and this
has previously been described as the cut off to separate DOC from colloid materials (Thurman




1985; Bolan et al., 1999). In extracting DOC concentrations from soils the mostly commonly
used approach (70% of all soil samples) was via soil leaching with no chemical treatment of the
soils, although some added filtered water to promote leaching. From the seven approaches
identified to extract water samples from terrestrial sites in determining DOC, 48% of samples
were collected using a variety of suction devices and 46% done via grab samples. Of the four
DOC measurements methods the most common approach was by combustion, with 90% of all
DOC concentrations measured using this approach. As such, in order to continue measuring
DOC concentrations in terrestrial permafrost ecosystems using the most consistent approach we
suggest using 0.45 µm filters, extracting pore water via some type of sucking device or soils via
leaching, and using a combustion based method to determine DOC concentrations
**Data availability**
All data will be made freely and publicly available on an online repository prior to publication
**Author contributions**
LH, DK, and LT designed and planned the systematic review approach; LH built the database.
LH and DK analyzed the data; LH wrote the manuscript draft; DK and LT edited and reviewed
the manuscript.
**Competing interests**
The authors declare that they have no conflict of interest.
**Acknowledgements**
We thank Konstantinos Vaziourakis, Mona Abbasi, Elizabeth Jakobsson, Marloes Groeneveld,
Sarah Shakil, and Jeffrey Hawkes for helpful discussions throughout the development and
writing of this manuscript.
**Financial support**
This work was supported by the Knut and Alice Wallenberg Foundation.



**References (in text)**

Aitkenhead, J. A., & McDowell, W. H. (2000). Soil C:N ratio as a predictor of annual riverine DOC flux at local and global scales. *Global Biogeochemical Cycles*, *14*(1). https://doi.org/10.1029/1999GB900083

Amon, R. M. W., Rinehart, A. J., Duan, S., Louchouarn, P., Prokushkin, A., Guggenberger, G., … Zhulidov, A. V. (2012). Dissolved organic matter sources in large Arctic rivers. *Geochimica et Cosmochimica Acta*, *94*, 217–237. https://doi.org/https://doi.org/10.1016/j.gca.2012.07.015

Arksey, H., & O'Malley, L. (2005). Scoping studies: Towards a methodological framework. *International Journal of Social Research Methodology: Theory and Practice*, *8*(1). https://doi.org/10.1080/1364557032000119616

Attermeyer, K., Catalán, N., Einarsdottir, K., Freixa, A., Groeneveld, M., Hawkes, J. A., … Tranvik, L. J. (2018). Organic Carbon Processing During Transport Through Boreal Inland Waters: Particles as Important Sites. *Journal of Geophysical Research: Biogeosciences*, *123*(8). https://doi.org/10.1029/2018JG004500

Beckebanze, L., Runkle, B. R. K., Walz, J., Wille, C., Holl, D., Helbig, M., … Kutzbach, L. (2022). Lateral carbon export has low impact on the net ecosystem carbon balance of a polygonal tundra catchment. *BIOGEOSCIENCES*, *19*(16), 3863–3876. https://doi.org/10.5194/bg-19-3863-2022

Beer, J., Lee, K., Whiticar, M., & Blodau, C. (2008). Geochemical controls on anaerobic organic matter decomposition in a northern peatland. *Limnology and Oceanography*, *53*(4), 1393–1407. https://doi.org/10.4319/lo.2008.53.4.1393

Biester, H., Knorr, K. H., Schellekens, J., Basler, A., & Hermanns, Y. M. (2014). Comparison of different methods to determine the degree of peat decomposition in peat bogs. *Biogeosciences*. https://doi.org/10.5194/bg-11-2691-2014

Blodau, C. (2002). Carbon cycling in peatlands — A review of processes and controls. *Environmental Reviews*, *10*(2), 111–134. https://doi.org/10.1139/a02-004

Bolan, N.S., Baskaran, S., Thiagarajan, S. (1999). Methods of Measurement of Dissolved Organic Carbon of Plant Origin in Soils, Manures, Sludges and Stream Water. In: Linskens, H.F., Jackson, J.F. (eds) Analysis of Plant Waste Materials. Modern Methods of Plant Analysis, vol 20. Springer, Berlin, Heidelberg. https://doi.org/10.1007/978-3-662-03887-1_1

Bristol, E. M., Connolly, C. T., Lorenson, T. D., Richmond, B. M., Ilgen, A. G., Choens, R. C., … McClelland, J. W. (2021). Geochemistry of Coastal Permafrost and Erosion-Driven Organic Matter Fluxes to the Beaufort Sea Near Drew Point, Alaska. *Frontiers in Earth Science*, *8*. https://doi.org/10.3389/feart.2020.598933





Bröder, L., Hirst, C., Opfergelt, S., Thomas, M., Vonk, J. E., Haghipour, N., … Fouché, J.
(2022). Contrasting Export of Particulate Organic Carbon From Greenlandic Glacial and
Nonglacial Streams. *Geophysical Research Letters*, *49*(21).
https://doi.org/10.1029/2022GL101210

Brown, J., Ferrians Jr., O. J., Heginbottom, J. A., & Melnikov, E. S. (1997). Circum-Arctic map
of permafrost and ground ice conditions. *USGS Numbered Series*, 1.
https://doi.org/10.1016/j.jallcom.2010.03.054

Bruhn, A. D., Stedmon, C. A., Comte, J., Matsuoka, A., Speetjens, N. J., Tanski, G., … Sjöstedt,
899        J. (2021). Terrestrial Dissolved Organic Matter Mobilized From Eroding Permafrost
Controls Microbial Community Composition and Growth in Arctic Coastal Zones.
*Frontiers in Earth Science*, *9*. https://doi.org/10.3389/feart.2021.640580

Burd, K., Estop-Aragonés, C., Tank, S. E., & Olefeldt, D. (2020). Lability of dissolved organic
carbon from boreal peatlands: interactions between permafrost thaw, wildfire, and season.
*Canadian Journal of Soil Science*, *13*(February), 1–13. https://doi.org/10.1139/cjss-2019-
0154

Camill, P. (2005). Permafrost thaw accelerates in boreal peatlands during late-20th century
climate warming. *Climatic Change*, *68*(1–2), 135–152. https://doi.org/10.1007/s10584-005-
4785-y

Chanton, J. P., Glaser, P. H., Chasar, L. S., Burdige, D. J., Hines, M. E., Siegel, D. I., … Cooper,
910        W. T. (2008). Radiocarbon evidence for the importance of surface vegetation on
fermentation and methanogenesis in contrasting types of boreal peatlands. *Global
Biogeochemical Cycles*, *22*(4), 1–11. https://doi.org/10.1029/2008GB003274

Chen, H., Yang, Z., Chu, R. K., Tolic, N., Liang, L., Graham, D. E., … Gu, B. (2018). Molecular
Insights into Arctic Soil Organic Matter Degradation under Warming. *ENVIRONMENTAL
SCIENCE & TECHNOLOGY*, *52*(8), 4555–4564. https://doi.org/10.1021/acs.est.7b05469

Connon, R. F., Quinton, W. L., Craig, J. R., & Hayashi, M. (2014). Changing hydrologic
connectivity due to permafrost thaw in the lower Liard River valley, NWT, Canada.
*Hydrological Processes*, *28*(14). https://doi.org/10.1002/hyp.10206

Dean, J. F., Meisel, O. H., Rosco, M. M., Marchesini, L. B., Garnett, M. H., Lenderink, H., …
Dolman, A. J. (2020). East Siberian Arctic inland waters emit mostly contemporary carbon.
*NATURE COMMUNICATIONS*, *11*(1). https://doi.org/10.1038/s41467-020-15511-6

Drake, T. W., Wickland, K. P., Spencer, R. G. M., McKnight, D. M., & Striegl, R. G. (2015).
Ancient low-molecular-weight organic acids in permafrost fuel rapid carbon dioxide
production upon thaw. *PROCEEDINGS OF THE NATIONAL ACADEMY OF SCIENCES
OF THE UNITED STATES OF AMERICA*, *112*(45), 13946–13951.
https://doi.org/10.1073/pnas.1511705112

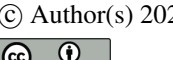



Estop-Aragones, C., Olefeldt, D., Abbott, B. W., Chanton, J. P., Czimczik, C. I., Dean, J. F., …
Anthony, K. W. (2020). Assessing the Potential for Mobilization of Old Soil Carbon After
Permafrost Thaw: A Synthesis of C-14 Measurements From the Northern Permafrost
Region. *GLOBAL BIOGEOCHEMICAL CYCLES*, *34*(9).
https://doi.org/10.1029/2020GB006672
Estop-Aragonés, Cristian, Czimczik, C. I., Heffernan, L., Gibson, C., Walker, J. C., Xu, X., &
Olefeldt, D. (2018). Respiration of aged soil carbon during fall in permafrost peatlands
enhanced by active layer deepening following wildfire but limited following thermokarst.
*Environmental Research Letters*, *13*(8). https://doi.org/10.1088/1748-9326/aad5f0
Ewing, S. A., Paces, J. B., O'Donnell, J. A., Jorgenson, M. T., Kanevskiy, M. Z., Aiken, G. R.,
… Striegl, R. (2015a). Uranium isotopes and dissolved organic carbon in loess permafrost:
Modeling the age of ancient ice. *GEOCHIMICA ET COSMOCHIMICA ACTA*, *152*, 143–
165. https://doi.org/10.1016/j.gca.2014.11.008
Ewing, S. A., Paces, J. B., O'Donnell, J. A., Jorgenson, M. T., Kanevskiy, M. Z., Aiken, G. R.,
… Striegl, R. (2015b). Uranium isotopes and dissolved organic carbon in loess permafrost:
Modeling the age of ancient ice. *Geochimica et Cosmochimica Acta*, *152*, 143–165.
https://doi.org/10.1016/j.gca.2014.11.008
Fritz, M., Vonk, J. E., & Lantuit, H. (2017). Collapsing Arctic coastlines. *Nature Climate
Change*. https://doi.org/10.1038/nclimate3188
Gentsch, N., Mikutta, R., Shibistova, O., Wild, B., Schnecker, J., Richter, A., … Guggenberger,
G. (2015). Properties and bioavailability of particulate and mineral-associated organic
matter in Arctic permafrost soils, Lower Kolyma Region, Russia. *European Journal of Soil
Science*, *66*(4). https://doi.org/10.1111/ejss.12269
Guggenberger, G., & Zech, W. (1993). Dissolved organic carbon control in acid forest soils of
the Fichtelgebirge (Germany) as revealed by distribution patterns and structural
composition analyses. *Geoderma*, *59*(1–4). https://doi.org/10.1016/0016-7061(93)90065-S
Guggenberger, Georg, Rodionov, A., Shibistova, O., Grabe, M., Kasansky, O. A., Fuchs, H., …
Flessa, H. (2008). Storage and mobility of black carbon in permafrost soils of the forest
tundra ecotone in Northern Siberia. *Global Change Biology*, *14*(6), 1367–1381.
https://doi.org/10.1111/j.1365-2486.2008.01568.x
Guo, Z., Wang, Y., Wan, Z., Zuo, Y., He, L., Li, D., … Xu, X. (2020). Soil dissolved organic
carbon in terrestrial ecosystems: Global budget, spatial distribution and controls. *Global
Ecology and Biogeography*, *29*(12). https://doi.org/10.1111/geb.13186
Hamard, S., Robroek, B. J. M., Allard, P. M., Signarbieux, C., Zhou, S., Saesong, T., … Jassey,
V. E. J. (2019). Effects of Sphagnum Leachate on Competitive Sphagnum Microbiome
Depend on Species and Time. *Frontiers in Microbiology*, *10*.
https://doi.org/10.3389/fmicb.2019.02042





Hansen, A. M., Kraus, T. E. C., Pellerin, B. A., Fleck, J. A., Downing, B. D., & Bergamaschi, B.
965       A. (2016). Optical properties of dissolved organic matter (DOM): Effects of biological and
photolytic degradation. *Limnology and Oceanography*, **61**(3), 1015–
1032. https://doi.org/10.1002/lno.10270

Heffernan, L., Cavaco, M. A., Bhatia, M. P., Estop-Aragonés, C., Knorr, K.-H., & Olefeldt, D.
(2022). High peatland methane emissions following permafrost thaw: enhanced acetoclastic
methanogenesis during early successional stages. *Biogeosciences*, *19*(12).
https://doi.org/10.5194/bg-19-3051-2022

Heffernan, L., Estop-Aragonés, C., Knorr, K.-H., Talbot, J., & Olefeldt, D. (2020). Long-term
impacts of permafrost thaw on carbon storage in peatlands: deep losses offset by surficial
accumulation. *Journal of Geophysical Research: Biogeosciences*, *2011*(2865),
e2019JG005501. https://doi.org/10.1029/2019JG005501

Holmes, R. M., McClelland, J. W., Peterson, B. J., Tank, S. E., Bulygina, E., Eglinton, T. I., …
Zimov, S. A. (2012). Seasonal and Annual Fluxes of Nutrients and Organic Matter from
Large Rivers to the Arctic Ocean and Surrounding Seas. *ESTUARIES AND COASTS*, *35*(2),
369–382. https://doi.org/10.1007/s12237-011-9386-6

Hugelius, G., Strauss, J., Zubrzycki, S., Harden, J. W., Schuur, E. A. G., Ping, C. L., … Kuhry,
P. (2014). Estimated stocks of circumpolar permafrost carbon with quantified uncertainty
ranges and identified data gaps. *Biogeosciences*, *11*(23), 6573–6593.
https://doi.org/10.5194/bg-11-6573-2014

Hugelius, Gustaf, Loisel, J., Chadburn, S., Jackson, R. B., Jones, M., MacDonald, G., … Yu, Z.
(2020). Large stocks of peatland carbon and nitrogen are vulnerable to permafrost thaw.
*Proceedings of the National Academy of Sciences*, *117*(34), 20438–20446.
https://doi.org/10.1073/pnas.1916387117

Hultman, J., Waldrop, M. P., Mackelprang, R., David, M. M., McFarland, J., Blazewicz, S. J., …
Jansson, J. K. (2015). Multi-omics of permafrost, active layer and thermokarst bog soil
microbiomes. *Nature*, *521*(7551). https://doi.org/10.1038/nature14238

Jorgenson, M. T., Shur, Y. L., & Pullman, E. R. (2006). Abrupt increase in permafrost
degradation in Arctic Alaska. *Geophysical Research Letters*, *33*(2).
https://doi.org/10.1029/2005GL024960

Kawahigashi, M., Kaiser, K., Rodionov, A., & Guggenberger, G. (2006). Sorption of dissolved
organic matter by mineral soils of the Siberian forest tundra. *GLOBAL CHANGE
BIOLOGY*, *12*(10), 1868–1877. https://doi.org/10.1111/j.1365-2486.2006.01203.x

Keskitalo, K. H., Bröder, L., Shakil, S., Zolkos, S., Tank, S. E., van Dongen, B. E., … Vonk, J.
E. (2021). Downstream Evolution of Particulate Organic Matter Composition From
Permafrost Thaw Slumps. *Frontiers in Earth Science*, *9*.
https://doi.org/10.3389/feart.2021.642675



Kicklighter, D. W., Hayes, D. J., McClelland, J. W., Peterson, B. J., McGuire, A. D., & Melillo, J. M. (2013). Insights and issues with simulating terrestrial DOC loading of Arctic river networks. *ECOLOGICAL APPLICATIONS*, *23*(8), 1817–1836. https://doi.org/10.1890/11-1050.1

Kokelj, S. V., & Jorgenson, M. T. (2013). Advances in thermokarst research. *Permafrost and Periglacial Processes*, *24*(2), 108–119. https://doi.org/10.1002/ppp.1779

Lajeunesse, M. J. (2011). On the meta-analysis of response ratios for studies with correlated and multi-group designs. *Ecology*, *92*(11). https://doi.org/10.1890/11-0423.1

Lajtha, K., Crow, S. E., Yano, Y., Kaushal, S. S., Sulzman, E., Sollins, P., & Spears, J. D. H. (2005). Detrital controls on soil solution N and dissolved organic matter in soils: A field experiment. *Biogeochemistry*, *76*(2). https://doi.org/10.1007/s10533-005-5071-9

Lamit, L. J., Romanowicz, K. J., Potvin, L. R., Lennon, J. T., Tringe, S. G., Chimner, R. A., … Lilleskov, E. A. (2021). Peatland microbial community responses to plant functional group and drought are depth-dependent. *Molecular Ecology*, *30*(20). https://doi.org/10.1111/mec.16125

Langeveld, J., Bouwman, A. F., van Hoek, W. J., Vilmin, L., Beusen, A. H. W., Mogollón, J. M., & Middelburg, J. J. (2020). Estimating dissolved carbon concentrations in global soils: a global database and model. *SN Applied Sciences*, *2*(10), 1–21. https://doi.org/10.1007/s42452-020-03290-0

Lara, M. J., Nitze, I., Grosse, G., Martin, P., & David McGuire, A. (2018). Reduced arctic tundra productivity linked with landform and climate change interactions. *Scientific Reports*, *8*(1). https://doi.org/10.1038/s41598-018-20692-8

Liljedahl, A. K., Boike, J., Daanen, R. P., Fedorov, A. N., Frost, G. V., Grosse, G., … Zona, D. (2016). Pan-Arctic ice-wedge degradation in warming permafrost and its influence on tundra hydrology. *Nature Geoscience*, *9*(4). https://doi.org/10.1038/ngeo2674

Limpens, J., Bohlin, E., & Nilsson, M. B. (2017). Phylogenetic or environmental control on the elemental and organo-chemical composition of Sphagnum mosses? *Plant and Soil*. https://doi.org/10.1007/s11104-017-3239-4

Ma, Q., Jin, H., Yu, C., & Bense, V. F. (2019). Dissolved organic carbon in permafrost regions: A review. *Science China Earth Sciences*. https://doi.org/10.1007/s11430-018-9309-6

MacDonald, E. N., Tank, S. E., Kokelj, S. V., Froese, D. G., & Hutchins, R. H. S. (2021). Permafrost-derived dissolved organic matter composition varies across permafrost end-members in the western Canadian Arctic. *Environmental Research Letters*, *16*(2). https://doi.org/10.1088/1748-9326/abd971

Manies, K. L., Jones, M. C., Waldrop, M. P., Leewis, M. C., Fuller, C., Cornman, R. S., &



Hoefke, K. (2021). Influence of Permafrost Type and Site History on Losses of Permafrost
Carbon After Thaw. *Journal of Geophysical Research: Biogeosciences*, *126*(11).
https://doi.org/10.1029/2021JG006396
McGuire, A. D., Lawrence, D. M., Koven, C., Clein, J. S., Burke, E., Chen, G., … Zhuang, Q.
(2018). Dependence of the evolution of carbon dynamics in the northern permafrost region
on the trajectory of climate change. *Proceedings of the National Academy of Sciences of the*
*United States of America*, *115*(15). https://doi.org/10.1073/pnas.1719903115
Mehmood, T., Liland, K. H., Snipen, L., & Sæbø, S. (2012). A review of variable selection
methods in Partial Least Squares Regression. *Chemometrics and Intelligent Laboratory*
*Systems*. https://doi.org/10.1016/j.chemolab.2012.07.010
Mevik, B. H., & Wehrens, R. (2007). The pls package: Principal component and partial least
squares regression in R. *Journal of Statistical Software*, *18*(2).
https://doi.org/10.18637/jss.v018.i02
Miner, K. R., Turetsky, M. R., Malina, E., Bartsch, A., Tamminen, J., McGuire, A. D., … Miller,
C. E. (2022). Permafrost carbon emissions in a changing Arctic. *Nature Reviews Earth and*
*Environment*. https://doi.org/10.1038/s43017-021-00230-3
Mohammed, A. A., Guimond, J. A., Bense, V. F., Jamieson, R. C., McKenzie, J. M., & Kurylyk,
B. L. (2022). Mobilization of subsurface carbon pools driven by permafrost thaw and
reactivation of groundwater flow: a virtual experiment. *Environmental Research Letters*,
*17*(12), 124036. https://doi.org/10.1088/1748-9326/ACA701
Monteux, S., Weedon, J. T., Blume-Werry, G., Gavazov, K., Jassey, V. E. J., Johansson, M., …
Dorrepaal, E. (2018). Long-term in situ permafrost thaw effects on bacterial communities
and potential aerobic respiration. *ISME Journal*, *12*(9), 2129–2141.
https://doi.org/10.1038/s41396-018-0176-z
Moore, T. R., & Dalva, M. (2001). Some controls on the release of dissolved organic carbon by
plant tissues and soils. *Soil Science*, *166*(1), 38–47. https://doi.org/10.1097/00010694-
200101000-00007

Neff, J. C., & Hooper, D. U. (2002). Vegetation and climate controls on potential CO2, DOC and
DON production in northern latitude soils. *Global Change Biology*, *8*(9), 872–884.
https://doi.org/10.1046/j.1365-2486.2002.00517.x
O'Donnell, J. A., Aiken, G. R., Butler, K. D., Guillemette, F., Podgorski, D. C., & Spencer, R.
G. M. (2016). DOM composition and transformation in boreal forest soils: The effects of
temperature and organic-horizon decomposition state. *Journal of Geophysical Research:*
*Biogeosciences*, *121*(10), 2727–2744. https://doi.org/10.1002/2016JG003431.Received
Olefeldt, D., Heffernan, L., Jones, M. C., Sannel, A. B. K., Treat, C. C., & Turetsky, M. R.
(2021). Permafrost Thaw in Northern Peatlands: Rapid Changes in Ecosystem and



Landscape Functions (pp. 27–67). https://doi.org/10.1007/978-3-030-71330-0_3
Olefeldt, D., & Roulet, N. T. (2012). Effects of permafrost and hydrology on the composition
and transport of dissolved organic carbon in a subarctic peatland complex. *Journal of*
*Geophysical Research: Biogeosciences*, *117*(1). https://doi.org/10.1029/2011JG001819

Olefeldt, D., & Roulet, N. T. (2014). Permafrost conditions in peatlands regulate magnitude,
timing, and chemical composition of catchment dissolved organic carbon export. *GLOBAL*
*CHANGE BIOLOGY*, *20*(10), 3122–3136. https://doi.org/10.1111/gcb.12607

Olefeldt, D., Roulet, N. T., Bergeron, O., Crill, P., Bäckstrand, K., & Christensen, T. R. (2012).
Net carbon accumulation of a high-latitude permafrost palsa mire similar to permafrost-free
peatlands. *Geophysical Research Letters*, *39*(3). https://doi.org/10.1029/2011GL050355

Olson, D. M., Dinerstein, E., Wikramanayake, E. D., Burgess, N. D., Powell, G. V. N.,
Underwood, E. C., … others. (2001). Terrestrial Ecoregions of the World: A New Map of
Life on Earth: A new global map of terrestrial ecoregions provides an innovative tool for
conserving biodiversity. *BioScience*, *51*(11).

Panneer Selvam, B., Lapierre, J.-F., Guillemette, F., Voigt, C., Lamprecht, R. E., Biasi, C., …
Berggren, M. (2017). Degradation potentials of dissolved organic carbon (DOC) from
thawed permafrost peat. *SCIENTIFIC REPORTS*, *7*, 45811.
https://doi.org/10.1038/srep45811

Peacock, M., Futter, M. N., Jutterström, S., Kothawala, D. N., Moldan, F., Stadmark, J., &
Evans, C. D. (2022). Three Decades of Changing Nutrient Stoichiometry from Source to
Sea on the Swedish West Coast. *Ecosystems*, *25*(8). https://doi.org/10.1007/s10021-022-
00798-x

Pries, C. E. H., Schuur, E. A. G., & Crummer, K. G. (2012). Holocene Carbon Stocks and
Carbon Accumulation Rates Altered in Soils Undergoing Permafrost Thaw. *Ecosystems*,
*15*(1). https://doi.org/10.1007/s10021-011-9500-4

Prokushkin, A. S., Gavrilenko, I. V., Abaimov, A. P., Prokushkin, S. G., & Samusenko, A. V.
(2006). Dissolved organic carbon in upland forested watersheds underlain by continuous
permafrost in Central Siberia. *Mitigation and Adaptation Strategies for Global Change*,
*11*(1), 223–240. https://doi.org/10.1007/s11027-006-1022-6

Prokushkin, A S, Kajimoto, T., Prokushkin, S. G., McDowell, W. H., Abaimov, A. P., &
Matsuura, Y. (2005). Climatic factors influencing fluxes of dissolved organic carbon from
the forest floor in a continuous-permafrost Siberian watershed. *CANADIAN JOURNAL OF*
*FOREST RESEARCH*, *35*(9), 2130–2140. https://doi.org/10.1139/X05-150

Prokushkin, Anatoly S., Kawahigashi, M., & Tokareva, I. V. (2008). Global Warming and
Dissolved Organic Carbon Release from Permafrost Soils. In *Permafrost Soils* (pp. 237–
250). https://doi.org/10.1007/978-3-540-69371-0_16



Quinton, W. L., Hayashi, M., & Chasmer, L. E. (2011). Permafrost-thaw-induced land-cover change in the Canadian subarctic: Implications for water resources. *Hydrological Processes*, *25*(1), 152–158. https://doi.org/10.1002/hyp.7894

Quinton, W. L., Hayashi, M., & Pietroniro, A. (2003). Connectivity and storage functions of channel fens and flat bogs in northern basins. *Hydrological Processes*. https://doi.org/10.1002/hyp.1369

Rantanen, M., Karpechko, A., Lipponen, A., Nordling, K., Hyvärinen, O., Ruosteenoja, K., … Laaksonen, A. (2021). The Arctic has warmed four times faster than the globe since 1980. *Nature Portfolio*, (2022), 0–29. https://doi.org/https://doi.org/10.1038/s43247-022-00498-3

Raymond, P. A., McClelland, J. W., Holmes, R. M., Zhulidov, A. V., Mull, K., Peterson, B. J., … Gurtovaya, T. Y. (2007). Flux and age of dissolved organic carbon exported to the Arctic Ocean: A carbon isotopic study of the five largest arctic rivers. *Global Biogeochemical Cycles*, *21*(4). https://doi.org/10.1029/2007GB002934

Ripley, B., Venables, B., Bates, D. M., Hornik, K., Gebhardt, A., & Firth, D. (2019). Package 'MASS' (Version 7.3-51.4). *Cran-R Project*.

Schaefer, K., Lantuit, H., Romanovsky, V. E., Schuur, E. A. G., & Witt, R. (2014). The impact of the permafrost carbon feedback on global climate. *Environmental Research Letters*. https://doi.org/10.1088/1748-9326/9/8/085003

Schuur, E. A. G., Bracho, R., Celis, G., Belshe, E. F., Ebert, C., Ledman, J., … Webb, E. E. (2021). Tundra Underlain By Thawing Permafrost Persistently Emits Carbon to the Atmosphere Over 15 Years of Measurements. *Journal of Geophysical Research: Biogeosciences*, *126*(6), 1–23. https://doi.org/10.1029/2020jg006044

Schuur, T., McGuire, A. D., Romanovsky, V., Schädel, C., & Mack, M. (2018). Chapter 11: Arctic and Boreal Carbon. Second State of the Carbon Cycle Report. *Second State of the Carbon Cycle Report (SOCCR2): A Sustained Assessment Report*, 428–468. Retrieved from https://carbon2018.globalchange.gov/chapter/11/

Selvam, B. P., Lapierre, J.-F., Guillemette, F., Voigt, C., Lamprecht, R. E., Biasi, C., … Berggren, M. (2017). Degradation potentials of dissolved organic carbon (DOC) from thawed permafrost peat. *SCIENTIFIC REPORTS*, *7*. https://doi.org/10.1038/srep45811

Semenchuk, P. R., Elberling, B., Amtorp, C., Winkler, J., Rumpf, S., Michelsen, A., & Cooper, E. J. (2015). Deeper snow alters soil nutrient availability and leaf nutrient status in high Arctic tundra. *Biogeochemistry*, *124*(1–3), 81–94. https://doi.org/10.1007/s10533-015-0082-7

Shakil, S., Tank, S. E., Kokelj, S. V., Vonk, J. E., & Zolkos, S. (2020). Particulate dominance of organic carbon mobilization from thaw slumps on the Peel Plateau, NT: Quantification and implications for stream systems and permafrost carbon release. *Environmental Research*



*Letters*, *15*(11). https://doi.org/10.1088/1748-9326/abac36

Sobek, S., Tranvik, L. J., Prairie, Y. T., Kortelainen, P., & Cole, J. J. (2007). Patterns and
regulation of dissolved organic carbon: An analysis of 7,500 widely distributed lakes.
*Limnology and Oceanography*, *52*(3). https://doi.org/10.4319/lo.2007.52.3.1208

Speetjens, N. J., Tanski, G., Martin, V., Wagner, J., Richter, A., Hugelius, G., … Vonk, J. E.
(2022). Dissolved organic matter characterization in soils and streams in a small coastal
low-arctic catchment. *Biogeosciences*, *19*(July), 3073–3097. Retrieved from
https://doi.org/10.5194/bg-19-3073-2022

Stolpmann, L., Coch, C., Morgenstern, A., Boike, J., Fritz, M., Herzschuh, U., … Grosse, G.
(2021). First pan-Arctic assessment of dissolved organic carbon in lakes of the permafrost
region. *BIOGEOSCIENCES*, *18*(12), 3917–3936. https://doi.org/10.5194/bg-18-3917-2021

Strauss, J., Laboor, S., Schirrmeister, L., Fedorov, A. N., Fortier, D., Froese, D., … Grosse, G.
(2021). Circum-Arctic Map of the Yedoma Permafrost Domain. *Frontiers in Earth Science*,
*9*. https://doi.org/10.3389/feart.2021.758360

Striegl, R. G., Aiken, G. R., Dornblaser, M. M., Raymond, P. A., & Wickland, K. P. (2005). A
decrease in discharge-normalized DOC export by the Yukon River during summer through
autumn. *GEOPHYSICAL RESEARCH LETTERS*, *32*(21).
https://doi.org/10.1029/2005GL024413

Tank, S. E., Frey, K. E., Striegl, R. G., Raymond, P. A., Holmes, R. M., McClelland, J. W., &
Peterson, B. J. (2012). Landscape-level controls on dissolved carbon flux from diverse
catchments of the circumboreal. *GLOBAL BIOGEOCHEMICAL CYCLES*, *26*.
https://doi.org/10.1029/2012GB004299

Tanski, G., Wagner, D., Knoblauch, C., Fritz, M., Sachs, T., & Lantuit, H. (2019). Rapid CO2
Release From Eroding Permafrost in Seawater. *Geophysical Research Letters*, *46*(20).
https://doi.org/10.1029/2019GL084303

Tanski, George, Bröder, L., Wagner, D., Knoblauch, C., Lantuit, H., Beer, C., … Vonk, J. E.
(2021). Permafrost Carbon and CO2 Pathways Differ at Contrasting Coastal Erosion Sites
in the Canadian Arctic. *Frontiers in Earth Science*, *9*.
https://doi.org/10.3389/feart.2021.630493

Tfaily, M. M., Hamdan, R., Corbett, J. E., Chanton, J. P., Glaser, P. H., & Cooper, W. T. (2013).
Investigating dissolved organic matter decomposition in northern peatlands using
complimentary analytical techniques. *Geochimica et Cosmochimica Acta*.
https://doi.org/10.1016/j.gca.2013.03.002

Thurman, E. M. (1985). Organic geochemistry of natural waters (Vol. 2). Springer Science &
Business Media.





Treat, C. C., Jones, M. C., Camill, P., Gallego-Sala, A., Garneau, M., Harden, J. W., …
Väliranta, M. (2016). Effects of permafrost aggradation on peat properties as determined
from a pan-Arctic synthesis of plant macrofossils. *Journal of Geophysical Research:*
*Biogeosciences*, *121*(1), 78–94. https://doi.org/10.1002/2015JG003061

Treat, Claire C., & Jones, M. C. (2018). Near-surface permafrost aggradation in Northern
Hemisphere peatlands shows regional and global trends during the past 6000 years.
*Holocene*. https://doi.org/10.1177/0959683617752858

Turetsky, M. R., Wieder, R. K., Vitt, D. H., Evans, R. J., & Scott, K. D. (2007). The
disappearance of relict permafrost in boreal north America: Effects on peatland carbon
storage and fluxes. *Global Change Biology*, *13*(9), 1922–1934.
https://doi.org/10.1111/j.1365-2486.2007.01381.x

Turetsky, Merritt R., Abbott, B. W., Jones, M. C., Anthony, K. W., Olefeldt, D., Schuur, E. A.
G., … McGuire, A. D. (2020). Carbon release through abrupt permafrost thaw. *Nature*
*Geoscience*. https://doi.org/10.1038/s41561-019-0526-0

USDA. (1999). *Soil Taxonomy: A Basic System of Soil Classification for Making and*
*Interpreting Soil Surveys, 2nd Edition*. *Landscape and Land Capacity*.

Varner, R. K., Crill, P. M., Frolking, S., McCalley, C. K., Burke, S. A., Chanton, J. P., … Palace,
1196        M. W. (2022). Permafrost thaw driven changes in hydrology and vegetation cover increase
trace gas emissions and climate forcing in Stordalen Mire from 1970 to 2014. *Philosophical*
*Transactions of the Royal Society A: Mathematical, Physical and Engineering Sciences*,
*380*(2215). https://doi.org/10.1098/rsta.2021.0022

Vonk, J E, Tank, S. E., Mann, P. J., Spencer, R. G. M., Treat, C. C., Striegl, R. G., … Wickland,
1201        K. P. (2015). Biodegradability of dissolved organic carbon in permafrost soils and aquatic
systems: a meta-analysis. *BIOGEOSCIENCES*, *12*(23), 6915–6930.
https://doi.org/10.5194/bg-12-6915-2015

Vonk, Jorien E., & Gustafsson, Ö. (2013). Permafrost-carbon complexities. *Nature Geoscience*.
https://doi.org/10.1038/ngeo1937

Weishaar, J. L., Aiken, G. R., Bergamaschi, B. A., Fram, M. S., Fujii, R., & Mopper, K. (2003).
Evaluation of specific ultraviolet absorbance as an indicator of the chemical composition
and reactivity of dissolved organic carbon. Environmental Science and Technology, 37(20),
4702–4708. https://doi.org/10.1021/es030360x

Weyhenmeyer, G. A., Fröberg, M., Karltun, E., Khalili, M., Kothawala, D., Temnerud, J., &
Tranvik, L. J. (2012). Selective decay of terrestrial organic carbon during transport from
land to sea. *Global Change Biology*, *18*(1). https://doi.org/10.1111/j.1365-
2486.2011.02544.x

Wickland, K.P., Neff, J. C., & Aiken, G. R. (2007). Dissolved organic carbon in Alaskan boreal





forest: Sources, chemical characteristics, and biodegradability. *Ecosystems*, *10*(8), 1323–
1340. https://doi.org/10.1007/s10021-007-9101-4

Wickland, Kimberly P, Waldrop, M. P., Aiken, G. R., Koch, J. C., Jorgenson, Mt., & Striegl, R.
G. (2018). Dissolved organic carbon and nitrogen release from boreal Holocene permafrost
and seasonally frozen soils of Alaska. *ENVIRONMENTAL RESEARCH LETTERS*, *13*(6).
https://doi.org/10.1088/1748-9326/aac4ad

Wild, B., Andersson, A., Broder, L., Vonk, J., Hugelius, G., McClelland, J. W., … Gustafsson,
O. (2019). Rivers across the Siberian Arctic unearth the patterns of carbon release from
thawing permafrost. *PROCEEDINGS OF THE NATIONAL ACADEMY OF SCIENCES OF
THE UNITED STATES OF AMERICA*, *116*(21), 10280–10285.
https://doi.org/10.1073/pnas.1811797116

Wild, B., Gentsch, N., Capek, P., Diáková, K., Alves, R. J. E., Bárta, J., … Richter, A. (2016).
Plant-derived compounds stimulate the decomposition of organic matter in arctic permafrost
soils. *Scientific Reports*, *6*. https://doi.org/10.1038/srep25607

Wild, B., Schnecker, J., Bárta, J., Čapek, P., Guggenberger, G., Hofhansl, F., … Richter, A.
(2013). Nitrogen dynamics in Turbic Cryosols from Siberia and Greenland. *Soil Biology
and Biochemistry*, *67*, 85–93. https://doi.org/https://doi.org/10.1016/j.soilbio.2013.08.004

Woo, M. (1986). Permafrost hydrology in north america. *Atmosphere - Ocean*, *24*(3).
https://doi.org/10.1080/07055900.1986.9649248









**Studies used to generate database**
Abbott, B. W., Jones, J. B., Godsey, S. E., Larouche, J. R., & Bowden, W. B. (2015). Patterns
and persistence of hydrologic carbon and nutrient export from collapsing upland permafrost.
*Biogeosciences*, *12*(12), 3725–3740. https://doi.org/10.5194/bg-12-3725-2015


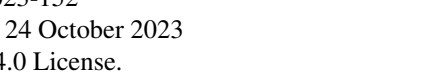

Abbott, B. W., Larouche, J. R., Jones, J. J. B., Bowden, W. B., & Balser, A. W. (2014). From
Thawing and Collapsing Permafrost. *Journal of Geophysical Research: Biogeosciences*,
*119*, 2049–2063. https://doi.org/10.1002/2014JG002678.Received
Beckebanze, L., Runkle, B. R. K., Walz, J., Wille, C., Holl, D., Helbig, M., … Kutzbach, L.
(2022). Lateral carbon export has low impact on the net ecosystem carbon balance of a
polygonal tundra catchment. *BIOGEOSCIENCES*, *19*(16), 3863–3876.
https://doi.org/10.5194/bg-19-3863-2022
Boddy, E., Roberts, P., Hill, P. W., Farrar, J., & Jones, D. L. (2008). Turnover of low molecular
weight dissolved organic C (DOC) and microbial C exhibit different temperature
sensitivities in Arctic tundra soils. *SOIL BIOLOGY & BIOCHEMISTRY*, *40*(7), 1557–1566.
https://doi.org/10.1016/j.soilbio.2008.01.030
Bristol, E. M., Connolly, C. T., Lorenson, T. D., Richmond, B. M., Ilgen, A. G., Choens, R. C.,
… McClelland, J. W. (2021). Geochemistry of Coastal Permafrost and Erosion-Driven
Organic Matter Fluxes to the Beaufort Sea Near Drew Point, Alaska. *Frontiers in Earth
Science*, *8*. https://doi.org/10.3389/feart.2020.598933
Bruhn, A. D., Stedmon, C. A., Comte, J., Matsuoka, A., Speetjens, N. J., Tanski, G., … Sjöstedt,
J. (2021). Terrestrial Dissolved Organic Matter Mobilized From Eroding Permafrost
Controls Microbial Community Composition and Growth in Arctic Coastal Zones.
*Frontiers in Earth Science*, *9*. https://doi.org/10.3389/feart.2021.640580
Buckeridge, K. M., & Grogan, P. (2008). Deepened snow alters soil microbial nutrient
limitations in arctic birch hummock tundra. *Applied Soil Ecology*, *39*(2), 210–222.
https://doi.org/https://doi.org/10.1016/j.apsoil.2007.12.010
Burd, K., Estop-Aragonés, C., Tank, S. E., & Olefeldt, D. (2020). Lability of dissolved organic
carbon from boreal peatlands: interactions between permafrost thaw, wildfire, and season.
*Canadian Journal of Soil Science*, *13*(February), 1–13. https://doi.org/10.1139/cjss-2019-
0154

Burd, K., Tank, S. E., Dion, N., Quinton, W. L., Spence, C., Tanentzap, A. J., & Olefeldt, D.
(2018). Seasonal shifts in export of DOC and nutrients from burned and unburned peatland-
rich catchments, Northwest Territories, Canada. *Hydrology and Earth System Sciences*,
4455–4472. https://doi.org/10.5194/hess-22-4455-2018
Carey, S. K. (2003). Dissolved organic carbon fluxes in a discontinuous permafrost subarctic
alpine catchment. *PERMAFROST AND PERIGLACIAL PROCESSES*, *14*(2), 161–171.
https://doi.org/10.1002/ppp.444
Chiasson-Poirier, G., Franssen, J., Lafreniere, M. J., Fortier, D., & Lamoureux, S. F. (2020).
Seasona evolution of active layer thaw depth and hillslope-stream connectivity in a
permafrost watershed. *WATER RESOURCES RESEARCH*, *56*(1).
https://doi.org/10.1029/2019WR025828



Connolly, C. T., Cardenas, M. B., Burkart, G. A., Spencer, R. G. M., & McClelland, J. W.
(2020). Groundwater as a major source of dissolved organic matter to Arctic coastal waters.
*NATURE COMMUNICATIONS*, *11*(1). https://doi.org/10.1038/s41467-020-15250-8

Cory, R. M., Crump, B. C., Dobkowski, J. A., & Kling, G. W. (2013). Surface exposure to
sunlight stimulates $CO_2$ release from permafrost soil carbon in the Arctic. *Proceedings of*
*the National Academy of Sciences*, *110*(9), 3429–3434.
https://doi.org/10.1073/pnas.1214104110

Deshpande, B. N., Crevecoeur, S., Matveev, A., & Vincent, W. F. (2016). Bacterial production
in subarctic peatland lakes enriched by thawing permafrost. *BIOGEOSCIENCES*, *13*(15),
4411–4427. https://doi.org/10.5194/bg-13-4411-2016

Douglas, T. A., Fortier, D., Shur, Y. L., Kanevskiy, M. Z., Guo, L., Cai, Y., & Bray, M. T.
(2011). Biogeochemical and Geocryological Characteristics of Wedge and Thermokarst-
Cave Ice in the CRREL Permafrost Tunnel, Alaska. *PERMAFROST AND PERIGLACIAL*
*PROCESSES*, *22*(2), 120–128. https://doi.org/10.1002/ppp.709

Drake, T. W., Wickland, K. P., Spencer, R. G. M., McKnight, D. M., & Striegl, R. G. (2015).
Ancient low-molecular-weight organic acids in permafrost fuel rapid carbon dioxide
production upon thaw. *PROCEEDINGS OF THE NATIONAL ACADEMY OF SCIENCES*
*OF THE UNITED STATES OF AMERICA*, *112*(45), 13946–13951.
https://doi.org/10.1073/pnas.1511705112

Dutta, K., Schuur, E. A. G., Neff, J. C., & Zimov, S. A. (2006). Potential carbon release from
permafrost soils of Northeastern Siberia. *GLOBAL CHANGE BIOLOGY*, *12*(12), 2336–
2351. https://doi.org/10.1111/j.1365-2486.2006.01259.x

Edwards, K. A., & Jefferies, R. L. (2013). Inter-annual and seasonal dynamics of soil microbial
biomass and nutrients in wet and dry low-Arctic sedge meadows. *Soil Biology and*
*Biochemistry*, *57*, 83–90. https://doi.org/https://doi.org/10.1016/j.soilbio.2012.07.018

Edwards, K. A., McCulloch, J., Kershaw], G. [Peter, & Jefferies, R. L. (2006). Soil microbial
and nutrient dynamics in a wet Arctic sedge meadow in late winter and early spring. *Soil*
*Biology and Biochemistry*, *38*(9), 2843–2851.
https://doi.org/https://doi.org/10.1016/j.soilbio.2006.04.042

Ernakovich, J. G., Lynch, L. M., Brewer, P. E., Calderon, F. J., & Wallenstein, M. D. (2017).
Redox and temperature-sensitive changes in microbial communities and soil chemistry
dictate greenhouse gas loss from thawed permafrost. *BIOGEOCHEMISTRY*, *134*(1–2),
183–200. https://doi.org/10.1007/s10533-017-0354-5

Ewing, S. A., Paces, J. B., O'Donnell, J. A., Jorgenson, M. T., Kanevskiy, M. Z., Aiken, G. R.,
… Striegl, R. (2015). Uranium isotopes and dissolved organic carbon in loess permafrost:
Modeling the age of ancient ice. *GEOCHIMICA ET COSMOCHIMICA ACTA*, *152*, 143–
165. https://doi.org/10.1016/j.gca.2014.11.008



Fenger-Nielsen, R., Hollesen, J., Matthiesen, H., Andersen, E. A. S., Westergaard-Nielsen, A., Harmsen, H., … Elberling, B. (2019). Footprints from the past: The influence of past human activities on vegetation and soil across five archaeological sites in Greenland. *Science of the Total Environment*, *654*, 895–905. https://doi.org/10.1016/j.scitotenv.2018.11.018

Fouché, J., Christiansen, C. T., Lafrenière, M. J., Grogan, P., & Lamoureux, S. F. (2020). Canadian permafrost stores large pools of ammonium and optically distinct dissolved organic matter. *Nature Communications*, *11*(1), 4500. https://doi.org/10.1038/s41467-020-18331-w

Fouche, J., Bouchez, C., Keller, C., Allard, M., & Ambrosi, J.-P. (2021). Seasonal cryogenic processes control supra-permafrost pore water chemistry in two contrasting Cryosols. *GEODERMA*, *401*. https://doi.org/10.1016/j.geoderma.2021.115302

Fouché, J., Keller, C., Allard, M., Ambrosi, J. P., Fouche, J., Keller, C., … Ambrosi, J. P. (2014). Increased CO2 fluxes under warming tests and soil solution chemistry in Histic and Turbic Cryosols, Salluit, Nunavik, Canada. *Soil Biology and Biochemistry*, *68*, 185–199. https://doi.org/https://doi.org/10.1016/j.soilbio.2013.10.007

Fritz, M., Opel, T., Tanski, G., Herzschuh, U., Meyer, H., Eulenburg, A., & Lantuit, H. (2015). Dissolved organic carbon (DOC) in Arctic ground ice. *CRYOSPHERE*, *9*(2), 737–752. https://doi.org/10.5194/tc-9-737-2015

Gagné, K. R., Ewers, S. C., Murphy, C. J., Daanen, R., Walter Anthony, K., & Guerard, J. J. (2020). Composition and photo-reactivity of organic matter from permafrost soils and surface waters in interior Alaska. *Environmental Science: Processes and Impacts*, *22*(7), 1525–1539. https://doi.org/10.1039/d0em00097c

Gao, L., Zhou, Z., Reyes V, A., & Guo, L. (2018). Yields and Characterization of Dissolved Organic Matter From Different Aged Soils in Northern Alaska. *JOURNAL OF GEOPHYSICAL RESEARCH-BIOGEOSCIENCES*, *123*(7), 2035–2052. https://doi.org/10.1029/2018JG004408

Herndon, E. M., Mann, B. F., Chowdhury, T. R., Yang, Z., Wullschleger, S. D., Graham, D., … Gu, B. (2015). Pathways of anaerobic organic matter decomposition in tundra soils from Barrow, Alaska. *JOURNAL OF GEOPHYSICAL RESEARCH-BIOGEOSCIENCES*, *120*(11), 2345–2359. https://doi.org/10.1002/2015JG003147

Herndon, E. M., Yang, Z., Bargar, J., Janot, N., Regier, T. Z., Graham, D. E., … Liang, L. (2015). Geochemical drivers of organic matter decomposition in arctic tundra soils. *BIOGEOCHEMISTRY*, *126*(3), 397–414. https://doi.org/10.1007/s10533-015-0165-5

Herndon, E., AlBashaireh, A., Singer, D., Chowdhury], T. [Roy, Gu, B., & Graham, D. (2017). Influence of iron redox cycling on organo-mineral associations in Arctic tundra soil. *Geochimica et Cosmochimica Acta*, *207*, 210–231. https://doi.org/https://doi.org/10.1016/j.gca.2017.02.034





Heslop, J. K., Chandra, S., Sobzcak, W. V, Davydov, S. P., Davydova, A. I., Spektor, V. V, &
Anthony, K. M. W. (2017). Variable respiration rates of incubated permafrost soil extracts
from the Kolyma River lowlands, north-east Siberia. *POLAR RESEARCH*, *36*.
https://doi.org/10.1080/17518369.2017.1305157
Hirst, C., Mauclet, E., Monhonval, A., Tihon, E., Ledman, J., Schuur, E. A. G., & Opfergelt, S.
(2022). Seasonal Changes in Hydrology and Permafrost Degradation Control Mineral
Element-Bound DOC Transport From Permafrost Soils to Streams. *GLOBAL*
*BIOGEOCHEMICAL CYCLES*, *36*(2). https://doi.org/10.1029/2021GB007105
Hodgkins, S. B., Tfaily, M. M., Podgorski, D. C., McCalley, C. K., Saleska, S. R., Crill, P. M.,
… Cooper, W. T. (2016). Elemental composition and optical properties reveal changes in
dissolved organic matter along a permafrost thaw chronosequence in a subarctic peatland.
*Geochimica et Cosmochimica Acta*, *187*, 123–140.
https://doi.org/10.1016/j.gca.2016.05.015
Jilkova, V., Devetter, M., Bryndova, M., Hajek, T., Kotas, P., Lulakova, P., … Macek, P. (2021).
Carbon Sequestration Related to Soil Physical and Chemical Properties in the High Arctic.
*GLOBAL BIOGEOCHEMICAL CYCLES*, *35*(9). https://doi.org/10.1029/2020GB006877
Kane, E. S., Chivers, M. R., Turetsky, M. R., Treat, C. C., Petersen, D. G., Waldrop, M., …
McGuire, A. D. (2013). Response of anaerobic carbon cycling to water table manipulation
in an Alaskan rich fen. *Soil Biology and Biochemistry*, *58*, 50–60.
https://doi.org/https://doi.org/10.1016/j.soilbio.2012.10.032
Kane, E. S., Valentine, D. W., Michaelson, G. J., Fox, J. D., & Ping, C.-L. (2006). Controls over
pathways of carbon efflux from soils along climate and black spruce productivity gradients
in interior Alaska. *Soil Biology and Biochemistry*, *38*(6), 1438–1450.
https://doi.org/https://doi.org/10.1016/j.soilbio.2005.11.004
Kane, E. S., Turetsky, M. R., Harden, J. W., McGuire, A. D., & Waddington, J. M. (2010).
Seasonal ice and hydrologic controls on dissolved organic carbon and nitrogen
concentrations in a boreal-rich fen. *JOURNAL OF GEOPHYSICAL RESEARCH-*
*BIOGEOSCIENCES*, *115*. https://doi.org/10.1029/2010JG001366
Kawahigashi, M., Prokushkin, A., & Sumida, H. (2011). Effect of fire on solute release from
organic horizons under larch forest in Central Siberian permafrost terrain. *Geoderma*,
*166*(1), 171–180. https://doi.org/https://doi.org/10.1016/j.geoderma.2011.07.027
Koch, J. C., Runkel, R. L., Striegl, R., & McKnight, D. M. (2013). Hydrologic controls on the
transport and cycling of carbon and nitrogen in a boreal catchment underlain by continuous
permafrost. *JOURNAL OF GEOPHYSICAL RESEARCH-BIOGEOSCIENCES*, *118*(2),
698–712. https://doi.org/10.1002/jgrg.20058
Lim, A. G., Loiko, S. V, Kuzmina, D. M., Krickov, I. V, Shirokova, L. S., Kulizhsky, S. P., …
Pokrovsky, O. S. (2021). Dispersed ground ice of permafrost peatlands: Potential





unaccounted carbon, nutrient and metal sources. *Chemosphere*, *266*, 128953.
https://doi.org/10.1016/j.chemosphere.2020.128953

Lindborg, T., Rydberg, J., Tröjbom, M., Berglund, S., Johansson, E., Löfgren, A., … Laudon, H.
(2016). Biogeochemical data from terrestrial and aquatic ecosystems in a periglacial
catchment, West Greenland. *Earth System Science Data*, *8*(2), 439–459.
https://doi.org/10.5194/essd-8-439-2016

Littlefair, C. A., & Tank, S. E. (2018). Biodegradability of Thermokarst Carbon in a Till-
Associated, Glacial Margin Landscape: The Case of the Peel Plateau, NWT, Canada.
*JOURNAL OF GEOPHYSICAL RESEARCH-BIOGEOSCIENCES*, *123*(10), 3293–3307.
https://doi.org/10.1029/2018JG004461

Liu, N., Michelsen, A., & Rinnan, R. (2020). Vegetation and soil responses to added carbon and
nutrients remain six years after discontinuation of long-term treatments. *Science of the Total
Environment*, *722*, 137885. https://doi.org/10.1016/j.scitotenv.2020.137885

Loiko, S. V, Pokrovsky, O. S., Raudina, T. V, Lim, A., Kolesnichenko, L. G., Shirokova, L. S.,
… Kirpotin, S. N. (2017). Abrupt permafrost collapse enhances organic carbon, CO2,
nutrient and metal release into surface waters. *Chemical Geology*, *471*, 153–165.
https://doi.org/https://doi.org/10.1016/j.chemgeo.2017.10.002

MacDonald, E. N., Tank, S. E., Kokelj, S. V., Froese, D. G., & Hutchins, R. H. S. (2021).
Permafrost-derived dissolved organic matter composition varies across permafrost end-
members in the western Canadian Arctic. *Environmental Research Letters*, *16*(2).
https://doi.org/10.1088/1748-9326/abd971

Mangal, V., DeGasparro, S., Beresford, D. V, & Guéguen, C. (2020). Linking molecular and
optical properties of dissolved organic matter across a soil-water interface on Akimiski
Island (Nunavut, Canada). *Science of The Total Environment*, *704*, 135415.
https://doi.org/https://doi.org/10.1016/j.scitotenv.2019.135415

Masyagina, O. V, Tokareva, I. V, & Prokushkin, A. S. (2016). Post fire organic matter
biodegradation in permafrost soils: Case study after experimental heating of mineral
horizons. *Science of The Total Environment*, *573*, 1255–1264.
https://doi.org/https://doi.org/10.1016/j.scitotenv.2016.04.195

McFarlane, K. J., Throckmorton, H. M., Heikoop, J. M., Newman, B. D., Hedgpeth, A. L.,
Repasch, M. N., … Wilson, C. J. (2022). Age and chemistry of dissolved organic carbon
reveal enhanced leaching of ancient labile carbon at the permafrost thaw zone.
*BIOGEOSCIENCES*, *19*(4), 1211–1223. https://doi.org/10.5194/bg-19-1211-2022

Mörsdorf, M. A., Baggesen, N. S., Yoccoz, N. G., Michelsen, A., Elberling, B., Ambus, P. L., &
Cooper, E. J. (2019). Deepened winter snow significantly influences the availability and
forms of nitrogen taken up by plants in High Arctic tundra. *Soil Biology and Biochemistry*,
*135*, 222–234. https://doi.org/https://doi.org/10.1016/j.soilbio.2019.05.009



Neff, J. C., & Hooper, D. U. (2002). Vegetation and climate controls on potential CO2, DOC and
DON production in northern latitude soils. *Global Change Biology*, 8(9), 872–884.
https://doi.org/10.1046/j.1365-2486.2002.00517.x
Nielsen, C. S., Michelsen, A., Strobel, B. W., Wulff, K., Banyasz, I., & Elberling, B. (2017).
Correlations between substrate availability, dissolved CH4, and CH4 emissions in an arctic
wetland subject to warming and plant removal. *JOURNAL OF GEOPHYSICAL
RESEARCH-BIOGEOSCIENCES*, 122(3), 645–660. https://doi.org/10.1002/2016JG003511
O'Donnell, J. A., Aiken, G. R., Butler, K. D., Guillemette, F., Podgorski, D. C., & Spencer, R.
G. M. (2016). DOM composition and transformation in boreal forest soils: The effects of
temperature and organic-horizon decomposition state. *JOURNAL OF GEOPHYSICAL
RESEARCH-BIOGEOSCIENCES*, 121(10), 2727–2744.
https://doi.org/10.1002/2016JG003431
O'Donnell, J. A., Turetsky, M. R., Harden, J. W., Manies, K. L., Pruett, L. E., Shetler, G., &
Neff, J. C. (2009). Interactive Effects of Fire, Soil Climate, and Moss on CO2 Fluxes in
Black Spruce Ecosystems of Interior Alaska. *ECOSYSTEMS*, 12(1), 57–72.
https://doi.org/10.1007/s10021-008-9206-4
Oiffer, L., & Siciliano, S. D. (2009). Methyl mercury production and loss in Arctic soil. *Science
of the Total Environment*, 407(5), 1691–1700.
https://doi.org/10.1016/j.scitotenv.2008.10.025
Olefeldt, D., & Roulet, N. T. (2012). Effects of permafrost and hydrology on the composition
and transport of dissolved organic carbon in a subarctic peatland complex. *Journal of
Geophysical Research: Biogeosciences*, 117(1). https://doi.org/10.1029/2011JG001819
Olefeldt, D., & Roulet, N. T. (2014). Permafrost conditions in peatlands regulate magnitude,
timing, and chemical composition of catchment dissolved organic carbon export. *Global
Change Biology*, 20(10), 3122–3136. https://doi.org/10.1111/gcb.12607
Olefeldt, D., Roulet, N. T., Bergeron, O., Crill, P., Bäckstrand, K., & Christensen, T. R. (2012).
Net carbon accumulation of a high-latitude permafrost palsa mire similar to permafrost-free
peatlands. *Geophysical Research Letters*. https://doi.org/10.1029/2011GL050355
Olsrud, M., & Christensen, T. R. (2011). Carbon partitioning in a wet and a semiwet subarctic
mire ecosystem based on in situ 14C pulse-labelling. *Soil Biology and Biochemistry*, 43(2),
231–239. https://doi.org/10.1016/j.soilbio.2010.09.034
Pastor, A., Poblador, S., Skovsholt, L. J., & Riis, T. (2020). Microbial carbon and nitrogen
processes in high-Arctic riparian soils. *PERMAFROST AND PERIGLACIAL PROCESSES*,
31(1), 223–236. https://doi.org/10.1002/ppp.2039
Patzner, M. S., Mueller, C. W., Malusova, M., Baur, M., Nikeleit, V., Scholten, T., … Bryce, C.
(2020). Iron mineral dissolution releases iron and associated organic carbon during



permafrost thaw. *Nature Communications*, *11*(1), 1–11. https://doi.org/10.1038/s41467-020-
20102-6

Patzner, M. S., Logan, M., McKenna, A. M., Young, R. B., Zhou, Z., Joss, H., … Bryce, C.
(2022). Microbial iron cycling during palsa hillslope collapse promotes greenhouse gas
emissions before complete permafrost thaw. *Communications Earth & Environment*, *3*(1),
76. https://doi.org/10.1038/s43247-022-00407-8
Payandi-Rolland, D., Shirokova, L. S., Tesfa, M., Bénézeth, P., Lim, A. G., Kuzmina, D., …
Pokrovsky, O. S. (2020). Dissolved organic matter biodegradation along a hydrological
continuum in permafrost peatlands. *Science of The Total Environment*, *749*, 141463.
https://doi.org/10.1016/J.SCITOTENV.2020.141463
Payandi-Rolland, D., Shirokova, L. S., Labonne, F., Bénézeth, P., & Pokrovsky, O. S. (2021).
Impact of freeze-thaw cycles on organic carbon and metals in waters of
permafrost  peatlands. *Chemosphere*, *279*, 130510.
https://doi.org/10.1016/j.chemosphere.2021.130510
Payandi-Rolland, D., Shirokova, L. S., Nakhle, P., Tesfa, M., Abdou, A., Causserand, C., …
Pokrovsky, O. S. (2020). Aerobic release and biodegradation of dissolved organic matter
from frozen peat: Effects of temperature and heterotrophic bacteria. *CHEMICAL
GEOLOGY*, *536*. https://doi.org/10.1016/j.chemgeo.2019.119448
Petersen, D. G., Blazewicz, S. J., Firestone, M., Herman, D. J., Turetsky, M., & Waldrop, M.
(2012). Abundance of microbial genes associated with nitrogen cycling as indices of
biogeochemical process rates across a vegetation gradient in Alaska. *Environmental
Microbiology*, *14*(4), 993–1008. https://doi.org/10.1111/j.1462-2920.2011.02679.x
Pokrovsky, O. S., Reynolds, B. C., Prokushkin, A. S., Schott, J., & Viers, J. (2013). Silicon
isotope variations in Central Siberian rivers during basalt weathering in permafrost-
dominated larch forests. *Chemical Geology*, *355*, 103–116.
https://doi.org/https://doi.org/10.1016/j.chemgeo.2013.07.016
Pokrovsky, O. S., Schott, J., Kudryavtzev, D. I., & Dupré, B. (2005). Basalt weathering in
Central Siberia under permafrost conditions. *Geochimica et Cosmochimica Acta*, *69*(24),
5659–5680. https://doi.org/10.1016/j.gca.2005.07.018
Pokrovsky, O. S., Manasypov, R. M., Loiko, S. V, & Shirokova, L. S. (2016). Organic and
organo-mineral colloids in discontinuous permafrost zone. *Geochimica et Cosmochimica
Acta*, *188*, 1–20. https://doi.org/https://doi.org/10.1016/j.gca.2016.05.035
Poulin, B. A., Ryan, J. N., Tate, M. T., Krabbenhoft, D. P., Hines, M. E., Barkay, T., … Aiken,
G. R. (2019). Geochemical Factors Controlling Dissolved Elemental Mercury and
Methylmercury Formation in Alaskan Wetlands of Varying Trophic Status. *Environmental
Science and Technology*, *53*(11), 6203–6213. https://doi.org/10.1021/acs.est.8b06041





Prokushkin, A. S., Gavrilenko, I. V, Abaimov, A. P., Prokushkin, S. G., & Samusenko, A. V.
(2006). Dissolved organic carbon in upland forested watersheds underlain by continuous
permafrost in Central Siberia. *Mitigation and Adaptation Strategies for Global Change*,
*11*(1), 223–240. https://doi.org/10.1007/s11027-006-1022-6

Prokushkin, A. S., Gleixner, G., McDowell, W. H., Ruehlow, S., & Schulze, E.-D. (2007).
Source- and substrate-specific export of dissolved organic matter from permafrost-
dominated forested watershed in central Siberia. *GLOBAL BIOGEOCHEMICAL CYCLES*,
*21*(4). https://doi.org/10.1029/2007GB002938

Prokushkin, A. S., Kajimoto, T., Prokushkin, S. G., McDowell, W. H., Abaimov, A. P., &
Matsuura, Y. (2005). Climatic factors influencing fluxes of dissolved organic carbon from
the forest floor in a continuous-permafrost Siberian watershed. *CANADIAN JOURNAL OF
FOREST RESEARCH*, *35*(9), 2130–2140. https://doi.org/10.1139/X05-150

Rasmussen, L. H., Michelsen, A., Ladegaard-Pedersen, P., Nielsen, C. S., & Elberling, B.
(2020). Arctic soil water chemistry in dry and wet tundra subject to snow addition, summer
warming and herbivory simulation. *Soil Biology and Biochemistry*, *141*, 107676.
https://doi.org/https://doi.org/10.1016/j.soilbio.2019.107676

Raudina, T. V, Loiko, S. V, Lim, A., Manasypov, R. M., Shirokova, L. S., Istigechev, G. I., …
Pokrovsky, O. S. (2018). Permafrost thaw and climate warming may decrease the CO2,
carbon, and metal concentration in peat soil waters of the Western Siberia Lowland. *Science
of The Total Environment*, *634*, 1004–1023.
https://doi.org/https://doi.org/10.1016/j.scitotenv.2018.04.059

Raudina, T. V, Loiko, S. V, Lim, A. G., Krickov, I. V, Shirokova, L. S., Istigechev, G. I., …
Pokrovsky, O. S. (2017). Dissolved organic carbon and major and trace elements in peat
porewater of sporadic, discontinuous, and continuous permafrost zones of western Siberia.
*BIOGEOSCIENCES*, *14*(14), 3561–3584. https://doi.org/10.5194/bg-14-3561-2017

Ro, H.-M., Ji, Y., & Lee, B. (2018). Interactive effect of soil moisture and temperature regimes
on the dynamics of soil organic carbon decomposition in a subarctic tundra soil.
*GEOSCIENCES JOURNAL*, *22*(1), 121–130. https://doi.org/10.1007/s12303-017-0052-2

Roehm, C. L., Giesler, R., & Karlsson, J. (2009). Bioavailability of terrestrial organic carbon to
lake bacteria: The case of a degrading subarctic permafrost mire complex. *JOURNAL OF
GEOPHYSICAL RESEARCH-BIOGEOSCIENCES*, *114*.
https://doi.org/10.1029/2008JG000863

Rogers, J. A., Galy, V., Kellerman, A. M., Chanton, J. P., Zimov, N., & Spencer, R. G. M.
(2021). Limited Presence of Permafrost Dissolved Organic Matter in the Kolyma River,
Siberia Revealed by Ramped Oxidation. *JOURNAL OF GEOPHYSICAL RESEARCH-
BIOGEOSCIENCES*, *126*(7). https://doi.org/10.1029/2020JG005977





Roth, V.-N., Dittmar, T., Gaupp, R., & Gleixner, G. (2013). Latitude and pH driven trends in the molecular composition of DOM across a north south transect along the Yenisei River. *Geochimica et Cosmochimica Acta*, *123*, 93–105. https://doi.org/https://doi.org/10.1016/j.gca.2013.09.002

Schostag, M., Stibal, M., Jacobsen, C. S., Baelum, J., Tas, N., Elberling, B., … Prieme, A. (2015). Distinct summer and winter bacterial communities in the active layer of Svalbard permafrost revealed by DNA- and RNA-based analyses. *FRONTIERS IN MICROBIOLOGY*, *6*. https://doi.org/10.3389/fmicb.2015.00399

Shakil, S., Tank, S. E., Kokelj, S. V., Vonk, J. E., & Zolkos, S. (2020). Particulate dominance of organic carbon mobilization from thaw slumps on the Peel Plateau, NT: Quantification and implications for stream systems and permafrost carbon release. *Environmental Research Letters*, *15*(11). https://doi.org/10.1088/1748-9326/abac36

Shatilla, N. J., & Carey, S. K. (2019). Assessing inter-annual and seasonal patterns of DOC and DOM quality across a complex alpine watershed underlain by discontinuous permafrost in Yukon, Canada. *Hydrology and Earth System Sciences*, *23*(9), 3571–3591. https://doi.org/10.5194/hess-23-3571-2019

Shirokova, L. S., Pokrovsky, O. S., Kirpotin, S. N., Desmukh, C., Pokrovsky, B. G., Audry, S., & Viers, J. (2013). Biogeochemistry of organic carbon, CO2, CH4, and trace elements in thermokarst water bodies in discontinuous permafrost zones of Western Siberia. *BIOGEOCHEMISTRY*, *113*(1–3), 573–593. https://doi.org/10.1007/s10533-012-9790-4

Shirokova, L. S., Bredoire, R., Rols, J.-L. L., & Pokrovsky, O. S. (2017). Moss and Peat Leachate Degradability by Heterotrophic Bacteria: The Fate of Organic Carbon and Trace Metals. *Geomicrobiology Journal*, *34*(8), 641–655. https://doi.org/10.1080/01490451.2015.1111470

Shirokova, L. S., Chupakov, A. V, Zabelina, S. A., Neverova, N. V, Payandi-Rolland, D., Causserand, C., … Pokrovsky, O. S. (2019). Humic surface waters of frozen peat bogs (permafrost zone) are highly resistant to bio- and photodegradation. *BIOGEOSCIENCES*, *16*(12), 2511–2526. https://doi.org/10.5194/bg-16-2511-2019

Shirokova, L. S., Labouret, J., Gurge, M., Gerard, E., Ivanova, I. S., Zabelina, S. A., & Pokrovsky, O. S. (2017). Impact of Cyanobacterial Associate and Heterotrophic Bacteria on Dissolved Organic Carbon and Metal in Moss and Peat Leachate: Application to Permafrost Thaw in Aquatic Environments. *AQUATIC GEOCHEMISTRY*, *23*(5–6), 331–358. https://doi.org/10.1007/s10498-017-9325-7

Sistla, S. A., Schaeffer, S., & Schimel, J. P. (2019). Plant community regulates decomposer response to freezing more strongly than the rate or extent of the freezing regime. *ECOSPHERE*, *10*(2). https://doi.org/10.1002/ecs2.2608



Speetjens, N. J., Tanski, G., Martin, V., Wagner, J., Richter, A., Hugelius, G., … Vonk, J. E.
(2022). Dissolved organic matter characterization in soils and streams in a small coastal
low-arctic catchment. *Biogeosciences*, *19*(July), 3073–3097. Retrieved from
https://doi.org/10.5194/bg-19-3073-2022
Stutter, M. I., & Billett, M. F. (2003). Biogeochemical controls on streamwater and soil solution
chemistry in a High Arctic environment. *Geoderma*, *113*(1), 127–146.
https://doi.org/https://doi.org/10.1016/S0016-7061(02)00335-X
Takano, S., Yamashita, Y., Tei, S., Liang, M., Shingubara, R., Morozumi, T., … Sugimoto, A.
(2021). Stable Water Isotope Assessment of Tundra Wetland Hydrology as a Potential
Source of Arctic Riverine Dissolved Organic Carbon in the Indigirka River Lowland,
Northeastern Siberia. *Frontiers in Earth Science*, *9*.
https://doi.org/10.3389/feart.2021.699365
Tanski, G., Couture, N., Lantuit, H., Eulenburg, A., & Fritz, M. (2016). Eroding permafrost
coasts release low amounts of dissolved organic carbon (DOC) from ground ice into the
nearshore zone of the Arctic Ocean. *Global Biogeochemical Cycles*, *30*(7), 1054–1068.
https://doi.org/10.1002/ 2015GB005337
Tanski, G., Lantuit, H., Ruttor, S., Knoblauch, C., Radosavljevic, B., Strauss, J., … Fritz, M.
(2017). Transformation of terrestrial organic matter along thermokarst-affected permafrost
coasts in the Arctic. *Science of the Total Environment*, *581–582*, 434–447.
https://doi.org/10.1016/j.scitotenv.2016.12.152
Textor, S. R., Wickland, K. P., Podgorski, D. C., Johnston, S. E., & Spencer, R. G. M. (2019).
Dissolved Organic Carbon Turnover in Permafrost-Influenced Watersheds of Interior
Alaska: Molecular Insights and the Priming Effect. *FRONTIERS IN EARTH SCIENCE*, *7*.
https://doi.org/10.3389/feart.2019.00275
Thompson, M. S., Giesler, R., Karlsson, J., & Klaminder, J. (2015). Size and characteristics of
the DOC pool in near-surface subarctic mire permafrost as a potential source for nearby
freshwaters. *Arctic, Antarctic, and Alpine Research*, *47*(1), 49–58.
https://doi.org/10.1657/AAAR0014-010
Treat, C. C., Wollheim, W. M., Varner, R. K., & Bowden, W. B. (2016). Longer thaw seasons
increase nitrogen availability for leaching during fall in tundra soils. *ENVIRONMENTAL
RESEARCH LETTERS*, *11*(6). https://doi.org/10.1088/1748-9326/11/6/064013
Trusiak, A., Treibergs, L. A., Kling, G. W., & Cory, R. M. (2018). The role of iron and reactive
oxygen species in the production of $CO_2$ in arctic soil waters. *GEOCHIMICA ET
COSMOCHIMICA ACTA*, *224*, 80–95. https://doi.org/10.1016/j.gca.2017.12.022
Voigt, C., Lamprecht, R. E., Marushchak, M. E., Lind, S. E., Novakovskiy, A., Aurela, M., …
Biasi, C. (2017). Warming of subarctic tundra increases emissions of all three important





greenhouse gases – carbon dioxide, methane, and nitrous oxide. *Global Change Biology*,
*23*(8), 3121–3138. https://doi.org/10.1111/gcb.13563
Voigt, C., Marushchak, M. E., Mastepanov, M., Lamprecht, R. E., Christensen, T. R.,
Dorodnikov, M., … Biasi, C. (2019). Ecosystem carbon response of an Arctic peatland to
simulated permafrost thaw. *Global Change Biology*, *25*(5), 1746–1764.
https://doi.org/10.1111/gcb.14574
Voigt, C., Marushchak, M. E., Lamprecht, R. E., Jackowicz-Korczyński, M., Lindgren, A.,
Mastepanov, M., … Biasi, C. (2017). Increased nitrous oxide emissions from Arctic
peatlands after permafrost thaw. *Proceedings of the National Academy of Sciences of the
United States of America*, *114*(24), 6238–6243. Retrieved from
https://www.jstor.org/stable/26484198
Vonk, J. E., Mann, P. J., Dowdy, K. L., Davydova, A., Davydov, S. P., Zimov, N., … Holmes,
R. M. (2013). Dissolved organic carbon loss from Yedoma permafrost amplified by ice
wedge thaw. *ENVIRONMENTAL RESEARCH LETTERS*, *8*(3).
https://doi.org/10.1088/1748-9326/8/3/035023
Vonk, J. E., Mann, P. J., Davydov, S., Davydova, A., Spencer, R. G. M., Schade, J., … Holmes,
R. M. (2013). High biolability of ancient permafrost carbon upon thaw. *GEOPHYSICAL
RESEARCH LETTERS*, *40*(11), 2689–2693. https://doi.org/10.1002/grl.50348
Waldrop, M. P., Harden, J. W., Turetsky, M. R., Petersen, D. G., McGuire, A. D., Briones, M. J.
I., … Pruett, L. E. (2012). Bacterial and enchytraeid abundance accelerate soil carbon
turnover along a lowland vegetation gradient in interior Alaska. *Soil Biology and
Biochemistry*, *50*, 188–198. https://doi.org/https://doi.org/10.1016/j.soilbio.2012.02.032
Waldrop, M. P., & Harden, J. W. (2008). Interactive effects of wildfire and permafrost on
microbial communities and soil processes in an Alaskan black spruce forest. *GLOBAL
CHANGE BIOLOGY*, *14*(11), 2591–2602. https://doi.org/10.1111/j.1365-
2486.2008.01661.x
Ward, C. P., & Cory, R. M. (2015). Chemical composition of dissolved organic matter draining
permafrost soils. *Geochimica et Cosmochimica Acta*, *167*, 63–79.
https://doi.org/https://doi.org/10.1016/j.gca.2015.07.001
Ward, C. P., Nalven, S. G., Crump, B. C., Kling, G. W., & Cory, R. M. (2017). Photochemical
alteration of organic carbon draining permafrost soils shifts microbial metabolic pathways
and stimulates respiration. *NATURE COMMUNICATIONS*, *8*.
https://doi.org/10.1038/s41467-017-00759-2
Whittinghill, K. A., Finlay, J. C., & Hobbie, S. E. (2014). Bioavailability of dissolved organic
carbon across a hillslope chronosequence in the Kuparuk River region, Alaska. *Soil Biology
and Biochemistry*, *79*, 25–33. https://doi.org/https://doi.org/10.1016/j.soilbio.2014.08.020



Wickland, K. P., Neff, J. C., & Aiken, G. R. (2007). Dissolved organic carbon in Alaskan boreal
forest: Sources, chemical characteristics, and biodegradability. *ECOSYSTEMS*, *10*(8),
1323–1340. https://doi.org/10.1007/s10021-007-9101-4

Wickland, K. P., Waldrop, M. P., Aiken, G. R., Koch, J. C., Jorgenson, Mt., & Striegl, R. G.
(2018). Dissolved organic carbon and nitrogen release from boreal Holocene permafrost and
seasonally frozen soils of Alaska. *ENVIRONMENTAL RESEARCH LETTERS*, *13*(6).
https://doi.org/10.1088/1748-9326/aac4ad

Yun, J., Jung, J. Y., Kwon, M. J., Seo, J., Nam, S., Lee, Y. K., & Kang, H. (2022). Temporal
Variations Rather than Long-Term Warming Control Extracellular Enzyme Activities and
Microbial Community Structures in the High Arctic Soil. *MICROBIAL ECOLOGY*, *84*(1),
168–181. https://doi.org/10.1007/s00248-021-01859-9

Zolkos, S., & Tank, S. E. (2019). *Permafrost geochemistry and retrogressive thaw slump
morphology (Peel Plateau, Canada), v. 1.0 (2017-2017)*. https://doi.org/10.5885/45573XD-
28DD57D553F14BF0
