# Peer review of "Review article: A systematic review of terrestrial dissolved organic carbon in northern permafrost"

_The Cryosphere, 2023_

## Author Response (AR1)

**Reviewer 1**

*General comments: My recommendation for this manuscript is to resubmit after major revisions. The results are of significant interest to The Cryosphere, but the current manuscript requires considerable modifications before considering immediate publication. Specifically, the introduction and discussion sections are currently written in a structure that is difficult for readers to understand. Additionally, numerous English grammar issues have been identified in the sentences. Focused improvements in these areas could greatly enhance the quality of the manuscript.*

We thank the reviewer for taking the time to provide their review of our manuscript. Several reviewers have highlighted the need to improve clarity and narrative structure to the manuscript, particularly the introduction. We have restructured the introduction to reflect these comments. We have also restructured elements to the discussion. We thank the reviewer for their helpful comments in guiding this restructuring. Below we have addressed their detailed comments and concerns.

*Detailed Comments*

- *[L55-56] I think you need references to back this up.*

The reference Olefeldt et al., (2016) has been added here

Olefeldt, D., Goswami, S., Grosse, G., Hayes, D., Hugelius, G., Kuhry, P., … Turetsky, M. R. (2016). Circumpolar distribution and carbon storage of thermokarst landscapes. Nature Communications, 7, 13043. https://doi.org/10.1038/ncomms13043

- *[L67-71] The sentence seems overly lengthy. It may be beneficial to break it down into shorter sentences for clarity.*

This has been split into two sentences

- *[L71-73] The sentence is grammatically incomplete and requires revision for clarity.*

This sentence has ben restructured and worded.

- *[L74-76] I'm uncertain whether the current sentence is necessary as the opening line of this paragraph, considering the section primarily discusses the lateral movement of DOC.*

This has been addressed in the restructuring of the introduction

- *[L79-81] The sentence seems more fitting for the previous paragraph. I'm unsure if it's essential for the narrative progression of the current section.*

This has been addressed in the restructuring of the introduction

- *[L74-99] The critical concepts in the second paragraph of the introduction might be 'the lateral transport of DOC,' 'permafrost landscape dynamics,' and their impacts on the 'Arctic freshwater ecosystems.' However, due to a lack of connectivity between these concepts in the paragraph, it is challenging to grasp the narrative and logic the author intends to convey. I think addressing this issue to clarify how these concepts interrelate is necessary.*

We agree and thank the reviewer for highlighting this. This has been addressed in the restructuring of the introduction

- *[L100-117] This paragraph should clearly articulate how the research was conducted based on the research status and limitations identified in the preceding sections, including any particular methodology used, the objectives of the study, and the hypothesis. However, as it stands, this information is difficult to discern.*

We have edited this paragraph (L136-156) in line with the restructuring of the introduction above. Now, we set out our aim/objective and hypotheses, which are in line with the previous introductory text. We briefly describe what the database contains but also highlight its importance. We have chosen not to explain methods used in too much detail as this comes in the proceeding paragraphs.

- *[L112-114] This sentence seems misplaced in the introduction and appears more appropriate for the methods section.*

This has been addressed in the restructuring of the introduction

- *[L122] Explaining the research question design in connection with the hypothesis presented in the introduction would be beneficial.*

This has been addressed in the restructuring of the introduction

- *[L598-614] Finding a significant distinction between this paragraph and the results section is challenging. I recommend integrating this paragraph into the results section. Otherwise, it might be better to use this section to outline the overall direction of the discussion. For example, you could describe how the results will be used to substantiate each hypothesis.*

The aim of this paragraph is to begin to broaden the story again after the results and before we move into the discussion. The results section is rather dense and we are using this paragraph to clearly highlight the main results we would like the reader to have fresh in their mind and easily accessible. The reviewers main concern in this particular comment seems to come down to a difference in writing style. As we can find no fundamental disagreement from the reviewer with respect to the content, we are choosing the leave the paragraph as it is for clarity.

- *[L615-629] This section aims to prove the hypothesis that soils rich in organic matter have higher DOC concentrations. It explains this by linking to the differences in soil classification. However, a more detailed explanation of the unique characteristics of each soil class that could influence DOC content variations is needed.*

We have amended the text and added references to highlight that DOC is derived largely from the leaching of the soil organic carbon pool (L78-80; L86-92; L261-265). Our argument is that a larger soil organic pool will likely lead to a larger DOC pool. The Histel and Histosol soils have a larger soil organic carbon pool as they are organic soils, whereas the others are mineral soils.

- *[L630-651] The content in this section is not being communicated clearly. It should be more concise and put the topic sentence in the head of the paragraph. Moreover, it might be better to establish discussion points that connect to the hypothesis and develop the logic accordingly.*

In the restructuring of the discussion we have removed this section. Where appropriate, we have added text from this section to the new section 4.2 Variation in DOC across ecosystems.

- *[L652-733] For clarity in this section, I recommend organizing the content in the following order: findings/claims based on the results of this study, supporting evidence from previous research, and the implications of this particular discussion point.*

This section has undergone restructuring. We have removed the opening paragraph and incorporated this text into either lower paragraphs or the introduction, where appropriate. We now have paragraphs discussing permafrost bogs and wetland, coastal tundra, and Yedoma and upland tundra sites.

- *There are additional detailed revision and improvement requests, but I will consider them after the major manuscript revisions have been addressed.*

We look forward to reading and addressing them.

**Reviewer 2**

*General comments: I would like to thank the authors for this impressive effort in synthesizing and collating this nice database. The permafrost community needs a better understanding of the magnitude of lateral organic carbon export in the Arctic region and its significance for the climate feedback of thawing permafrost. In that regard, this study is timely and relevant.*

*However, I have a few major concerns below before I can properly assess the quality of the manuscript.*

We thank the reviewer for recognizing the important contribution of this work to the permafrost community and for providing this thorough and constructive criticism of the manuscript. We greatly appreciate their efforts in providing valuable suggestions on how to improve the manuscript and in highlighting the positive aspects of our work. Below, please find our response to each comment and suggestion provided.

*My main concern is regarding the validity of the interpretation of DOC variability across ecosystem types because the authors, despite mentioning it, ignore the important effect of the various soil porewater extraction methods on the results. It is known that different extraction methods yield different soil porewater recovery rates with large effect on solute concentrations. These effects can't be ignored. Otherwise, the authors would be overinterpreting the data. I suggest some possible additional analyses below to potentially sort this out.*

The issue of not accounting for variability in DOC concentrations due to extraction and analysis methods has been highlighted by several reviewers. We agree with these concerns, and have addressed this issue by adding text to section "2.4 Database analysis" to highlight how differences in methods effect DOC concentrations, changing some text in section 3.1 Database generation, and adding a new section to the results (3.4 Effect of extraction and analysis methods on DOC concentrations).

In short, we have removed any text from "Section 3.1 Database generation" regarding statistical analysis of extraction methods on DOC concentrations. Now, we just outline the proportion of the DOC concentrations contained in the database that are attributed to each method. We have then added a new section ("Section 3.4 Effect of extraction and analysis methods on DOC concentrations") to explicitly address how we consider variability introduced via extraction method. In this section, along with corresponding tables in the supplementary (Table S2-S7) we show that the absolute trends observed in DOC concentrations across our study regions and ecosystems (as outlined in Section 3.2 and 3.3) are also observed when we account for the differing extraction methods. Furthermore, the large variability observed in DOC concentrations across study region and ecosystems is not reduced when accounting for extraction method. Thus, while

extraction methods do have an impact on DOC concentrations, they end up having a similar effect across all study regions and ecosystems. In the final section of the manuscript ("4.5 Future considerations for study design"), we highlight that there is a need for a standardized methodology for extracting, assessing and determining DOC concentrations.

*In addition, the introduction is lacking a clear narrative which brings more confusion to the reader throughout the manuscript. While the authors try to explain why it is important to consider DOC and its export during permafrost thaw, they don't mention previous efforts in reporting soil DOC and current understanding of the variability in soil DOC, current state of the art etc. The aim of the paper only comes at l. 327, after mentioning the non-negligible effect of extraction methods, this is too late and should be included in the introduction. And again, the "assessment of methods" is a prerequisite to being able to "assess the concentration and mobilization of DOC in terrestrial permafrost ecosystems".*

We thank the reviewer for highlighting these issues with the paper. We have restructured the introduction and added the aims earlier in the introduction. More details on how we addressed these concerns are below.

*In general, I also wonder whether this study falls under the "Review" or "Original research paper" type. I am missing a critical analysis of the available research and practices on terrestrial DOC in permafrost to date for it to be a proper review paper. The review is only partial and includes only the data collation aspect. On the other hand, there quite a lot of data analysis which could well fit under an original paper. This has implications for the title of the paper, where currently "A systematic review of…" should be removed from the title.*

This is an interesting comment and one that has led to some reflection upon the paper. The "systematic review" element of the title refers only to the systematic approach used to compile and assess the current literature and build the database (Methods Section 2.1 – 2.3). The analysis of the database is an attempt to, highlight broad patterns in DOC concentrations across the northern permafrost region and the terrestrial ecosystems found there within. This approach was used to reduce any potential biases in identifying relevant studies pertaining to DOC concentrations in northern permafrost regions. We wanted to be as exhaustive and inclusive as possible in the process of compiling the

database. By doing so, it provides the permafrost community with a sound and useful quantitative assessment of DOC concentrations currently, and highlights topics in need of future research (L143-145). The meta-analysis of data generated via this systematic approach is limited in this manuscript to our use of response ratios when assessing DOC concentrations following thermokarst (L344-358; Section 3.6 L734-750; Figure 7). We do not say that this paper is a meta-analysis of DOC concentrations in northern permafrost, nor do we state meta-analysis as one of our objectives with the manuscript. We agree with the reviewer that the previous title may lead to some confusion over what the paper is attempting to do, thus we have removed systematic review from the title of the paper and the new title is "Terrestrial dissolved organic carbon in northern permafrost".

Regarding the type of paper under which this manuscript falls under, we believe our manuscript falls under the review article category when considering the guidelines provided by The Cryosphere ([https://www.the-cryosphere.net/about/manuscript_types.html](https://www.the-cryosphere.net/about/manuscript_types.html)). Under these guidelines a review article is one where the authors "summarize the status of knowledge and outline future directions of research within the journal scope." We provide a (a) summary of DOC concentrations in northern terrestrial permafrost through our descriptive analysis (Figure 2, 3, and 4); (b) attempt to describe the variability observed in these concentrations (Figure 5); and (c) highlight the limited available data describing DOC loss (via export or mineralization) in northern permafrost (Section 3.6; Figure 6, 7) and outline areas for future research (Section 4.5 Future considerations for study design). Hence, we prefer to keep the manuscript as a "Review", unless the editor thinks it should be re-classified.

*Detailed Comments*

1. *The novelty of the study is not just limited to what is stated l. 108-112, but to be able to better identify this novelty, a properly constructed introduction is required with reference to previous studies, e.g., Guo et al. 2020, Langeveldt et al 2020.*

   *Previous syntheses on Permafrost carbon cycling mentioned DOC export as an important fluxes but it is still overlooked e.g.,:*

*https://www.annualreviews.org/doi/10.1146/annurev-environ-012220-011847*

*https://link.springer.com/article/10.1007/s13280-016-0872-8*

We have restructured our introduction to include more explanation of previous efforts to synthesize the permafrost carbon feedback and have highlighted how DOC is often mentioned as being potentialy relevant, but has been overlooked and has not been quantified, as a  component of this feedback. This we think this provides a sound base for claiming the novelty of this study.

2. *L. 94-99, it is true that the different sources of DOC will impact its biodegradability, however, the current dataset doesn't allow to distinguish the different sources, this is, I believe, outside the scope of this study. First step is to describe and understand factors driving the natural variability in DOC, whether the current set of data is able to represent real variability or whether methodological artefacts are the main driver behind DOC variability.*

We have removed this sentence.

3. *L. 114-117 where do the hypotheses come from? This is also where the introduction is lacking a clear narrative, the hypotheses are not introduced based on previous research findings, not anchored in current understanding. The reader needs some background.*

We have restructured the introduction in an attempt to address this issue. Now the introduction builds to these hypotheses.

4. *L. 122-127 a similar point to the previous point, why asking those questions? The introduction should be built around the various hypotheses and research questions so that the questions do not appear out of nowhere here.*

As above, the restructuring was aimed at addressing this issue.

5. *L. 215-226 and Fig. 3. In contrast to the other categorical variables displayed in Fig. 2 and introduced earlier, (i.e., ecoregions, soil classes, permafrost zones, thermal horizons), the ecosystem types are never introduced properly despite their central role in the data analysis later on, also in the PLS. To avoid future confusion and improve clarity, the*

*different ecosystem types should be described, or the authors should point to relevant references where the reader can learn about the different ecosytems. Reference to Table S1 would be useful here and please add some description and relevant references for each ecosystem type.*

We have added information on each ecosystem type to this section including references from L275-289.

6. *L. 210-214 I count only 10 extraction methods while l. 313 and onwards, 11 extraction methods are mentioned. Here again, some background information on the extraction methods is required, such as expected recovery, passive versus active, destructive or not, references to studies describing the methods. Include a table with references.*

The missing extraction method (dry leaching) has been added to the text here.

7. *Since the extraction methods can yield very different porewater recovery, e.g.,*

   *https://www.sciencedirect.com/science/article/pii/S0883292706002289*

   *https://www.sciencedirect.com/science/article/pii/S0038071705003111*

   *https://bsssjournals.onlinelibrary.wiley.com/doi/10.1111/j.1365-2389.1996.tb01413.x*

   *the effect of the extraction method cannot be ignored before analyzing for differences across ecosystem types etc. L. 312-325 is a good starting point and highlight the importance of "water-based versus solid (soil)" extraction methods which showed the highest ANOVA F score, among all reported ANOVA results. To me, this shows that extraction methods are more important than ecosystem types for explaining DOC variability. Am I wrong? This section must show whether the extraction methods are responsible for a larger share of the observed variability in DOC than other categorical variables, such as ecosystem types. If this is the case, then the remaining of the analysis must be adapted and e.g., performed by "extraction method".*

Please see above where we have addressed this concern of the reviewer. In short, we have added extra details to our analysis section (section 2.4) discussing how we assessed

differences in DOC concentrations from each method and added a new section to the manuscript (Section 3.4 Effect of extraction and analysis methods on DOC concentrations) that addresses this issue.

> *What happens to the findings in section 3.2 and 3.3 if the analysis is performed by "extraction method".*

When exploring the effects of extraction and analysis methods on DOC concentrations we find the same patterns as observed in section 3.2 and 3.3. These are shown in the supplementary in Table S2-7.

> 8. *L. 831-837 This information is coming way too late, this should start your results section so that you can probably rule out some of the issues I raise above. Need to be expanded and to describe how the extraction methods would impact observed DOC variability.*

We agree that this information comes too late. We have left this section (L1029-1035) as is to finish the manuscript on a point where we use this information to suggest future best practices. To address this concern of the reviewer, we have added a paragraph with more information regarding the breakdown of methods including the proportion of the dataset using each method to determine DOC concentrations in section 3.1 (L400-413).

> 9. *L. 310 to 312, the median of the DOC grouped by filter size are quite different and to declare "We consider the effects of filter size to be minor" just after that, is a bit odd. The large variation in the medians here might in fact reflect some differences caused by extraction methods. Same comment for carbon measurement methods further below, if you cannot control for the extraction method, the values you report don't have a proper sense. What if you group the data by extraction method?*

We agree that this statement must be justified and have addressed this in the new Section "3.4 Effect of extraction and analysis methods on DOC concentrations". Here (L562-575) we address the impact filter size on DOC concentrations. While there are differences between filter size, we conclude that for the overall comparison of DOC across our study, filter size is unimportant because it has a similar effect across all regions.

*10. The reason to use the PLS is somewhat confused. What is the actual aim of the PLS? L. 241-243 it is mentioned that PLS is used to "assess the performance of continuous and categorical variables in predicting DOC concentrations". But later, l. 464-480, the results of the PLS are used to better understand the variability in DOC, make links between processes and DOC variability. If the aim was to predict DOC, the authors should only use variables that are easily available at larger spatial scales, I wonder what the value is of predicting DOC with TDN and C:N ratio. In fact, the point is not to predict, but to investigate relationships to improve our process understanding.*

To address this concern, we have removed language that suggests we are attempting to predict DOC concentrations. Rather, we are trying to use the relationship between DOC and the categorical and continuous variables within our dataset to help explain some of the variability we observed within the DOC concentrations. We have made changes in language used to describe why this analysis was performed (L324-343). Also we have made the following changes;

L619 – changed "predict" to "determine the drivers"

L628 – changed "effect on" to "relationship with"

*In general, this section 2.4 is somewhat too dense and is a mixture of various statistical analyses whose aims are not completely clear. The authors should divide this section in sub-sections where, for each analysis, the objective is clearly stated. Here I am missing a relevant method to distinguish between the effect of DOC extraction method versus ecosystem types and other environmental variables in the DOC variability.*

We have restructured section 2.4 where the first paragraph discusses analysis pertaining to difference in DOC concentrations across regions and ecosystems, the second is assessing differences amongst extraction and analysis methods, the third discusses the PLS regression, and the final paragraph discusses the use of the response ratio.

*11. Fig. 5 How were the predictor variables selected? Figure S. 5 shows a complete PLS with all variables considered but is not referenced in the main text. The authors mentioned the "Variable Importance in Projections (VIP)" l. 244 but those values are not reported. Please add a table (eventually in the supplementary) where all variables have this "VIP"*

*score and a description of how the variables were selected, was it only based on this VIP*
*>1?*

PLS predictor variables were selected based on VIP score and the systematic removal of variables with different VIP scores. We have updated this text (L324-343) to better highlight how predictors and the most parsimonious model was selected. We have also added in two references highlighting and explaining the approach we took. The PLS shown in Figure S5 is included in the supplementary because we think it may be of interest for readers to assess the importance of different continuous and categorical variables are impacting DOC concentrations.

> *L. 248-249, why did the authors decide to split the ecosystem classes in 2? The answer might be along the lines 754-780 but it comes too late and it not linked.*

This text has been updated (L33-343) to highlight our reasoning to subdivide the data.

> 12. *L. 735-753 this section seems weak, where the authors end up saying that they cannot conclude based on too little data.*

Correct, this a negative outcome of this study. However, negative results such as this can provide important knowledge to the permafrost community. One of our objectives was to be able to assess the impact of thermokarst formation on DOC mobilization. However, using our systematic approach we were unable to gather enough data. Here we are highlighting this limitation in both our approach, but also the gap in currently available literature and our fundamental understanding. We have added text on L929-934 and L937-939 to make this point clearer.

> 13. *BDOC data. L. 303-304 the authors mentioned that they consider later the smaller sample size during result interpretation, but this is not completely clear. Reading through l. 530-541, which is based on all BDOC data, including 3-days to 90 days incubation data, VERSUS l. 542-554, which is based on the highest BDOC values available for each ecosystem type, it is still not completely clear whether the same conclusions are found.*

To try and remove this confusion with how we compare and interpret BDOC data we have added text to L690-693 and L717-722. We highlight that due to limited data availability we look at the data in two-ways. The first is using BDOC from all measurement days and compares pristine and disturbed sites. The second approach compares max BDOC for each ecosystem, including pristine and disturbed. Both highlight that Yedoma and permafrost wetland sites experience the highest BDOC

14. *Since there is a systematic increase with time, by ecosystem type in the BDOC data isn't it possible to correct for the length of the incubation experiment in Fig. 6?*

This may not be possible as we find that only Yedoma and permafrost wetland sites change over time (L711-714). Thus, there is not a systematic increase in DOC over time for all sites. BDOC is at its greatest for all sites between 40-90 days, thus on L703-720 we compare maximum BDOC (max BDOC) across ecosystems. Here, we are not interested in the kinetics of DOC degradation which include a time factor. Rather, we are interested in max BDOC which we interpret as a sign of potential lability of that DOC pool to degrade.

*Other comments*

1. *Supplementary Table and Figures are named S1, S2, etc in the SI while they are referenced as A1, A2 in the main text. Some of the figures are not even referenced (this is also pointed out in some comments above and below).*

This mistake has been fixed. All supplementary material is now referred to as Figure Sx or Table Sx

2. *L. 112-114 this belongs to the methods.*

This has been moved to the methods L166-168

3. *L. 238 how did you control for the month? Maybe this is what Fig. S2 is about, would need to be referenced.*

Month was controlled for by including it as the covariate factor in the ANCOVA we performed. We chose to perform an ANCOVA specifically for this reason. We have added additional text to this section to ensure that this point is clearer to readers.

4. *L. 282, why did you keep this reference then?*

This reference is kept as it fitted out criteria to be included using our systematic approach (Table 1 &2). We determined that studies must contain DOC concentration or mobilization data, this study contained DOC export data which we consider to be mobilization.

5. *L. 379-382 the description of the different classes would be better in the text (Methods section)*

We agree, and have moved this to L261-264 in the methods section

6. *L. 383-386 same here.*

Same as above, we have moved this section to the methods (L254-259)

7. *L. 475-477 Have you tested that? "no clear or obvious trends in SoilC, TDN, C:N ratios, and SUVA across ecosystem types"*

This has not been tested, rather it is our interpretation of observations of Figure S3. E have added "observable" to the text to highlight this.

**Reviewer 3**

*General Comments*

*I agree the study provides 'unique and valuable insights' into C dynamics in circumpolar north ecoregions. It pulls together an enormous database of DOC concentrations and fluxes from hundreds of sites to compare export and biodegradability of DOC from different ecosystems. The authors under-sell the scope and breadth of their review.*

*The assessment of sampling approach and DOC analyser method is also interesting. This, alongside the recommendations for study design in the discussion, will be a valuable resource for future studies*

We thank the reviewer for their time in preparing this review of our manuscript and for their help in bettering the paper. We appreciate recognition of scope and breadth of the review and we agree the paper can be an important contribution to the field. We have addressed the reviewers concerns below

*However, there are some issues to be addressed. The main aim of the study not clear, and the introduction doesn't map on to the hypotheses or results, making it hard to determine the reasons behind the hypotheses. I am concerned that significant differences between filter size, extraction and measurement method were not incorporated into the analysis of DOC concentration, as there were clear impacts of these on the DOC concentrations that could explain differences between studies that are currently being attributed to ecosystem or another factor. The figure captions are so long it makes the figures feel really complex, whereas they are actually relatively simple figures showing clear results.*

Many of the reviewers' concerns with the manuscript have also been highlighted by other reviewers and we have addressed them within the manuscript.

We have restructured the introduction to improve the narrative structure and build towards our aims, questions and hypotheses.

Regarding the concerns over DOC extraction and analysis methods, we have added a new section "Section 3.4 Effect of extraction and analysis methods on DOC concentrations". Here (L480-493) we address the impact of these methods on DOC concentrations.

*Specific Comments*

*Abstract*

*Line 24 – do you mean vast pools of water containing SOC (like peat pools/ponds) or vast stores of SOC?*

Here we mean stores of carbon, we have changed "pools" to "stores"

*Introduction*

*Line 56 – Mentioning RCP 8.5 here makes it seem like this is the only scenario in which there will be permafrost melt. Could you make it clear that there will be increased melt across a range of scenarios?*

We have added text to the previous sentence to highlight that widespread thawing is occurring and predicted for current and several future climate scenarios with the reference Olefeldt et al., 2016

Olefeldt, D., Goswami, S., Grosse, G., Hayes, D., Hugelius, G., Kuhry, P., … Turetsky, M. R. (2016). Circumpolar distribution and carbon storage of thermokarst landscapes. Nature Communications, 7, 13043. https://doi.org/10.1038/ncomms13043

*Line 61 – I think 'additionally' would be more appropriate than 'alternatively' here, as surely C will be lost via both pathways?*

We have changed "Alternatively" to "Additionally"

*Line 83 – Do you mean 'deepening' here?*

Yes, thanks for catching that. We have changed it to "Deepening"

*Line 101 – 'top 3 m of terrestrial ecosystems' – it isn't clear what you mean here.*

We have added in soil as we are discussing the top 3 m in the soil surface, not including any above ground biomass

*Line 116 – hypothesis 3 –there is nothing in the introduction that explains this hypothesis – why would DOC be most biodegradable from those ecosystems?*

We have now added text, including citations (L113) to the introduction that supports this hypothesis

*Methods*

*Line 138 – why was this additional search carried out on Google Scholar, when it was included in the original search?*

We have removed "Once this initial database was compiled" as the Google Scholar addition were part of the main database

*Table 2 – can you include numbers of studies that were included after each screening stage? Or is this information included in the results?*

We have included how many papers were included at the start (577) and at the end (111) of the screening process on L369. We do not include how many papers were removed at each step of the screening process as when we tried to do so we found that the listing of these numbers created unnecessary confusion in the text. As each screening phase is based on similar criteria, just applied in different ways/detail to the text, the same type of information was used to determine if a paper was included or removed during the screening process. Thus, a beginning and final number are hopefully enough to satisfy the reader.

*Table 2 – several of these criteria seem quite similar to those in Table 1 (e.g. language, type of study), can you make it clear what was screened at each stage?*

Yes, this is correct. Table 1 identifies the preliminary screening stage. This screening stage was enforced during our searches of the database search engines. This is mentioned on L144. Table 2 represents the screening terms when reading the title, abstract, keywords (primary screening), and also in the main body of the text (secondary screening). This is mentioned on L152-156.

*Line 185 – why did you remove DOC concentrations over 500 mg L⁻¹?*

We removed DOC concentrations above this limit as they are uncommon and act as outliers. They represented <2% of all DOC concentrations and are unlikely to be found *in situ*

*Line 202 – why did you choose 20% as the cut-off point?*

We chose 20% as the cut off point to try and include as many continuous variables as possible. The proportion itself (20%) is rather arbitrary so when deciding on picking a cutoff point this seemed as good a point as any, given we could then include as many continuous variables as possible.

*Line 215 – there are a lot of classification types and categories listed here – could you include more detail or definitions, or an example of a site classification to show how they work together? e.g. DOC sample from site X from continuous permafrost in Arctic tundra, on histosol soil, from permafrost-free horizon.*

We have added text to this section (L273-287) to better describe each ecosystem type. We have also added text in the section above (L254-259 and L261-264) to better describe the thermal horizons and soil types.

*Line 239 – this definition of permafrost lens and active layer samples in amongst the data analysis is strange – could it be moved to the section above where the categories and classifications are introduced?*

This has been moved up to L254-258 to where we introduce active layer and permafrost lens thermal horizons.

*Results*

*Line 282 – why was the Olefeldt et al study included if it did not report DOC concentrations?*

In our search criteria (Table 1 and 2), we were focused on extracting data pertaining to DOC concentrations or mobilization rates. This study, while not containing DOC concentrations did provide DOC mobilization rates. Therefore, it met the inclusion criteria outlined in Tables 1 and 2, and remains in the database.

*Line 291 – if the number of DOC mobilisation measurements is so low and therefore the results are not considered, why is a lot of the introduction about mobilisation? It made it seem like mobilisation would be a focus of the study.*

We agree, we had hoped to extract more data on mobilization rates during the systematic review process. We have restructured the introduction and while we still discuss mobilization it is not a key component of it.

*Line 310 – this shows a significant effect of filter size on DOC concentration, yet you state "we consider the effects of filter size to be minor" without any justification. Please explain why you assessed the impact of filter size and then discount the result.*

We agree and this was a concern of multiple reviewers. To address this, we have added a new section to the results (3.4 Effect of extraction and analysis methods on DOC concentrations) where we have addressed these concerns.

*Line 312 – I understand that these results were not the focus of the study, however they are interesting and worth reporting. Could you include a table to show the number of DOC concentrations or studies by each filter size, extraction method and measurement methods? It could go in the supplementary information. I think it will help future studies choose which method to use, and be highly citable.*

We agree and think this to be a good idea. Rather than creating a table, we have added the proportion of DOC concentrations measured using each different approach (L400-413). This information is then used to back up our suggestion of a standardized approach (L1037-1041)

*Line 344 – linking ecoregion to latitude – could you include an average (mean, median, whichever is appropriate) of the latitude of each ecoregion if you are linking that to DOC concentration?*

We don't think there is an appropriate mean/median for each ecoregion as they differ across the circumpolar permafrost region. For example, the southerly extent of the boreal forest differs between Scandinavia and Canada/Alaska. The ecoregions are decided upon from Olson et al., (2001) (cited on L250). We have added in the text (L454-455) to address that by the latitudinal effect seen across ecoregion we mean that from north to south there is arctic tundra to sub-arctic tundra, to boreal (both continental and sub-arctic).

*Line 346 – could you standardise/shorten the way you report median, LQ and UQ? The current way leads to long chunks of text/numbers in brackets that break up the sentences and make it hard to follow. The n values for each permafrost zone, ecoregion, soil class and thermal horizon are in Figure 2, so you don't need to put those in the brackets. Something like (58 (20-107) mg L$^{-1}$) would be shorter than (n = 442; 58 mg L-1; LQ = 20 mg L-1; UQ = 107 mg L-1) as it is currently.*

Yes, we agree and thank the reviewer for this suggestion. To make it easier for the reader we now include median ± interquartile range. For example, above the reviewer uses (n = 442; 58 mg L-1; LQ = 20 mg L-1; UQ = 107 mg L-1). Now in the manuscript this is reported as 58 ± 87 mg L. We have added this information on how we report median and values and the interquartile range to L315-317

*Figure 2 – this is good, clear way to represent database findings*

Thank you

*Line 406 – Table S1 states there were 145 DOC concentrations from 9 studies in the Yeodoma ecosystem, whereas the text here says 118 DOC concentrations from 9 studies.*

Correct, thank you for catching this error. It also highlighted other errors in the number of concentrations per ecosystem we report. We have now updated Table S1 we the correct number of concentrations per ecosystem and references.

*Line 427 – this paragraph is so difficult to read as there are so many numbers in the text.*

Hopefully the adjustments to how we report median values has improved readability in this context.

*Figure 4 caption – there are seven categories shown in the plot, out of eight total. In the caption, you state that you have not shown results for peatland or permafrost-free sites, which would make nine categories in total. Can you make sure you are consistent with how the ecosystem categories are reported?*

We have removed peatland from the text here to improve clarity. Now, we say that all sites without permafrost are not included here. These would include thermokarst areas where complete thawing of permafrost has occurred from any of the ecosystem sites in the figure, or peatland sites (which do not contain any permafrost or reported permafrost history)

*Line 451 – could this go into the methods data analysis section? It is describing the analysis rather than results.*

We think this is more a reporting of a result of the database and relevant to this results section. If moved to the methods section, then it would come before Section 3.1 Database generation, which we think would be confusing and out of order.

*Line 468 – this sentence doesn't make sense "The positive relationship between DOC and total dissolved nitrogen soil carbon content (SoilC)…"*

We have reworded this to "The positive relationship of DOC with total dissolved nitrogen and soil carbon content (SoilC)."

*Line 470 – Some of this is discussion rather than results. The sentence about aromatic content makes it sound like you had data of aromatic content to compare with SUVA values.*

We have moved the text describing the relationship between SUVA and aromatics, and C:N and decomposition, this to the methods section (L237-241)

*Line 476 – you mention figure A3 (which is called S3 in the SI) but not figure A2 anywhere in the text.*

Thanks for picking this up. All instances of using A instead of S to indicate supplementary have now been changed

*Line 489 – the figure captions are so long. I didn't think the figures were that complicated, but the caption makes it seem much more complex.*

We understand that the captions are quite long and the figures themselves are rather simple. However, given the wide range of site categorical variables and continuous variables presented (eg in Figure 2 and 5) we think that shortening them would make their interpretation more difficult for the reader. Where possible we have removed text that is unnecessary from the captions to help address this concern.

*Line 511 – mobilisation of DOC – I thought there wasn't enough data to assess mobilisation?*

Our aim was to develop a database that can address DOC concentrations and mobilization across the circumpolar permafrost region. There was limited data on DOC mobilization but we still thought it important to analyze this. We do not include any analysis on DOC export or lateral flow. We do however include analysis of BDOC as we had data on this. We have added a caveat to our analysis on L690-693 and L717-722 to highlight that this should be interpreted with caution due to low sample size.

*Line 512 – I am confused by this section – it seems to be repeating results from section 3.2. Can you make it clear which characteristics/properties are assessed in each section?*

The distinction between this and section 3.2 is that in section 3.6 we are comparing DOC concentrations across ecosystems that are pristine or disturbed, i.e., after thermokarst formation. The aim of this section is to assess the response of DOC concentrations to thermokarst formation, thus the title of this section is "3.6 Response and mobilization of DOC and BDOC to thermokarst formation" In 3.2 we only compare across study regions, one of which includes ecosystems (data is taken from both pristine and disturbed sites for each ecosystem) and is titled "3.2 DOC concentrations and study regions"

*Line 530 – these BDOC results are interesting, yet it feels like they are being buried in this section.*

Due to the low number of BDOC data available, we do not move it up. We have added text to this section to caveat these results, but we agree that they are interesting and are sorry that there was not more data available to increase the importance of them. We include a recommendation to gather more of this data in future studies (L1013-1015).

*Discussion*

*Line 598 – again, referring to the top 3 m is not clear what you mean. Do you mean the top 3 m of soil? Were plants included in height?*

Yes, the top 3 m of soil. We have changed this to include of soil

*Line 618 – were these studies included in your review?*

No, the inclusion criteria (Table 1 and 2) of our systematic approach excluded all review articles.

*Line 631 – were the thermal layers consistently deep across all sites?*

No, these layers will be heterogenous both across and within sites.

*Technical Corrections*

*Line 203 – brackets aren't quite right here*

This has been changed

*Line 343 – "violin plots of both he discontinuous…" he should be "the"*

This has been changed

*Line 402 – should Table S1 be referenced here?*

Yes, we have added reference to Table S1 here

*Figure 4 caption – "and (b) (b) the number…" too many (b).*

We have removed this duplication

*Line 811 – "Whereas the database…" this sentence does not make sense.*

Thanks for catching this. We have changed this text

**Tatiana Raudina**

*General comments: I would like to thank the authors for attempting to systematize these databases. A lot of effort and time was spent collecting and processing them, and it is wonderful that they are considered in such kind of study (review article). However, I would like to leave some comments for possible improvement of accessibility of the data and their interpretation.*

We would like to thank the reviewer for the considerate and helpful review of the manuscript. Especially given their involvement with several papers included in building the database. Your input in very valuable and we have carefully considered your comments. Thank you for your time and efforts towards this manuscript.

*Detailed Comments*

*1. Line 215 - Sites were classified according to ecosystem type, and these included coastal tundra, forest, peatland, permafrost bog, permafrost wetland, retrogressive thaw slump, upland tundra, and Yedoma. Ecosystem classification is based on the general site description in the article, the*

*provided ecosystem classification within the article, and site data including vegetation composition, permafrost conditions, and ecoregion.*

*I am wondering how peatland, permafrost bog, permafrost wetland were identified? A wetland is an area saturated with water and includes various aquatic ecosystems. In your research, you focused on terrestrial ecosystems. Was there some narrower sampling within the wetlands or were all heterogeneities and water bodies taken into account, including lakes, hollows, streams, etc.? Because this could also be the reason that permafrost wetland had the lowest DOC concentrations (7-10 mg L-1). Another question regarding peatlands. Were they permafrost-affected and where was the main location of these ecosystems? The authors that "Our goal was to assess the concentration and mobilization of DOC in terrestrial permafrost ecosystems", however, in Fig.1 there is only one peatland sampling location. For a better understanding, it would be helpful to provide clear definitions/descriptions of these ecosystems and what specific landscapes you included within them.*

We have added text to section 2.3 database generation (L275-289) where we provide more detail on each ecosystem classification and citations to each ecosystem type. We identified each ecosystem type based on the information provided by the authors, their classification, and our interpretation of that description. In general, the authors classification of the ecosystem was used as it agreed with our interpretation. We tried to keep these classifications broad. We understand that there are differences in DOC concentrations within sites and across microtopographic features. However, we did not aim to assess variability at this smaller geographical-scale. We agree that this is an important step in assessing DOC concentrations across the circumpolar permafrost region, but outside the scope of this study. We hope future studies will build upon this initial attempt to address this question.

With regards peatlands and permafrost bogs, we include permafrost bogs to be sites with intact permafrost (and the active layer and permafrost lens thermal horizons) or have a relatively recent history of permafrost where the bog is classified as a thermokarst bog. That is, it has a clear post thaw autogenic ecosystem succession trajectory and differs from the intact, permafrost bog it used to be prior to thawing.

*Accordingly, have you somehow considered the microtopography/spatial heterogeneity of your ecosystems (for example, mounds/fens/hollows/etc.)? Please note that even within the same type of ecosystems, DOC concentrations will differ by more than two times (e.g.: https://www.mdpi.com/2076-3263/9/7/291 or https://doi.org/10.5194/bg-14-3561-2017). It is not clear whether you averaged (or provided median values) for all the data within one ecosystem?*

As mentioned above, analysis of DOC concentrations across such small-scale geographically heterogeneity was not included in our study. When possible, we recorded each microform and feature from where sampling was conducted and this informration is included in our database. We hope that this study and the public availability of our database will be used to generate future study ideas around this topic.

*2. Line 209 – We also included the soil class found at the site (Histel, Histosol, Orthel, and Turbel; USDA, 1999) and whether the DOC was from the organic or mineral soil. You also mentioned that organic layer depth was included and that the highest DOC concentrations are found within organic-rich Histosol and Histel soils.*

*Have these variables (samples taken from mineral and organic horizons) been considered in the PLS? Since there is no information about these variables in the description of the results. Aren't they significant drivers of DOC concentrations? In general, the result of the PLS is not clear (Line 464). That is, the DOC concentrations are largely determined by the TDN and C:N ratio?*

We have added text to section "2.4 Database analysis" in an attempt to address the concerns of multiple reviewers. Our goal with the PLS was to assess main drivers of DOC concentration across ecosystems, including disturbance and thermal horizons (with these included as categorical variables) and continuous environmental data with >20% coverage of the DOC data. We agree that mineral vs organic soil would be a driver of DOC concentrations, but rather we were more interested in exploring the relationship between DOC and continuous variables within ecosystems. The aim of this was to address how disturbance and ecosystem effects drive DOC across the circumpolar permafrost region. We chose the most parsimonious PLS model based on VIP scores. The model included in the manuscript represents that which contains the best predictor variables. We have also reworded how we discuss the results of the PLS.

*3. Line 312 - DOC concentrations were found to be significantly different between samples subject to the 11 different extraction methods used (ANOVA: $F_{(10, 2515)} = 21.8$, $p < 0.001$), and between water based and soil (solid) based extraction methods (ANOVA: $F_{(1, 2524)} = 182.1$, $p < 0.001$).*

*Indeed, the extraction method, sample subject location, as well as the size of the filter, have a large impact on the DOC concentration. Why then did you decide not to focus on extraction methods and filter sizes? After all, this is a review article and must take such important parameters into account. For a better understanding, it is worth explaining what specific extraction methods were used?*

We agree and this point has rightly been brought up by several authors. We did not explicitly focus on method as it was not our original aim of assessing DOC concentrations and mobilization rates across the northern circumpolar permafrost region. It is still not an aim of the paper, but we do agree that it is a factor that needs to be considered. To address this concern, we have added a new section to the results "3.4 Effect of extraction and analysis method on DOC concentrations" and Table S2-7 in the supplementary highlighting the differences in methods.

*4. Line 403 - According to the text it follows that "The majority of permafrost wetland sample locations were found in Russia. However, Fig. 1 shows different information and most ecosystem types are permafrost bog. At the same time, most of the studies done in Russia, which you included to generate database deal with permafrost bog or permafrost peatlands (e.g.:*

*https://doi.org/https://doi.org/10.1016/j.chemgeo.2017.10.002*

*https://doi.org/10.1016/j.scitotenv.2018.04.059*

*https://doi.org/10.1016/j.chemosphere.2020.128953*

*https://doi.org/https://doi.org/10.1016/j.chemgeo.2013.07.016 and others.*

This is due to the way our map generation overlays certain data points. As Permafrost Bog is listed higher when plotting the points on the map it covers the Permafrost Wetland points. We have attempted several versions of this map to try include as many points as possible but it becomes over crowded and difficult to interpret. We have added

text to the figure caption to highlight this and link readers to the database which contains coordinates for all sites. We decided to show rather the geographical spread of sites.

*It is also worth noting that there is not a single reference to studies from Western Siberia in the text (although Fig. 1 shows that quite a lot of DOC concentrations were taken from sites in Western Siberia).*

Good catch, we have added text to the manuscript supported by references from this area to L955 and L966

*5. Line 808 - Our results suggest that the high concentrations of DOC in permafrost bogs remains relatively stable upon thermokarst formation, although individual studies do indicate that thawing peat may provide a reactive source of DOC (Panneer Selvam et al., 2017).*

*How do you explain your results? Because the thawing of frozen peat can actually lead to a large release of DOC and macro- and microelements (e.g.:*

[https://www.sciencedirect.com/science/article/abs/pii/S0045653520331507](https://www.sciencedirect.com/science/article/abs/pii/S0045653520331507)

Our results only indicate that DOC concentrations remain stable, or rather net DOC remains the same. Indeed there may be a large release of DOC following thaw, as some studies have shown. However, this may be a labile pool that is rapidly mineralized and lost, thus the new carbon sources are lost to the atmosphere as they are laterally transported and only aged material (DOC and POC) reaches the aquatic network. For example, Burd et al., (2020) show that DOC is rapidly mineralized following thermokarst formation in a boreal peatland complex. However, they conclude that this is likely due to the rapid turnover of new DOC derived from vegetation that has colonized post thaw rather than the mineralization of previous frozen material. We acknowledge a discrepancy between individual studies and the results in our database, and our main response is that more studies across the entire permafrost region on the lability of newly exposed DOC are needed. Perhaps an interesting discussion point we can have some day.

Burd, K., Estop-Aragonés, C., Tank, S. E., & Olefeldt, D. (2020). Lability of dissolved organic carbon from boreal peatlands: interactions between permafrost thaw, wildfire, and season. Canadian Journal of Soil Science, 13(February), 1–13. https://doi.org/10.1139/cjss-2019-0154

---

## Referee Report (RR1)

**Cryosphere journal review:**

Review article: Terrestrial dissolved organic carbon in northern permafrost

Liam Heffernan, Dolly N. Kothawala, Lars J. Tranvik

tc-2023-152

**General Comments**

I thank the authors for their reply to my comments and concerns about the manuscript. The changes made in response to all reviewers comments have improved the manuscript. I've noted a few typos and minor changes that could be made below, but on the whole, I think it is a good paper that will add to the body of evidence on DOC concentrations. I also think it is important to highlight the data gaps found by compiling this database. The lack of mobilisation measurements and data show a clear gap in the C cycle of northern peatlands.

**Specific Comments**

**Abstract**

The DOC concentrations in lines 36 to 40 are confusing: Are these maximum concentrations? "DOC concentrations were greatest in the sporadic permafrost zone (101 mg L-1) while lower concentrations were found in the discontinuous (60 mg L-1) and continuous (59 mg L-1) permafrost zones."

Then reporting median values as 'highest medians'? These values are smaller than those in the previous sentence.

"The highest median DOC concentrations of 66 mg L-1 and 63 mg L-1 were found in coastal tundra and permafrost bog ecosystems, respectively. Coastal tundra (130 mg L-1), permafrost bogs (78 mg L-1), and permafrost wetlands (57 mg L-1) had the highest DOC concentrations in the permafrost lens, representing a potentially long-term store of DOC."

**Introduction**

Paragraph 1 makes the purpose of the paper really clear

Line 77 typo 'concentrations'

**Method**

The literature search method is mostly clear. I'm still not sure why the search on Google Scholar is mentioned twice in section 2.1 though. Could Tables 1 and 2 could be combined into one table, reducing repetition?

Was the carbon:nitrogen ratio (mentioned in line 207) a ratio of DOC:DON, or soil C content:soil N content?

Line 212: "Ratios in C:N have been shown…" do you mean decomposition rate, or potential? Or how decomposed the soil already is?

Line 220 – 269: Could some of the site, ecosystem and soil info be put into a table? The list in prose is quite hard to follow as it contains so much information. I appreciate some of this text was added in response to a query in my first review.

**Results**

I appreciate the addition of section 3.4 to the results, and the change in the way median and interquartile ranges are reported.

Line 489 typo: acro

Line 505 typo: in in

**Discussion**

Line 798 typo: assessed..

---

## Author Response (AR2)

*Review article: Terrestrial dissolved organic carbon in northern permafrost*

*Liam Heffernan, Dolly N. Kothawala, Lars J. Tranvik*

*tc-2023-152*

*General Comments*

*I thank the authors for their reply to my comments and concerns about the manuscript. The changes made in response to all reviewers comments have improved the manuscript. I've noted a few typos and minor changes that could be made below, but on the whole, I think it is a good paper that will add to the body of evidence on DOC concentrations. I also think it is important to highlight the data gaps found by compiling this database. The lack of mobilisation measurements and data show a clear gap in the C cycle of northern peatlands.*

We thank the reviewer for taking the time to provide us with this follow up review of the manuscript. We are happy that you are satisfied with our response to theirs, and the other reviews, concerns and comments. Below we have addressed their detailed comments.

*Specific Comments*

*Abstract*

*The DOC concentrations in lines 36 to 40 are confusing: Are these maximum concentrations? "DOC concentrations were greatest in the sporadic permafrost zone (101 mg L-1) while lower concentrations were found in the discontinuous (60 mg L-1) and continuous (59 mg L-1) permafrost zones."*

*Then reporting median values as 'highest medians'? These values are smaller than those in the previous sentence.*

*"The highest median DOC concentrations of 66 mg L-1 and 63 mg L-1 were found in coastal tundra and permafrost bog ecosystems, respectively. Coastal tundra (130 mg L-1), permafrost bogs (78 mg L-1), and permafrost wetlands (57 mg L-1) had the highest DOC concentrations in the permafrost lens, representing a potentially long-term store of DOC."*

We have added text in the abstract to clear this up. Now we state that across permafrost zones the highest median concentration of DOC is found in the sporadic, but across a[permafrost zones there is variability in each ecosystem. Then we say that across ecosystem type the highest median DOC were found in coastal tundra and permafrost bog sites. Hopefully this clears things up

*Introduction*

*Paragraph 1 makes the purpose of the paper really clear*

We thanks the reviewer for this comment

*Line 77 typo 'concentrations'*

Changed

*Method*

*The literature search method is mostly clear. I'm still not sure why the search on Google Scholar is mentioned twice in section 2.1 though. Could Tables 1 and 2 could be combined into one table, reducing repetition?*

We have removed the second mention of using Google Scholar

We understand that there is repetition between the two tables but would rather to keep them separate. Table 1 is the criteria used when searching for papers on the electronic databases. Table 2 is the criteria used once these potentially relevant papers were identified. They are two very different but important steps in the process and we do not want to confuse them by combining.

*Was the carbon:nitrogen ratio (mentioned in line 207) a ratio of DOC:DON, or soil C content:soil N content?*

This is thee soil carbon:nitrogen ratio, we have added soil to the text to clarify this

*Line 212: "Ratios in C:N have been shown…" do you mean decomposition rate, or potential? Or how decomposed the soil already is?*

We have added that this indicates that decomposition has previously occurred

*Line 220 – 269: Could some of the site, ecosystem and soil info be put into a table? The list in prose is quite hard to follow as it contains so much information. I appreciate some of this text was added in response to a query in my first review.*

We have added a table to the supplementary that includes all categorical variables included in analysis of the database

*Results*

*I appreciate the addition of section 3.4 to the results, and the change in the way median and interquartile ranges are reported.*

Thanks, we think that this new way to report them is much clearer and appreciate prior input

*Line 489 typo: acro*

This has been removed

*Line 505 typo: in in*

Second "in" has been removed

*Discussion*

*Line 798 typo: assessed..*

Changed